# Learning the Best Under Constraints: A Duality-Based Framework

**Mingjie Hu** [1 2]  **Enlu Zhou** [2]  **Jianqiang Hu** [1]

## Abstract

This paper studies a constrained linear best arm identification problem with covariate selection in the fixed-confidence setting, where each arm is evaluated across multiple performance metrics. The mean performance of each metric depends linearly on the feature vectors of both arms and covariates. The goal is to identify the arm with the highest expected value of one targeted metric while ensuring that the means of the remaining metrics stay below specified thresholds for each covariate. We first establish an instance-dependent lower bound on the sample complexity, formulated as a multi-level optimization problem that captures both feasibility and optimality. We then prove that this bound is tight by designing an algorithm that asymptotically matches it. Since the original algorithm is computationally intensive, we develop a relaxed version of the bound through a surrogate optimization problem and derive its convex dual. Using this bound, we propose a duality-based decomposition algorithm that is computationally efficient, updating only two coordinates and performing a single gradient step per iteration. We further show that the algorithm achieves the relaxed bound in theory and demonstrates its practical effectiveness through numerical experiments.

## 1. Introduction

Best arm identification (BAI) is a well-studied problem in machine learning, with broad applications in areas such as large language models (Shi et al., 2024), quantum computing (Wanner et al., 2025), and pharmaceutical development (Wang et al., 2024). This paper studies a constrained linear BAI problem with covariate selection. In this set-

ting, each arm is evaluated across multiple performance metrics, where the mean of each metric is modeled as a linear function of feature vectors associated with both arms and covariates. Given a specific covariate, the goal is to identify the arm with the highest expected value in a target metric, while ensuring that the means of the remaining metrics remain below predefined thresholds. At each time step $t$, the agent selects an arm-covariate pair to sample and observes an independent random performance vector covering all metrics. In the fixed-confidence setting, the agent seeks to learn the underlying performance functions through sampling, identify the best arm for each covariate with probability at least $1-\delta$, and minimize the total number of samples required.

Compared to the canonical BAI setting, constrained linear BAI with covariate selection is particularly well-suited for personalized decision-making problems. For example, in personalized medicine (Shen et al., 2021), each treatment option (arm) is associated with multiple performance metrics, such as therapeutic efficacy and side effects, which can only be observed through noisy clinical trial data. The mean outcome of each metric depends on both patient characteristics (covariates) and the chemical composition of the drug. The objective is to identify the drug with the highest expected efficacy while ensuring that the expected side effects remain below predefined thresholds. Similar scenarios arise in inventory management (Ban & Rudin, 2019), where metrics like revenue, lead time, and customer satisfaction depend on observable factors such as seasonality, economic indicators, and market conditions, as well as the chosen order quantity. The goal is to identify the order quantity that maximizes average revenue while ensuring that the mean values of the other metrics remain within acceptable limits.

Two key challenges set constrained linear BAI with covariate selection apart from the canonical BAI problem (Garivier & Kaufmann, 2016), making existing algorithms insufficient for this setting. First, unlike the standard BAI framework, which focuses solely on identifying the optimal arm, the constrained version requires balancing both optimality and feasibility. This trade-off between optimality and feasibility requires new theoretical insights to understand its effect on sample complexity and to guide the design of optimal algorithms. Second, covariate selection introduces an additional layer of complexity. The agent must determine an

---

[1]School of Management, Fudan University, Shanghai, CHINA [2]H. Milton Stewart School of Industrial and Systems Engineering, Georgia Institute of Technology, Atlanta, USA. Correspondence to: Mingjie Hu <humj21@m.fudan.edu.cn>.

*Proceedings of the 43rd International Conference on Machine Learning*, Seoul, South Korea. PMLR 306, 2026. Copyright 2026 by the author(s).

optimal sampling rule over arm-covariate pairs at each iteration. In contrast, canonical linear BAI (Jedra & Proutiere, 2020) and contextual bandit settings (Slivkins et al., 2019) typically assume that covariates are passively observed, limiting the agent's control to selecting a single arm. As we demonstrate in this work, leveraging both linear structure and active covariate selection can significantly improve sampling efficiency and necessitates a fundamentally different algorithmic approach.

The contributions of this paper are summarized as follows:

- Motivated by practical personalized decision-making scenarios, we study a constrained BAI problem with covariate selection. We derive an instance-dependent lower bound on the sample complexity, formulated as a multi-level optimization problem, and characterize how both the feasibility and optimality of each arm influence this bound. Moreover, we demonstrate the tightness of this bound by constructing a Track-and-Stop algorithm whose sample complexity matches it asymptotically.

- Due to the computational intractability of the Track-and-Stop algorithm, we introduce a relaxed sample complexity bound derived from a surrogate optimization problem. We further derive its convex dual, which possesses favorable structural properties and can be solved efficiently. Notably, the dual formulation provides a closed-form mapping to the primal optimal solution and offers an intuitive interpretation of the optimal sampling ratio.

- Leveraging the specific structure of the dual problem, we propose a duality-based decomposition algorithm. This algorithm has two key features: first, it updates two coordinates of the dual solution at a time; second, it performs a one-step gradient descent at each iteration. These features contribute to its high efficiency. We theoretically demonstrate that the algorithm's sample complexity attains the relaxed bound and validate its practical effectiveness through numerical experiments.

Our study connects to three principal strands of the existing literature:

**Best Arm Identification.** BAI is one of the most extensively studied problems in the bandit literature (Audibert & Bubeck, 2010; Gabillon et al., 2012). This work contributes to the growing body of research on BAI in the fixed-confidence setting, also known as pure exploration (Kaufmann et al., 2016; Garivier & Kaufmann, 2016; Juneja & Krishnasamy, 2019; Degenne & Koolen, 2019), which focuses on deriving instance-dependent lower bounds on sample complexity and designing adaptive, asymptotically optimal

algorithms (Degenne et al., 2019; Wang et al., 2021). Jedra & Proutiere (2020) extended these results to the linear BAI setting. Our formulation generalizes both the canonical and linear BAI problems as special cases. Furthermore, the proposed algorithm introduces a duality-based perspective, enhancing both efficiency and practicality compared to methods that rely on access to an optimization oracle.

**Constrained Best Arm Identification.** The multi-performance constrained BAI problem has received relatively limited attention in the literature. While recent studies have begun exploring multi-objective settings aimed at identifying the Pareto set (Kone et al., 2023; 2024b;a; 2025), these problems are fundamentally different from our constrained formulation, and the algorithms proposed in those works are not applicable to our setting. Yang et al. (2025) and Hu & Hu (2024) consider constrained BAI problems that are more closely related to ours. However, Yang et al. (2025) proposes a top-two Thompson sampling algorithm under a fixed-budget setting, without leveraging linear structure or considering covariate information, resulting in a simplified optimization problem compared to our setting. Meanwhile, Hu & Hu (2024) primarily focuses on risk constraints rather than the mean-based constraints studied here, and their algorithm is not readily adaptable to our framework.

**Covariate Selection.** Decision-making with covariate information has been a central research theme across various domains, including operations research (Bertsimas & Kallus, 2020), simulation optimization (Shen et al., 2021; Du et al., 2024), and bandit problems (Lattimore & Szepesvári, 2020; Kato & Ariu, 2021). However, the covariate selection problem studied in this paper differs from the classical contextual bandit setting, where covariates are observed passively and drawn randomly. Kato et al. (2024) investigates covariate selection in the context of experimental design, focusing on minimizing the semi-parametric efficiency bound. In contrast, we extend the notion of covariate selection to the BAI setting under the fixed-confidence framework, with the objective of minimizing the sample complexity.

## 2. Problem Formulation

This section presents the formulation of the constrained BAI problem with covariate selection and introduces the notation used throughout the paper.

Consider $K$ different arms, denoted by $\mathcal{X} = \{x_1, \ldots, x_K\}$, where each arm is associated with a vector $x_i$. We assume a finite set of $M$ possible covariates, denoted by $\mathcal{C} = \{c_1, \ldots, c_M\}$. For problems involving continuous covariate spaces, it is common to discretize the feature space and group covariate values accordingly. The performance of arm $x_i$ under covariate $c_j$ is represented by a random vector

$(F(x_i, c_j), G(x_i, c_j)) \in \mathbb{R}^2$, where $F(x_i, c_j)$ and $G(x_i, c_j)$ correspond to the objective-related and constraint-related performance metrics, respectively. The agent aims to solve the following stochastic optimization problem:

$$\max_{x_i \in \mathcal{X}} f(x_i, c_j) \triangleq \mathbb{E}[F(x_i, c_j)]$$
$$\text{s.t.} \quad g(x_i, c_j) \triangleq \mathbb{E}[G(x_i, c_j)] \leq b, \quad (1)$$

for each covariate $c_j \in \mathcal{C}$. For notational simplicity, we consider a single-constraint setting. Extending our theoretical results and algorithm to accommodate multiple constraints is straightforward (see Appendix A.3). A problem instance is defined as

$$\mathcal{P} = (f(x_i, c_j), g(x_i, c_j))_{x_i \in \mathcal{X}, c_j \in \mathcal{C}}.$$

To facilitate the analysis, we adopt the following standard assumptions, which are commonly used in the BAI literature.

**Assumption 2.1.** The problem instance $\mathcal{P}$ belongs to the regular set $\mathcal{S}$ of instances such that, for each covariate $c_j \in \mathcal{C}$, there exists a unique best arm $x_{i^*(c_j)}$ that solves problem (1), and no arm lies exactly on the constraint, i.e.,

$$g(x_i, c_j) \neq b, \forall x_i \in \mathcal{X}.$$

**Assumption 2.2.** For each arm-covariate pair $(x_i, c_j) \in \mathcal{X} \times \mathcal{C}$, the mean performances are given by

$$f(x_i, c_j) = \theta^\top \phi(x_i, c_j)$$

and

$$g(x_i, c_j) = \beta^\top \phi(x_i, c_j),$$

where $\phi(\cdot, \cdot) : \mathcal{X} \times \mathcal{C} \to \mathbb{R}^D$ is a known feature map, and $\theta, \beta \in \mathbb{R}^D$ are unknown parameter vectors.

**Assumption 2.3.** The observed performances are given by

$$F(x_i, c_j) = f(x_i, c_j) + \epsilon_{ij}$$

and

$$G(x_i, c_j) = g(x_i, c_j) + \epsilon'_{ij},$$

where the noise terms $\epsilon_{ij}$ and $\epsilon'_{ij}$ are independent Gaussian random variables with mean zero and variance $\sigma_{ij}^2$.

Assumption 2.1 is standard in the canonical BAI literature (Garivier & Kaufmann, 2016; Jedra & Proutiere, 2020). Extending our framework to the standard $\epsilon$-optimal and feasible BAI setting in Degenne & Koolen (2019) requires substantial new technical developments and is beyond the scope of this paper. Assumption 2.2 imposes a linear relationship between the mean performances and feature vectors. Despite its simplicity, the linear model effectively captures structural relationships across arms and covariates, enhances interpretability, and is widely used in linear bandit

problems (Soare et al., 2014; Jedra & Proutiere, 2020) as well as personalized medicine (Shen et al., 2021; Du et al., 2024). Lastly, the Gaussian noise assumption in Assumption 2.3 is a standard choice in classical linear regression and enables the derivation of a tractable characterization of the sample complexity lower bound.

**Design points.** In this paper, we use a fixed set of design points, denoted by $\mathcal{Z} = \{z_1, \ldots, z_D\}$, to estimate $\theta$ and $\beta$. We assume that the feature vectors associated with these design points span $\mathbb{R}^D$. Each design point $z_h$ corresponds to an arm-covariate pair $(x_i, c_j) \in \mathcal{X} \times \mathcal{C}$, and we simplify the notation by writing $F(z_h) = F(x_i, c_j)$ and $G(z_h) = G(x_i, c_j)$. The motivations for adopting a fixed set of design points can be categorized into three aspects. First, De la Garza (1954) shows that to estimate the $D$-dimensional parameters $\theta$ and $\beta$ via regression, sampling a properly chosen set of $D$ linearly independent design points can be sufficient for estimating these parameters. Second, this formulation has been widely used in the transductive linear bandits literature (Fiez et al., 2019). Third, concentrating on a fixed set of $D$ design points allows for the decomposition of regression variance, which facilitates the design of efficient algorithms.

**Learning problem.** In the online setting, at each iteration $t$, the agent selects a design point $z_{h(t)} \in \mathcal{Z}$ to sample. It then observes a random performance vector $Z_t = (Z_t^{(1)}, Z_t^{(2)})$, drawn independently according to the distribution of the corresponding random vector $(F(z_{h(t)}), G(z_{h(t)}))$. An algorithm in this setting is characterized by three components: the sampling rule $\{z_{h(t)}\}_t$, which determines the design point to sample based on the historical sampling decisions and observations up to time $t$; the stopping rule $\tau_\delta$, which decides when to terminate the algorithm based on the collected information and the confidence level $\delta$; for notational simplicity, we write $\tau$ instead of $\tau_\delta$ when there is no ambiguity; and the recommendation rule $\{x_{\hat{i}(c_j, \tau)}\}_{c_j \in \mathcal{C}}$, which specifies the recommended best arm for each covariate $c_j \in \mathcal{C}$. The goal is to find a $\delta$-Probably Approximately Correct (PAC) algorithm (see Definition 2.4) while minimizing the sample complexity $\mathbb{E}[\tau]$.

**Definition 2.4** ($\delta$-PAC algorithm). An algorithm

$$\mathcal{L} = (\{z_{h(t)}\}_t; \tau; \{x_{\hat{i}(c_j, \tau)}\}_{c_j \in \mathcal{C}})$$

is said to be $\delta$-PAC if for every problem instance $\mathcal{P} \in \mathcal{S}$, it satisfies

$$\mathbb{P}(\forall c_j \in \mathcal{C}, x_{\hat{i}(c_j, \tau)} = x_{i^*(c_j)}) \geq 1 - \delta. \quad (2)$$

**Notation.** For a positive integer $K$, let $[K] = \{1, \ldots, K\}$. Denote by $N_h(t)$ the number of samples drawn from design point $z_h$ up to time $t$, and define the corresponding sampling ratio $\omega_h(t) = N_h(t)/t$. Let $\Omega \triangleq \{\omega \in \mathbb{R}_+^D : \sum_{h \in [D]} \omega_h = $

1} denote the probability simplex over the design points. Let $\mathbb{I}(\cdot)$ denote the indicator function, which takes the value 1 if the condition is true, and 0 otherwise.

# 3. Sample Complexity

In this section, we first derive a lower bound on the sample complexity. We then introduce a Track-and-Stop algorithm that asymptotically achieves this lower bound. However, this algorithm is computationally expensive, motivating the development of a duality-based approach. This perspective enables the design of a more efficient algorithm, which we present in the next section.

## 3.1. Sample Complexity Lower Bound

This subsection presents a tight, instance-dependent lower bound on the sample complexity $\mathbb{E}[\tau]$, which provides a benchmark for evaluating the performance of any $\delta$-PAC algorithm.

The characterization of sample complexity relies on the transportation lemma from (Kaufmann et al., 2016), which establishes a relationship between the sample complexity, the Kullback-Leibler (KL) divergence between two problem instances, and the confidence level $\delta$. However, the constrained BAI problem with covariate selection is more challenging. Specifically, different types of arms contribute differently to the sample complexity depending on their feasibility and optimality. To capture this effect, we classify the arms into four disjoint categories for each covariate: the best arm $x_{i^*(c_j)}$, suboptimal feasible arms

$$\mathcal{D}_1(c_j) \triangleq \{x_i : f(x_i, c_j) < f(x_{i^*(c_j)}, c_j), g(x_i, c_j) \leq b\},$$

infeasible arms with better performance

$$\mathcal{D}_2(c_j) \triangleq \{x_i : f(x_i, c_j) \geq f(x_{i^*(c_j)}, c_j), g(x_i, c_j) > b\},$$

and infeasible arms with worse performance

$$\mathcal{D}_3(c_j) \triangleq \{x_i : f(x_i, c_j) < f(x_{i^*(c_j)}, c_j), g(x_i, c_j) > b\}.$$

Then, leveraging the linear structure in Assumption 2.2 and the Gaussian noise in Assumption 2.3, we derive a tractable instance-dependent lower bound on the sample complexity in Theorem 3.1.

**Theorem 3.1.** *Under Assumptions 2.1-2.3, for a fixed confidence level $\delta \in (0, 1/2)$, any $\delta$-PAC algorithm applied to the problem instance $\mathcal{P} \in \mathcal{S}$ must satisfy*

$$\mathbb{E}[\tau] \geq \mathcal{H}^*(\mathcal{P}) kl(\delta, 1 - \delta), \tag{3}$$

*which implies*

$$\liminf_{\delta \to 0} \frac{\mathbb{E}[\tau]}{\log(1/\delta)} \geq \mathcal{H}^*(\mathcal{P}), \tag{4}$$

*where*

$$\mathcal{H}^*(\mathcal{P})^{-1} = \max_{\omega \in \Omega} \min_{c_j \in \mathcal{C}} \Gamma(\omega, c_j, \mathcal{P}), \tag{5}$$

*and $\Gamma(\omega, c_j, \mathcal{P})$ is defined as*

$$\frac{1}{2} \min \left[ \min_{x_i \neq x_{i^*(c_j)}} \left( \frac{((\phi(x_{i^*(c_j)}, c_j) - \phi(x_i, c_j))^\top \theta)^2}{\|\phi(x_{i^*(c_j)}, c_j) - \phi(x_i, c_j)\|^2_{\Lambda(\omega)^{-1}}} \mathbb{I}_0 \right. \right.$$
$$\left. + \frac{(b - \beta^\top \phi(x_i, c_j))^2}{\|\phi(x_i, c_j)\|^2_{\Lambda(\omega)^{-1}}} \mathbb{I}(x_i \in \mathcal{D}_2(c_j) \cup \mathcal{D}_3(c_j)) \right),$$
$$\left. \frac{(b - \beta^\top \phi(x_{i^*(c_j)}, c_j))^2}{\|\phi(x_{i^*(c_j)}, c_j)\|^2_{\Lambda(\omega)^{-1}}} \right]. \tag{6}$$

*Here, $\mathbb{I}_0$ denotes the indicator function, i.e.,*

$$\mathbb{I}_0 = \mathbb{I}(x_i \in \mathcal{D}_1(c_j) \cup \mathcal{D}_3(c_j)),$$

*$\Lambda(\omega)$ is the weighted design matrix defined as*

$$\Lambda(\omega) = \sum_{z_h \in \mathcal{Z}} \frac{\omega_h}{\sigma_h^2} \phi(z_h) \phi(z_h)^\top,$$

*and*

$$kl(\delta, 1 - \delta) \triangleq \delta \log\left(\frac{\delta}{1 - \delta}\right) + (1 - \delta) \log\left(\frac{1 - \delta}{\delta}\right).$$

When $\Lambda(\omega)$ is singular, $\|\cdot\|_{\Lambda(\omega)^{-1}}$ is interpreted in the extended sense, with uncovered directions assigned infinite variance. The derivation of the sample complexity result in Theorem 3.1 has an intuitive game-theoretic interpretation: the agent aims to select a randomized sampling strategy $\omega \in \Omega$ that maximizes the KL divergence between two instances, while the environment chooses an alternative instance $\tilde{\mathcal{P}}$ that is difficult to distinguish from $\mathcal{P}$. In the case of Gaussian noise, this formulation yields the closed-form expression in (6). Additionally, the sample complexity is influenced by the feasibility of the best arm $x_{i^*(c_j)}$, the performance of infeasible arms (both better arms in $\mathcal{D}_2(c_j)$ and worse arms in $\mathcal{D}_3(c_j)$), and the optimality of suboptimal feasible arms in $\mathcal{D}_1(c_j)$ as well as infeasible arms with worse performance in $\mathcal{D}_3(c_j)$.

Theorem 3.1 can be viewed as an extension of the linear BAI problem to the constrained setting with covariate selection. When the agent knows that all arms are feasible and there is only one covariate, Theorem 3.1 reduces to the sample complexity result in (Jedra & Proutiere, 2020), making it a special case of our framework.

## 3.2. Sample Complexity Upper Bound

This section demonstrates the existence of an algorithm that asymptotically matches the sample complexity lower bound up to a constant factor in Theorem 3.1 as $\delta \to 0$.

**Definition 3.2** (Asymptotic optimality). An algorithm

$$\mathcal{L} = (\{z_{h(t)}\}_t; \tau; \{x_{\hat{i}(c_j,\tau)}\}_{c_j \in \mathcal{C}})$$

is said to be asymptotically optimal if for every problem instance $\mathcal{P} \in \mathcal{S}$, it is $\delta$-PAC and

$$\limsup_{\delta \to 0} \frac{\mathbb{E}[\tau]}{\log(1/\delta)} \lesssim \mathcal{H}^*(\mathcal{P}). \qquad (7)$$

The intuition behind the algorithm design is as follows. The sample complexity lower bound in Theorem 3.1 depends on the hardness of the problem instance $\mathcal{H}^*(\mathcal{P})$ and the confidence level $\delta$. The quantity $\mathcal{H}^*(\mathcal{P})$ is defined through an optimization problem that yields the optimal static sampling ratio

$$\omega^*(\mathcal{P}) = \arg\max_{\omega \in \Omega} \min_{c_j \in \mathcal{C}} \Gamma(\omega, c_j, \mathcal{P}). \qquad (8)$$

Therefore, an optimal algorithm must ensure that the empirical sampling ratio $\omega(t) = \{\omega_h(t)\}_{h \in [D]}$ converges to the optimal ratio $\omega^*(\mathcal{P})$. For simplicity, we assume throughout this discussion that the optimal ratio is unique; the extension to the non-unique case is straightforward and is detailed in the proof of Proposition 3.3.

Since the problem instance $\mathcal{P}$ is unknown, we must estimate it based on empirical observations. For each design point $z_h \in \mathcal{Z}$, define the empirical estimates of the mean performances $f(z_h)$ and $g(z_h)$ up to time $t$ as

$$\bar{F}(z_h; t) = \frac{1}{N_h(t)} \sum_{s \leq t} Z_s^{(1)} \mathbb{I}(z_{h(s)} = z_h),$$

$$\bar{G}(z_h; t) = \frac{1}{N_h(t)} \sum_{s \leq t} Z_s^{(2)} \mathbb{I}(z_{h(s)} = z_h). \qquad (9)$$

Then, the least squares estimators of the unknown parameters $\theta$ and $\beta$ up to time $t$ are given by

$$\hat{\theta}(t) = \Lambda(\omega(t))^{-1} \sum_{z_h \in \mathcal{Z}} \frac{\omega_h(t)}{\sigma_h^2} \phi(z_h) \bar{F}(z_h; t),$$

$$\hat{\beta}(t) = \Lambda(\omega(t))^{-1} \sum_{z_h \in \mathcal{Z}} \frac{\omega_h(t)}{\sigma_h^2} \phi(z_h) \bar{G}(z_h; t). \qquad (10)$$

Using the least squares estimators in (10), we estimate $\mathcal{P}$ by $\hat{\mathcal{P}}(t)$, calculated from $\hat{\theta}(t)$ and $\hat{\beta}(t)$, and compute the corresponding empirical static ratio $\omega^*(\hat{\mathcal{P}}(t))$.

To ensure that the estimate $\hat{\mathcal{P}}(t)$ converges to the true problem instance $\mathcal{P}$, it is necessary to sample each design point infinitely often. Define the set of undersampled design points up to time $t$ as

$$\mathcal{B}_t = \{z_h \in \mathcal{Z} : N_h(t) < \sqrt{t} - D/2\}. \qquad (11)$$

Consider the following sampling rule

$$z_{h(t+1)} = \begin{cases} \arg\min_{z_h \in \mathcal{B}_t} N_h(t) & \text{if } \mathcal{B}_t \neq \emptyset, \\ \arg\min_{z_h \in \mathcal{Z}} N_h(t) - t\omega_h^*(\hat{\mathcal{P}}(t)) & \text{otherwise} \end{cases},$$

$$(12)$$

which continuously updates the estimate $\hat{\mathcal{P}}(t)$ and adaptively tracks the empirical static ratio $\omega^*(\hat{\mathcal{P}}(t))$. Under this rule, we can show that $\hat{\mathcal{P}}(t) \to \mathcal{P}$ and $\omega(t) \to \omega^*(\mathcal{P})$ as $t \to \infty$.

Finally, we apply the generalized likelihood ratio test method to ensure that the algorithm satisfies the $\delta$-PAC guarantee described in Definition 2.4. Define the stopping rule as

$$\tau = \inf\{t \in \mathbb{N} : t\mathcal{H}(\omega(t), \hat{\mathcal{P}}(t))^{-1} > \rho(t, \delta)\}, \qquad (13)$$

where

$$\mathcal{H}(\omega(t), \hat{\mathcal{P}}(t))^{-1} = \min_{c_j \in \mathcal{C}} \Gamma(\omega(t), c_j, \hat{\mathcal{P}}(t)). \qquad (14)$$

This rule ensures the algorithm terminates once the accumulated empirical evidence exceeds the confidence threshold $\rho(t, \delta)$, thus supporting the $\delta$-PAC guarantee and contributing to its asymptotic optimality, as shown in Proposition 3.3.

This algorithmic framework, known as Track-and-Stop, is widely used to address the BAI problem in various settings (Garivier & Kaufmann, 2016; Juneja & Krishnasamy, 2019; Jedra & Proutiere, 2020). Further details are provided in Algorithm 1.

---

**Algorithm 1** Track-and-Stop Algorithm

**Input:** Covariate set $\mathcal{C}$, arm set $\mathcal{X}$, design point set $\mathcal{Z}$, confidence level $\delta$.

**Initialization:** Sample each design point $z_h \in \mathcal{Z}$ $n_0$ times.

Set $t \leftarrow n_0 D$ and update $N_h(t), \omega_h(t), \hat{\mathcal{P}}(t), \Lambda(\omega(t))$.

**while** $t\mathcal{H}(\omega(t), \hat{\mathcal{P}}(t))^{-1} < \rho(t, \delta)$ **do**
    **if** $\mathcal{B}_t \neq \emptyset$ **then**
        $z_{h(t+1)} = \arg\min_{z_h \in \mathcal{B}_t} N_h(t)$
    **else**
        $\omega^*(\hat{\mathcal{P}}(t)) \leftarrow \arg\max_{\omega \in \Omega} \mathcal{H}(\omega, \hat{\mathcal{P}}(t))^{-1}$
        $z_{h(t+1)} = \arg\min_{z_h \in \mathcal{Z}} N_h(t) - t\omega_h^*(\hat{\mathcal{P}}(t))$
    Sample the design point $z_{h(t+1)}$ and obtain the observation $Z_{t+1}$.
    Set $t \leftarrow t + 1$, and update $N_h(t), \omega_h(t), \hat{\mathcal{P}}(t), \Lambda(\omega(t))$.
**return** For each covariate $c_j \in \mathcal{C}$, recommend the estimated best arm:

$$x_{\hat{i}(c_j;\tau)} = \arg\max_{x_i \in \mathcal{X}} \hat{\theta}(\tau)^\top \phi(x_i, c_j)$$

$$\text{s.t.} \quad \hat{\beta}(\tau)^\top \phi(x_i, c_j) \leq b$$

---

When no arm is estimated to be feasible for some covariate, the algorithm samples uniformly and postpones the stopping rule until the estimated feasible set is nonempty for every covariate. This rule excludes pathological cases and does not affect the asymptotic analysis.

**Proposition 3.3.** *Under Assumptions 2.1-2.3, for any fixed $\alpha > 1$, there exists a constant $C > 0$ such that, with the stopping rule in (13) and*

$$\rho(t, \delta) = \log\left(\frac{Ct^\alpha \log(1/\delta)^{4D+1}}{\delta}\right), \qquad (15)$$

*Algorithm 1 is $\delta$-PAC and satisfies*

$$\limsup_{\delta \to 0} \frac{\mathbb{E}[\tau]}{\log(1/\delta)} \lesssim \mathcal{H}^*(\mathcal{P}).$$

Proposition 3.3 follows directly by extending the proof technique of (Jedra & Proutiere, 2020). It shows that the sample complexity upper bound of Algorithm 1 matches the lower bound in Theorem 3.1 up to a constant factor in the asymptotic regime as $\delta \to 0$.

### 3.3. A Duality Perspective

Although Algorithm 1 provides strong theoretical guarantees, it is impractical for implementation. The primary challenge arises from the fact that the lower bound involves a complex, multi-level optimization problem, which makes computing $\omega^*(\hat{\mathcal{P}}(t))$ at each iteration computationally prohibitive. Additionally, the presence of constraints and the linear structure complicate the analysis of the KKT conditions, unlike in the canonical BAI setting (Kaufmann et al., 2016), making it difficult to apply existing algorithms to our problem.

**Surrogate Objective Function.** We first introduce a surrogate objective function to reduce the computational burden. By merging the sets $\mathcal{D}_2(c_j)$ and $\mathcal{D}_3(c_j)$ for each covariate $c_j \in \mathcal{C}$ and focusing solely on the feasibility of the corresponding arms, we derive the following surrogate objective function $\Gamma^s(\omega, c_j, \mathcal{P})$ for $\Gamma(\omega, c_j, \mathcal{P})$ in (6):

$$\frac{1}{2}\min_{x_i \in \mathcal{X}}\left(\frac{((\phi(x_{i^*(c_j)}, c_j) - \phi(x_i, c_j))^\top \theta)^2}{\|\phi(x_{i^*(c_j)}, c_j) - \phi(x_i, c_j)\|^2_{\Lambda(\omega)^{-1}}}\mathbb{I}_1 \right.$$
$$\left. + \frac{(b - \beta^\top \phi(x_i, c_j))^2}{\|\phi(x_i, c_j)\|^2_{\Lambda(\omega)^{-1}}}\mathbb{I}_2\right), \qquad (16)$$

where $\mathbb{I}_1$ is the indicator function $\mathbb{I}(x_i \in \mathcal{D}_1(c_j))$, and $\mathbb{I}_2$ is the indicator function $\mathbb{I}(x_i \in \{x_{i^*(c_j)}\} \cup \mathcal{D}_2(c_j) \cup \mathcal{D}_3(c_j))$. By convention, any term multiplied by a zero indicator is defined as zero.

Compared to the original objective function $\Gamma(\omega, c_j, \mathcal{P})$, the surrogate function $\Gamma^s(\omega, c_j, \mathcal{P})$ exhibits a better decompo-

sition property, which can be leveraged to design a highly efficient algorithm.

**Lemma 3.4.** *Define*

$$\mathcal{U}^*(\mathcal{P})^{-1} = \max_{\omega \in \Omega} \min_{c_j \in \mathcal{C}} \Gamma^s(\omega, c_j, \mathcal{P}). \qquad (17)$$

*Then, it holds that*

$$\mathcal{H}^*(\mathcal{P}) \leq \mathcal{U}^*(\mathcal{P}).$$

Lemma 3.4 shows that the surrogate complexity $\mathcal{U}^*(\mathcal{P})$ provides an upper bound for the original complexity $\mathcal{H}^*(\mathcal{P})$ under the original objective function. This implies that $\mathcal{U}^*(\mathcal{P})$ can serve as a relaxed performance measure for the algorithms. In Appendix A.6, we establish a constant relaxation gap, i.e., $\mathcal{U}^*(\mathcal{P}) \leq C\mathcal{H}^*(\mathcal{P})$ for some problem instance-dependent constant $C > 1$.

**Dual Optimization Problem.** Although the primal multi-level optimization problem

$$\max_{\omega \in \Omega} \min_{c_j \in \mathcal{C}} \Gamma^s(\omega, c_j, \mathcal{P}) \qquad (18)$$

is complex, it admits a dual problem that can be efficiently solved using a decomposition algorithm.

**Theorem 3.5.** *The dual problem of (18) has the same optimizer as the following equivalent minimization problem*

$$\min_\lambda \mathcal{Q}(\lambda, \mathcal{P}) = -\sum_{h \in [D]}\sqrt{\sum_{i \in [K], j \in [M]} \lambda_{ij}\chi_h(x_i, c_j)}$$

$$s.t. \quad \sum_{i \in [K], j \in [M]} \lambda_{ij} = 1, \quad \lambda_{ij} \geq 0, \quad \forall i \in [K], j \in [M],$$
$$(19)$$

*where for each $c_j \in \mathcal{C}$, and if $x_i \in \mathcal{D}_1(c_j)$,*

$$\chi_h(x_i, c_j) = \frac{\sigma_h^2\left[(\Phi^\top)^{-1}(\phi(x_{i^*(c_j)}, c_j) - \phi(x_i, c_j))\right]_h^2}{\left((\phi(x_{i^*(c_j)}, c_j) - \phi(x_i, c_j))^\top \theta\right)^2}, \qquad (20)$$

*if $x_i \in \{x_{i^*(c_j)}\} \cup \mathcal{D}_2(c_j) \cup \mathcal{D}_3(c_j)$,*

$$\chi_h(x_i, c_j) = \frac{\sigma_h^2\left[(\Phi^\top)^{-1}\phi(x_i, c_j)\right]_h^2}{(b - \beta^\top \phi(x_i, c_j))^2}, \qquad (21)$$

*$\Phi$ is the $D \times D$ design matrix, and $[v]_h$ denotes the $h$th element of the vector $v$.*

The dual optimization problem in (19) is a convex optimization problem over the unit simplex, which can be efficiently solved using off-the-shelf gradient-based algorithms. The following Lemma 3.6 establishes that strong duality holds.

**Lemma 3.6.** *The primal optimization problem in (18) is convex, strong duality holds, and it admits a unique optimal solution.*

According to Lemma 3.6, given a dual optimal solution $\lambda^*$, an optimal static sampling ratio $\omega^*(\mathcal{P})$ can be recovered as follows:

$$\omega_h^*(\mathcal{P}) = \frac{\sqrt{\sum_{i\in[K],j\in[M]} \lambda_{ij}^* \chi_h(x_i,c_j)}}{\sum_{l\in[D]} \sqrt{\sum_{i\in[K],j\in[M]} \lambda_{ij}^* \chi_l(x_i,c_j)}}. \quad (22)$$

We provide an intuitive explanation of the optimal static sampling ratio $\omega^*(\mathcal{P})$. The optimal dual solution $\lambda^*$ represents the importance of each arm-covariate pair. The term $\chi_h(x_i,c_j)$ quantifies the benefit of sampling the design point $z_h$ for identifying a specific arm-covariate pair $(x_i,c_j)$. This quantity depends on the signal variance, the location in the feature space, and the optimality or feasibility gap. Consequently, the optimal sampling ratio must balance these factors, weighted by the relative importance of each arm-covariate pair, to minimize the overall sample complexity.

## 4. Duality-Based Decomposition Algorithm

In this section, we introduce a duality-based decomposition algorithm based on Theorem 3.5. Furthermore, we demonstrate that this algorithm asymptotically achieves the relaxed sample complexity bound $\mathcal{U}^*(\mathcal{P}) \log(1/\delta)$.

Leveraging the specific structure of problem (19), we design a decomposition algorithm that updates two coordinates at a time to reduce computational complexity.

**Lemma 4.1.** *Let $\lambda$ be a feasible dual solution at which $\mathcal{Q}(\lambda, \mathcal{P})$ is differentiable and such that $\lambda_{mn} > 0$ for some $m \in [K], n \in [M]$. Then, $\lambda$ is a stationary point of problem (19) if and only if*

$$\nabla \mathcal{Q}(\lambda, \mathcal{P})^\top d \geq 0, \forall d \in \mathcal{D}^{m,n}(\lambda), \quad (23)$$

*where $\mathcal{D}^{m,n}(\lambda) = \{e_{ij} - e_{mn} : i \neq m \, or \, j \neq n\} \cup \{e_{mn} - e_{ij} : i \neq m \, or \, j \neq n, \lambda_{ij} > 0\}$, $e_{ij} \in \mathbb{R}^{KM}$ is obtained by letting $\lambda_{ij}$ equal to one and other elements equal to zero.*

Note that (Lin et al., 2009) analyzes the decomposition structure of general singly linearly constrained problems with lower and upper bounds, and our dual problem (19) falls within this class. However, the problem is more challenging in our case because the problem instance $\mathcal{P}$ is unknown. Similar to Algorithm 1, we replace $\mathcal{P}$ with the estimated instance $\hat{\mathcal{P}}(t)$ to solve the empirical version of problem (19). Instead of performing full gradient descent to obtain the optimal static sampling ratio $\omega^*(\hat{\mathcal{P}}(t))$, we apply a single gradient step, alternating with the estimate update $\hat{\mathcal{P}}(t)$, which is sufficient to ensure asymptotic convergence while significantly reducing computational cost.

Algorithm 2 outlines the one-step gradient descent procedure. It begins by selecting a positive coordinate and then

searches over the corresponding reduced direction set to determine a descent direction and the maximal feasible step size. If the decrease in the objective function exceeds a given threshold, the algorithm employs the canonical line search to determine the step size and update the dual solution. A feasible sampling ratio can then be computed using (22).

We also compare the per-iteration complexity of Algorithm 1 and 2 (see Appendix A.11), showing that the proposed procedure is highly efficient.

---

**Algorithm 2** One-Step Gradient Descent Algorithm

**Input:** Covariate set $\mathcal{C}$, arm set $\mathcal{X}$, design point set $\mathcal{Z}$, a small positive constant $\kappa_0$ and $\eta < \frac{1}{KM}$, $\hat{\mathcal{P}}(t), \hat{\theta}(t), \hat{\beta}(t), \lambda(t-1)$.

**Initialization:** Let

$$x_{\hat{i}(c_j;t)} = \arg\max_{x_i \in \mathcal{X}} \hat{\theta}(t)^\top \phi(x_i, c_j)$$

$$\text{s.t.} \quad \hat{\beta}(t)^\top \phi(x_i, c_j) \leq b$$

for each covariate $c_j \in \mathcal{C}$.
Randomly choose $(m(t), n(t))$ from

$$\{(i,j) : \lambda_{ij}(t-1) \geq \eta\}.$$

Compute the descent direction $d(t)$ and the maximum feasible step size $s^{\max}$:

$$d(t) \in \arg\min_d s^{\max}(d) \nabla \mathcal{Q}(\lambda(t-1), \hat{\mathcal{P}}(t))^\top d,$$

$$\text{s.t.} \quad d \in \mathcal{D}^{(m(t),n(t))}(\lambda(t-1)),$$

$$s^{\max}(d) = \max\{s \geq 0 : \lambda(t-1) + sd \in [0,1]^{KM}\},$$

$$s^{\max} = s^{\max}(d(t)).$$

Define $\mathcal{W}(t) = \nabla \mathcal{Q}(\lambda(t-1), \hat{\mathcal{P}}(t))^\top d(t)$.

**if**

$$\mathcal{W}(t) < \max\{-\kappa_0, -(\log t/t)^{1/4}\}$$

*and*

$$s^{max}\mathcal{W}(t) < \max\{-\kappa_0, -(\log t/t)^{1/2}\}$$

**then**

$$\lambda(t) = \lambda(t-1) + s(t)d(t)$$

where $s(t) = $ LineSearch Algorithm $(d(t), s^{max})$

**else**

$$\lambda(t) = \lambda(t-1)$$

**Return:** Sampling ratio $\gamma(\hat{\mathcal{P}}(t))$ calculated according to (22) based on $\lambda(t)$.

---

The one-step gradient descent idea has appeared in the sim-

ulation literature (Zhou et al., 2024; Du et al., 2024), but our approach differs in two key ways. First, we tackle a more complex constrained BAI problem with covariate selection, which has not been previously explored. Second, we analyze the algorithm in the fixed-confidence setting to assess its statistical validity and sample complexity, whereas existing work focuses on sampling ratio convergence under the fixed-budget setting.

The algorithmic framework is the same as Algorithm 1, except for a modified sampling rule:

$$
z_{h(t+1)} = \begin{cases} \arg\min_{z_h \in \mathcal{B}_t} N_h(t) & \text{if} \quad \mathcal{B}_t \neq \emptyset \\ \arg\min_{z_h \in \mathcal{Z}} N_h(t) - t\gamma_h(\hat{\mathcal{P}}(t)) & \text{otherwise} \end{cases},
$$
(24)

where
$$
\gamma(\hat{\mathcal{P}}(t)) = \{\gamma_h(\hat{\mathcal{P}}(t))\}_{h \in [D]}
$$
(25)

denotes the sampling ratio returned by Algorithm 2. To mitigate the effect of estimation error, we reset $\lambda(t)$ to the uniform vector $1/(KM)$ whenever the estimated optimal arm for at least one covariate changes. If the estimated feasible set is empty for some covariate, we skip the dual update, reset the input dual iterate to the uniform vector, and return the uniform sampling ratio over $\mathcal{Z}$. This default rule does not affect the asymptotic analysis, since under Assumption 2.1 and the consistency of $\hat{\mathcal{P}}(t)$, the estimated feasible set is nonempty, and the estimated optimal arms stabilize almost surely for all sufficiently large $t$. We refer to this algorithm as the duality-based decomposition algorithm.

Theorem 4.2 shows that the algorithm asymptotically matches the relaxed bound $\mathcal{U}^*(\mathcal{P}) \log(1/\delta)$ on sample complexity up to a constant factor.

**Theorem 4.2.** *Under Assumptions 2.1-2.3, the duality-based decomposition algorithm is $\delta$-PAC and satisfies*

$$
\mathbb{P}\left( \limsup_{\delta \to 0} \frac{\tau}{\log(1/\delta)} \leq \mathcal{U}^*(\mathcal{P}) \right) = 1,
$$
(26)

*and*

$$
\limsup_{\delta \to 0} \frac{\mathbb{E}[\tau]}{\log(1/\delta)} \lesssim \mathcal{U}^*(\mathcal{P}).
$$
(27)

## 5. Numerical Experiment

In this section, we evaluate the practical performance of the proposed duality-based decomposition algorithm. Detailed parameter settings and pseudo-code are provided in Appendix A.12.

We consider a problem with two covariates, four arms, and one constraint. For the first covariate, there is one optimal arm and three suboptimal arms. For the second, there is one optimal, one suboptimal, and two infeasible arms, i.e., one with better performance and one with worse performance than the optimal arm.

Since no existing methods directly address our problem, we propose the following benchmarks for comparison:

- **USR**: Allocate an equal number of samples to each design point.

- **BCSR**: A modified Best Challenger algorithm (Garivier & Kaufmann, 2016) based solely on arm optimality, representing the state-of-the-art for BAI.

- **GOSR**: A greedy algorithm for problem (18) that relies solely on arm optimality.

- **GFSR**: A greedy algorithm for problem (18) that relies solely on arm feasibility.

We refer to our proposed duality-based decomposition algorithm as **DSR**.

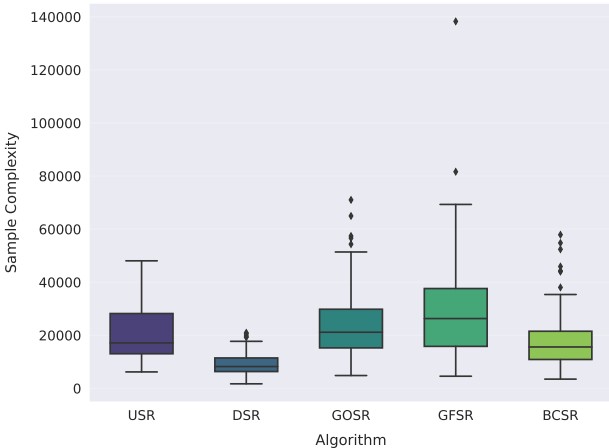

*Figure 1.* Empirical sample complexity over 100 runs

Figures 1 and 2 illustrate the empirical sample complexity and probability of correct identification (PCI) based on 100 independent macro-replications of various algorithms, with $\delta = 0.1$ and $n_0 = 1$. The results demonstrate that DSR achieves the lowest sample complexity among all benchmarks, with an average of 9205.46 samples. Furthermore, the findings highlight the statistical conservatism of the fixed-confidence setting: with 4000 samples, the empirical PCI of both DSR and GOSR exceeds the target PCI. Notably, the DSR algorithm outperforms all other benchmarks in terms of the PCI measure.

We further evaluate the robustness of the proposed algorithm when the Gaussian noise assumption is violated. The problem setting remains the same, while the observation noise is generated from a standard $t$-distribution with relatively small variance. Figure 3 reports the empirical sample complexity of different algorithms based on 30 macro-replications.

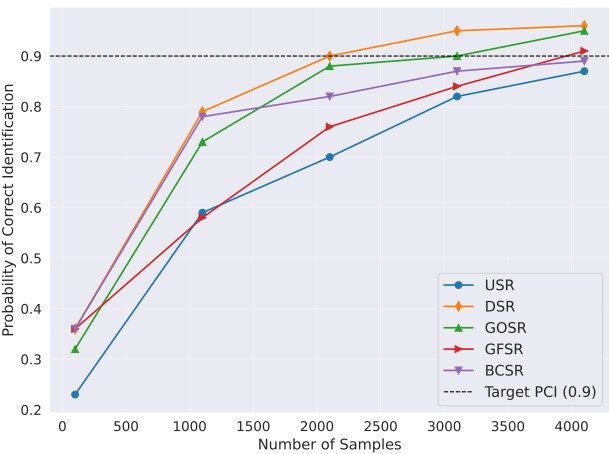

*Figure 2.* Empirical PCI over 100 runs

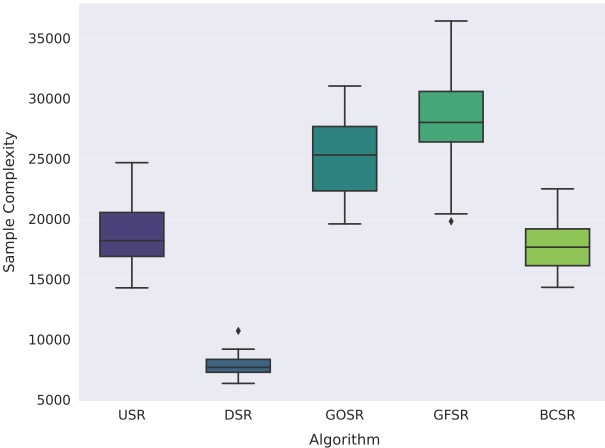

*Figure 3.* Empirical sample complexity over $t$-distribution noise

The results show that DSR remains highly effective under heavy-tailed noise. Compared with the benchmark policies, DSR achieves the lowest sample complexity and exhibits a relatively stable performance across replications. This suggests that the proposed duality-based decomposition algorithm is not overly sensitive to deviations from the Gaussian noise assumption and maintains strong empirical robustness under non-Gaussian, heavy-tailed observations.

Additional experiments under different problem instances are reported in Appendix A.12. The results are broadly consistent with the findings above, showing that DSR maintains its advantage over the benchmark algorithms. We also provide an application example on personalized treatment for diabetes management in Appendix A.13, illustrating the potential practical applicability of the proposed method.

## 6. Conclusion

This paper studies a constrained linear BAI problem with co-variate selection, where each arm has multiple performance metrics, and the goal is to identify the best feasible arm per covariate. Our main contributions include an instance-dependent lower bound, a relaxed bound derived from a surrogate optimization problem, a duality-based formulation, and an efficient decomposition algorithm with theoretical guarantees. This work opens several avenues for future research, including extending the framework to continuous covariate spaces and generalizing the linear model to more flexible statistical structures, such as Gaussian Process Regression.

## Acknowledgements

Mingjie Hu and Jianqiang Hu are supported by the National Natural Science Foundation of China (NSFC) under grants 72033003, 72350710219, 72342006, and 72293565. Enlu Zhou is grateful for the support of the Air Force Office of Scientific Research (AFOSR) under Grant FA9550-25-1-0310 and the National Science Foundation under Award ECCS-2419562.

## Impact Statement

This paper presents work whose goal is to advance the field of Machine Learning. There are many potential societal consequences of our work, none of which we feel must be specifically highlighted here.

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

## A. Technical Appendices and Supplementary Material

### A.1. Large Language Models usage

ChatGPT was used for wording refinement and expression improvement.

### A.2. Proof of Theorem 3.1

*Proof.* To prove Theorem 3.1, we first introduce additional notation that was simplified or omitted in the main paper for clarity. Let $x_{i^*(c_j,\mathcal{P})}$ denote the best arm for covariate $c_j$ under the problem instance $\mathcal{P}$; when no ambiguity arises, we abbreviate this as $x_{i^*(c_j)}$. We define $d(f(z_h), \tilde{f}(z_h))$ as the KL divergence between two Gaussian random variables with means $f(z_h)$ and $\tilde{f}(z_h)$, sharing a common variance $\sigma_h^2$. The subscript $h$ indexes design points; for instance, if $z_h$ corresponds to the arm-covariate pair $(x_i, c_j)$, then $f(z_h) = f(x_i, c_j), \sigma_h^2 = \sigma_{ij}^2$.

A problem instance can be represented as $\mathcal{P} = (f(x_i, c_j), g(x_i, c_j))_{x_i \in \mathcal{X}, c_j \in \mathcal{C}}$. Consider the set of alternative instances

$$\mathcal{A}(\mathcal{P}) = \left\{ \tilde{\mathcal{P}} \in \mathcal{S} : \exists c_j \in \mathcal{C}, x_{i^*(c_j,\mathcal{P})} \neq x_{i^*(c_j,\tilde{\mathcal{P}})} \right\}, \tag{28}$$

which includes all problem instances $\tilde{\mathcal{P}} = (\tilde{f}(x_i, c_j), \tilde{g}(x_i, c_j))_{x_i \in \mathcal{X}, c_j \in \mathcal{C}}$ for which the optimal arm differs from that of $\mathcal{P}$ for at least one covariate. Throughout the proof, $\mathcal{S}$ is understood as the class of regular instances in which each covariate has a unique optimal feasible arm, and infeasibility alternatives are taken over the closure of valid instances that retain at least one feasible arm; hence, degenerate perturbations with no feasible arm do not affect the infimum calculations.

In the fixed confidence setting, for a given confidence level $\delta \in (0, 1)$, the $\delta$-PAC condition requires that

$$\mathbb{P}_{\mathcal{P}}\left( \forall c_j \in \mathcal{C}, x_{\hat{i}(c_j,\tau)} = x_{i^*(c_j,\mathcal{P})} \right) \geq 1 - \delta, \tag{29}$$

and for any alternative instance $\tilde{\mathcal{P}} \in \mathcal{A}(\mathcal{P})$,

$$\mathbb{P}_{\tilde{\mathcal{P}}}\left( \forall c_j \in \mathcal{C}, x_{\hat{i}(c_j,\tau)} = x_{i^*(c_j,\mathcal{P})} \right) \leq \delta. \tag{30}$$

As the event

$$\left\{ \forall c_j \in \mathcal{C}, x_{\hat{i}(c_j,\tau)} = x_{i^*(c_j,\mathcal{P})} \right\} \tag{31}$$

belongs to the filtration generated by all observations collected up to the stopping time $\tau$. Thus, applying the transportation inequality (Lemma 1) from Kaufmann et al. (2016), we obtain a fundamental information-theoretic lower bound:

$$\forall \tilde{\mathcal{P}} \in \mathcal{A}(\mathcal{P}), \sum_{h \in [D]} \mathbb{E}[N_h]\left( d(f(z_h), \tilde{f}(z_h)) + d(g(z_h), \tilde{g}(z_h)) \right) \geq kl(\delta, 1 - \delta). \tag{32}$$

Consequently, we have the following sequence of inequalities:

$$\begin{aligned}
kl(\delta, 1-\delta) &\leq \inf_{\tilde{\mathcal{P}} \in \mathcal{A}(\mathcal{P})} \sum_{h \in [D]} \mathbb{E}[N_h]\left( d(f(z_h), \tilde{f}(z_h)) + d(g(z_h), \tilde{g}(z_h)) \right) \\
&= \mathbb{E}[\tau] \inf_{\tilde{\mathcal{P}} \in \mathcal{A}(\mathcal{P})} \sum_{h \in [D]} \frac{\mathbb{E}[N_h]}{\mathbb{E}[\tau]}\left( d(f(z_h), \tilde{f}(z_h)) + d(g(z_h), \tilde{g}(z_h)) \right) \\
&= \mathbb{E}[\tau] \inf_{\tilde{\mathcal{P}} \in \mathcal{A}(\mathcal{P})} \sum_{h \in [D]} \omega_h \left( d(f(z_h), \tilde{f}(z_h)) + d(g(z_h), \tilde{g}(z_h)) \right) \\
&\leq \mathbb{E}[\tau] \sup_{\omega \in \Omega} \inf_{\tilde{\mathcal{P}} \in \mathcal{A}(\mathcal{P})} \sum_{h \in [D]} \omega_h \left( d(f(z_h), \tilde{f}(z_h)) + d(g(z_h), \tilde{g}(z_h)) \right),
\end{aligned} \tag{33}$$

where $\omega_h = \mathbb{E}[N_h]/\mathbb{E}[\tau]$ denotes the expected sampling proportion at design point $z_h$. Since $\sum_{h\in[D]} N_h = \tau$, we have $\omega \in \Omega$.

This leads to the following lower bound on the sample complexity:

$$\mathbb{E}[\tau] \geq \mathcal{H}^*(\mathcal{P})kl(\delta, 1-\delta), \tag{34}$$

where the instance-dependent complexity term is defined as

$$
\begin{aligned}
\mathcal{H}^*(\mathcal{P})^{-1} &= \sup_{\omega\in\Omega} \inf_{\tilde{\mathcal{P}}\in\mathcal{A}(\mathcal{P})} \mathcal{H}(\omega, \mathcal{P}, \tilde{\mathcal{P}})^{-1} \\
&= \sup_{\omega\in\Omega} \inf_{\tilde{\mathcal{P}}\in\mathcal{A}(\mathcal{P})} \sum_{h\in[D]} \omega_h \left( d(f(z_h), \tilde{f}(z_h)) + d(g(z_h), \tilde{g}(z_h)) \right).
\end{aligned}
\tag{35}
$$

For each covariate $c_j \in \mathcal{C}$, define the following sets:

$$\mathcal{O}(x_{i^*(c_j, \mathcal{P})}, c_j) = \left\{ \tilde{\mathcal{P}} \in \mathcal{S} : \tilde{\beta}^\top \phi(x_{i^*(c_j, \mathcal{P})}, c_j) > b \right\}, \tag{36}$$

and

$$\mathcal{O}(x_i, c_j) = \left\{ \tilde{\mathcal{P}} \in \mathcal{S} : \tilde{\theta}^\top (\phi(x_i, c_j) - \phi(x_{i^*(c_j, \mathcal{P})}, c_j)) > 0, \tilde{\beta}^\top \phi(x_i, c_j) \leq b \right\}. \tag{37}$$

Then, the set $\mathcal{A}(\mathcal{P})$ can be decomposed as

$$
\begin{aligned}
\mathcal{A}(\mathcal{P}) &= \left\{ \tilde{\mathcal{P}} \in \mathcal{S} : \exists c_j \in \mathcal{C}, x_{i^*(c_j, \mathcal{P})} \neq x_{i^*(c_j, \tilde{\mathcal{P}})} \right\} \\
&= \bigcup_{c_j\in\mathcal{C}} \left\{ \tilde{\mathcal{P}} \in \mathcal{S} : x_{i^*(c_j, \mathcal{P})} \neq x_{i^*(c_j, \tilde{\mathcal{P}})} \right\} \\
&= \bigcup_{c_j\in\mathcal{C}} \left( \left\{ \tilde{\mathcal{P}} \in \mathcal{S} : \tilde{\beta}^\top \phi(x_{i^*(c_j, \mathcal{P})}, c_j) > b \right\} \right. \\
&\qquad \left. \bigcup \left\{ \tilde{\mathcal{P}} \in \mathcal{S} : \exists x_i \in \mathcal{X}, \tilde{\theta}^\top (\phi(x_i, c_j) - \phi(x_{i^*(c_j, \mathcal{P})}, c_j)) > 0, \tilde{\beta}^\top \phi(x_i, c_j) \leq b \right\} \right) \\
&= \bigcup_{c_j\in\mathcal{C}} \left( \mathcal{O}(x_{i^*(c_j, \mathcal{P})}, c_j) \bigcup \left( \bigcup_{x_i\in\mathcal{X}\setminus\{x_{i^*(c_j, \mathcal{P})}\}} \mathcal{O}(x_i, c_j) \right) \right)
\end{aligned}
\tag{38}
$$

Then, we can express $\mathcal{H}^*(\mathcal{P})^{-1}$ as:

$$
\begin{aligned}
\mathcal{H}^*(\mathcal{P})^{-1} &= \sup_{\omega\in\Omega} \inf_{\tilde{\mathcal{P}}\in\mathcal{A}(\mathcal{P})} \mathcal{H}(\omega, \mathcal{P}, \tilde{\mathcal{P}})^{-1} \\
&= \sup_{\omega\in\Omega} \min_{c_j\in\mathcal{C}} \min \left( \inf_{\tilde{\mathcal{P}}\in\mathcal{O}(x_{i^*(c_j, \mathcal{P})}, c_j)} \mathcal{H}(\omega, \mathcal{P}, \tilde{\mathcal{P}})^{-1}, \min_{x_i\in\mathcal{X}\setminus\{x_{i^*(c_j, \mathcal{P})}\}} \inf_{\tilde{\mathcal{P}}\in\mathcal{O}(x_i, c_j)} \mathcal{H}(\omega, \mathcal{P}, \tilde{\mathcal{P}})^{-1} \right).
\end{aligned}
\tag{39}
$$

Next, we leverage the linear model structure and Gaussian noise assumptions from Assumptions 2.2 and 2.3 to derive a closed-form expression for $\mathcal{H}^*(\mathcal{P})$. Recall that for two univariate Gaussian distributions with equal variance, the KL divergence is given by

$$d(f(z_h), \tilde{f}(z_h)) = \frac{(f(z_h) - \tilde{f}(z_h))^2}{2\sigma_h^2} = \frac{(\theta - \tilde{\theta})^\top \phi(z_h)\phi(z_h)^\top (\theta - \tilde{\theta})}{2\sigma_h^2}. \tag{40}$$

Using this result, the function $\mathcal{H}(\omega, \mathcal{P}, \tilde{\mathcal{P}})^{-1}$ admits the following closed-form:

$$\mathcal{H}(\omega, \mathcal{P}, \tilde{\mathcal{P}})^{-1} = \sum_{h\in[D]} \omega_h \left( \frac{(\theta - \tilde{\theta})^\top \phi(z_h)\phi(z_h)^\top (\theta - \tilde{\theta})}{2\sigma_h^2} + \frac{(\beta - \tilde{\beta})^\top \phi(z_h)\phi(z_h)^\top (\beta - \tilde{\beta})}{2\sigma_h^2} \right). \tag{41}$$

Throughout the following infimum computations, replacing the strict inequality constraints in these alternative sets by their closures does not change the infimum; hence, we solve the corresponding boundary problems. We now consider the following sub-optimization problem:

$$
\inf_{\tilde{\mathcal{P}} \in \mathcal{O}(x_{i^*(c_j, \mathcal{P})}, c_j)} \mathcal{H}(\omega, \mathcal{P}, \tilde{\mathcal{P}})^{-1}
$$

$$
= \inf_{\tilde{\beta}^\top \phi(x_{i^*(c_j, \mathcal{P})}, c_j) > b} \sum_{h \in [D]} \omega_h \left( \frac{(\theta - \tilde{\theta})^\top \phi(z_h)\phi(z_h)^\top (\theta - \tilde{\theta})}{2\sigma_h^2} + \frac{(\beta - \tilde{\beta})^\top \phi(z_h)\phi(z_h)^\top (\beta - \tilde{\beta})}{2\sigma_h^2} \right)
$$

$$
= \inf_{\tilde{\beta}^\top \phi(x_{i^*(c_j, \mathcal{P})}, c_j) > b} \sum_{h \in [D]} \omega_h \frac{(\beta - \tilde{\beta})^\top \phi(z_h)\phi(z_h)^\top (\beta - \tilde{\beta})}{2\sigma_h^2} \tag{42}
$$

$$
= \frac{1}{2} \inf_{\tilde{\beta}^\top \phi(x_{i^*(c_j, \mathcal{P})}, c_j) > b} (\beta - \tilde{\beta})^\top \left( \sum_{h \in [D]} \omega_h \frac{\phi(z_h)\phi(z_h)^\top}{\sigma_h^2} \right) (\beta - \tilde{\beta})
$$

$$
= \frac{1}{2} \inf_{\tilde{\beta}^\top \phi(x_{i^*(c_j, \mathcal{P})}, c_j) > b} (\beta - \tilde{\beta})^\top \Lambda(\omega)(\beta - \tilde{\beta}),
$$

where we define

$$
\Lambda(\omega) = \sum_{h \in [D]} \omega_h \frac{\phi(z_h)\phi(z_h)^\top}{\sigma_h^2}. \tag{43}
$$

Thus, the subproblem reduces to the following constrained quadratic minimization:

$$
\inf_{\tilde{\beta}} \quad (\beta - \tilde{\beta})^\top \Lambda(\omega)(\beta - \tilde{\beta})
$$

$$
\text{s.t.} \quad \tilde{\beta}^\top \phi(x_{i^*(c_j, \mathcal{P})}, c_j) > b \quad (\lambda) \tag{44}
$$

For the KKT derivations below, we first consider the case where $\Lambda(\omega)$ is nonsingular. When $\Lambda(\omega)$ is singular, the same optimal values are understood under the extended inverse convention, where uncovered directions are assigned infinite variance. The Karush–Kuhn–Tucker (KKT) conditions for the above optimization problem are given by

$$
2\Lambda(\omega)(\beta - \tilde{\beta}) + \lambda\phi(x_{i^*(c_j, \mathcal{P})}, c_j) = 0
$$

$$
\tilde{\beta}^\top \phi(x_{i^*(c_j, \mathcal{P})}, c_j) = b, \tag{45}
$$

where $\lambda$ is the Lagrange multiplier associated with the inequality constraint. According to the first equation in (45), it holds that

$$
\tilde{\beta} = \beta + \frac{1}{2}\lambda\Lambda(\omega)^{-1}\phi(x_{i^*(c_j, \mathcal{P})}, c_j). \tag{46}
$$

Plug (54) into the second equation in (45), we have that

$$
\lambda^* = \frac{2(b - \beta^\top \phi(x_{i^*(c_j, \mathcal{P})}, c_j))}{\|\phi(x_{i^*(c_j, \mathcal{P})}, c_j)\|_{\Lambda(\omega)^{-1}}^2}. \tag{47}
$$

Plug (55) into (54) yields the optimal solution

$$
\tilde{\beta}^* = \beta + \frac{b - \beta^\top \phi(x_{i^*(c_j, \mathcal{P})}, c_j)}{\|\phi(x_{i^*(c_j, \mathcal{P})}, c_j)\|_{\Lambda(\omega)^{-1}}^2} \Lambda(\omega)^{-1}\phi(x_{i^*(c_j, \mathcal{P})}, c_j). \tag{48}
$$

The corresponding optimal value of the objective function is

$$
\frac{(b - \beta^\top \phi(x_{i^*(c_j, \mathcal{P})}, c_j))^2}{\|\phi(x_{i^*(c_j, \mathcal{P})}, c_j)\|_{\Lambda(\omega)^{-1}}^2}. \tag{49}
$$

Next, we consider the complementary sub-optimization problem

$$
\min_{x_i \in \mathcal{X} \setminus \{x_{i^*(c_j, \mathcal{P})}\}} \inf_{\tilde{\mathcal{P}} \in \mathcal{O}(x_i, c_j)} \mathcal{H}(\omega, \mathcal{P}, \tilde{\mathcal{P}})^{-1}
$$
$$
= \min \left( \min_{x_i \in \mathcal{D}_1(c_j)} \inf_{\tilde{\mathcal{P}} \in \mathcal{O}(x_i, c_j)} \mathcal{H}(\omega, \mathcal{P}, \tilde{\mathcal{P}})^{-1}, \min_{x_i \in \mathcal{D}_2(c_j)} \inf_{\tilde{\mathcal{P}} \in \mathcal{O}(x_i, c_j)} \mathcal{H}(\omega, \mathcal{P}, \tilde{\mathcal{P}})^{-1}, \right. \tag{50}
$$
$$
\left. \min_{x_i \in \mathcal{D}_3(c_j)} \inf_{\tilde{\mathcal{P}} \in \mathcal{O}(x_i, c_j)} \mathcal{H}(\omega, \mathcal{P}, \tilde{\mathcal{P}})^{-1} \right).
$$

Consider the analysis of the following optimization problem as an example:

$$
\min_{x_i \in \mathcal{D}_1(c_j)} \inf_{\tilde{\mathcal{P}} \in \mathcal{O}(x_i, c_j)} \mathcal{H}(\omega, \mathcal{P}, \tilde{\mathcal{P}})^{-1}
$$
$$
= \min_{x_i \in \mathcal{D}_1(c_j)} \inf_{\tilde{\mathcal{P}} \in \mathcal{O}(x_i, c_j)} \sum_{h \in [D]} \omega_h \left( \frac{(\theta - \tilde{\theta})^\top \phi(z_h) \phi(z_h)^\top (\theta - \tilde{\theta})}{2\sigma_h^2} + \frac{(\beta - \tilde{\beta})^\top \phi(z_h) \phi(z_h)^\top (\beta - \tilde{\beta})}{2\sigma_h^2} \right)
$$
$$
= \min_{x_i \in \mathcal{D}_1(c_j)} \inf_{\tilde{\mathcal{P}} \in \mathcal{O}(x_i, c_j)} \sum_{h \in [D]} \omega_h \left( \frac{(\theta - \tilde{\theta})^\top \phi(z_h) \phi(z_h)^\top (\theta - \tilde{\theta})}{2\sigma_h^2} \right) \tag{51}
$$
$$
= \frac{1}{2} \min_{x_i \in \mathcal{D}_1(c_j)} \inf_{\tilde{\mathcal{P}} \in \mathcal{O}(x_i, c_j)} (\theta - \tilde{\theta})^\top \Lambda(\omega)(\theta - \tilde{\theta})
$$

The inner optimization problem is therefore

$$
\inf_{\tilde{\theta}} \quad (\theta - \tilde{\theta})^\top \Lambda(\omega)(\theta - \tilde{\theta})
$$
$$
\text{s.t.} \quad \tilde{\theta}^\top (\phi(x_i, c_j) - \phi(x_{i^*(c_j, \mathcal{P})}, c_j)) \geq 0 \quad (\lambda) \tag{52}
$$

The KKT conditions are given by

$$
2\Lambda(\omega)(\theta - \tilde{\theta}) + \lambda(\phi(x_i, c_j) - \phi(x_{i^*(c_j, \mathcal{P})}, c_j)) = 0
$$
$$
\tilde{\theta}^\top (\phi(x_i, c_j) - \phi(x_{i^*(c_j, \mathcal{P})}, c_j)) = 0 \tag{53}
$$

According to the first equation in (53), it holds that

$$
\tilde{\theta} = \theta + \frac{1}{2} \lambda \Lambda(\omega)^{-1} (\phi(x_i, c_j) - \phi(x_{i^*(c_j, \mathcal{P})}, c_j)). \tag{54}
$$

Plug (54) into the second equation in (53), we have that

$$
\lambda^* = \frac{2(\theta^\top (\phi(x_{i^*(c_j, \mathcal{P})}, c_j) - \phi(x_i, c_j)))}{\|\phi(x_i, c_j) - \phi(x_{i^*(c_j, \mathcal{P})}, c_j)\|_{\Lambda(\omega)^{-1}}^2}. \tag{55}
$$

Plug (55) into (54) yields the optimal solution.

$$
\tilde{\theta}^* = \theta + \frac{\theta^\top (\phi(x_{i^*(c_j, \mathcal{P})}, c_j) - \phi(x_i, c_j))}{\|\phi(x_{i^*(c_j, \mathcal{P})}, c_j) - \phi(x_i, c_j)\|_{\Lambda(\omega)^{-1}}^2} \Lambda(\omega)^{-1} (\phi(x_i, c_j) - \phi(x_{i^*(c_j, \mathcal{P})}, c_j)), \tag{56}
$$

The corresponding optimal value is

$$
\frac{(\theta^\top (\phi(x_{i^*(c_j, \mathcal{P})}, c_j) - \phi(x_i, c_j)))^2}{\|\phi(x_{i^*(c_j, \mathcal{P})}, c_j) - \phi(x_i, c_j)\|_{\Lambda(\omega)^{-1}}^2}. \tag{57}
$$

The analyses for the subproblems

$$
\min_{x_i \in \mathcal{D}_2(c_j)} \inf_{\tilde{\mathcal{P}} \in \mathcal{O}(x_i, c_j)} \mathcal{H}(\omega, \mathcal{P}, \tilde{\mathcal{P}})^{-1} \tag{58}
$$

and

$$\min_{x_i \in \mathcal{D}_3(c_j)} \inf_{\tilde{\mathcal{P}} \in \mathcal{O}(x_i, c_j)} \mathcal{H}(\omega, \mathcal{P}, \tilde{\mathcal{P}})^{-1} \tag{59}$$

follow analogous steps. For arms $x_i \in \mathcal{D}_2(c_j)$, if $f(x_i, c_j) = f(x_{i^*(c_j, \mathcal{P})}, c_j)$, then making $x_i$ feasible only creates a tie in the objective value. However, an arbitrarily small perturbation of $\theta$ can make $x_i$ strictly better than $x_{i^*(c_j, \mathcal{P})}$, and this perturbation does not affect the value of the infimum. Hence, the infimum for arms in $\mathcal{D}_2(c_j)$ is determined by the feasibility constraint alone.

Their optimal values are respectively

$$\frac{(b - \beta^\top \phi(x_i, c_j))^2}{\|\phi(x_i, c_j)\|_{\Lambda(\omega)^{-1}}^2} \tag{60}$$

and

$$\frac{(\theta^\top (\phi(x_{i^*(c_j, \mathcal{P})}, c_j) - \phi(x_i, c_j)))^2}{\|\phi(x_{i^*(c_j, \mathcal{P})}, c_j) - \phi(x_i, c_j)\|_{\Lambda(\omega)^{-1}}^2} + \frac{(b - \beta^\top \phi(x_i, c_j))^2}{\|\phi(x_i, c_j)\|_{\Lambda(\omega)^{-1}}^2}. \tag{61}$$

Finally, we conclude that $\mathcal{H}^*(\mathcal{P})^{-1} = \max_{\omega \in \Omega} \min_{c_j \in \mathcal{C}} \Gamma(\omega, c_j, \mathcal{P})$, where

$$\Gamma(\omega, c_j, \mathcal{P}) = \frac{1}{2} \min \left( \min_{x_i \neq x_{i^*(c_j, \mathcal{P})}} \left( \frac{((\phi(x_{i^*(c_j, \mathcal{P})}, c_j) - \phi(x_i, c_j))^\top \theta)^2}{\|\phi(x_{i^*(c_j, \mathcal{P})}, c_j) - \phi(x_i, c_j)\|_{\Lambda(\omega)^{-1}}^2} \mathbb{I}(x_i \in \mathcal{D}_1(c_j) \cup \mathcal{D}_3(c_j)) \right. \right.$$
$$\left. \left. + \frac{(b - \beta^\top \phi(x_i, c_j))^2}{\|\phi(x_i, c_j)\|_{\Lambda(\omega)^{-1}}^2} \mathbb{I}(x_i \in \mathcal{D}_2(c_j) \cup \mathcal{D}_3(c_j)) \right), \frac{(b - \beta^\top \phi(x_{i^*(c_j, \mathcal{P})}, c_j))^2}{\|\phi(x_{i^*(c_j, \mathcal{P})}, c_j)\|_{\Lambda(\omega)^{-1}}^2} \right). \tag{62}$$

$\square$

### A.3. Multiple Constraints Setting

In this section, we present the sample complexity lower bound for the multiple-constraint setting, which follows directly from an extension of the proof of Theorem 3.1. In the multi-constraint setting, each arm corresponds to a random performance vector $(F(x_i, c_j), G_1(x_i, c_j), \ldots, G_H(x_i, c_j))$, and the sample complexity must separately account for both feasible and infeasible constraints of each arm. For the $s$-th constraint of the arm-covariate pair $(x_i, c_j)$, the mean performance is given by $g_s(x_i, c_j) = \beta_s^\top \phi(x_i, c_j)$.

For each arm-covariate pair $(x_i, c_j)$, define the feasible and infeasible constraint index sets as

$$\mathcal{F}(x_i, c_j) \triangleq \{s \in [H] : g_s(x_i, c_j) \leq b\}, \qquad \mathcal{I}(x_i, c_j) \triangleq \{s \in [H] : g_s(x_i, c_j) > b\}.$$

Accordingly, for each covariate $c_j \in \mathcal{C}$, the arm categories are extended as

$$\mathcal{D}_1(c_j) \triangleq \{x_i : f(x_i, c_j) < f(x_{i^*(c_j)}, c_j), \mathcal{I}(x_i, c_j) = \emptyset\},$$

$$\mathcal{D}_2(c_j) \triangleq \{x_i : f(x_i, c_j) \geq f(x_{i^*(c_j)}, c_j), \mathcal{I}(x_i, c_j) \neq \emptyset\},$$

and

$$\mathcal{D}_3(c_j) \triangleq \{x_i : f(x_i, c_j) < f(x_{i^*(c_j)}, c_j), \mathcal{I}(x_i, c_j) \neq \emptyset\}.$$

**Theorem A.1.** *Under Assumptions 2.1-2.3, for a fixed confidence level $\delta \in (0, 1/2)$, any $\delta$-PAC algorithm applied to problem instance $\mathcal{P} \in \mathcal{S}$ must satisfy*

$$\mathbb{E}[\tau] \geq \mathcal{H}^*(\mathcal{P}) kl(\delta, 1 - \delta), \tag{63}$$

*which leads to*

$$\liminf_{\delta \to 0} \frac{\mathbb{E}[\tau]}{\log(1/\delta)} \geq \mathcal{H}^*(\mathcal{P}), \tag{64}$$

*where* $\mathcal{H}^*(\mathcal{P})^{-1} = \max_{\omega \in \Omega} \min_{c_j \in \mathcal{C}} \Gamma(\omega, c_j, \mathcal{P}),$

$$
\begin{aligned}
\Gamma(\omega, c_j, \mathcal{P}) = \frac{1}{2} \min \Bigg( & \min_{x_i \neq x_{i^*(c_j)}} \left( \frac{((\phi(x_{i^*(c_j)}, c_j) - \phi(x_i, c_j))^\top \theta)^2}{\|\phi(x_{i^*(c_j)}, c_j) - \phi(x_i, c_j)\|^2_{\Lambda(\omega)^{-1}}} \mathbb{I}\big(x_i \in \mathcal{D}_1(c_j) \cup \mathcal{D}_3(c_j)\big) \right. \\
& + \sum_{s \in \mathcal{I}(x_i, c_j)} \frac{(b - \beta_s^\top \phi(x_i, c_j))^2}{\|\phi(x_i, c_j)\|^2_{\Lambda(\omega)^{-1}}} \mathbb{I}\big(x_i \in \mathcal{D}_2(c_j) \cup \mathcal{D}_3(c_j)\big) \Bigg), \min_{s \in \mathcal{F}(x_{i^*(c_j)}, c_j)} \frac{(b - \beta_s^\top \phi(x_{i^*(c_j)}, c_j))^2}{\|\phi(x_{i^*(c_j)}, c_j)\|^2_{\Lambda(\omega)^{-1}}} \Bigg),
\end{aligned}
\tag{65}
$$

$\Lambda(\omega) = \sum_{z_h \in \mathcal{Z}} \frac{\omega_h}{\sigma_h^2} \phi(z_h) \phi(z_h)^\top$, *and*

$$
kl(\delta, 1 - \delta) \triangleq \delta \log \left( \frac{\delta}{1 - \delta} \right) + (1 - \delta) \log \left( \frac{1 - \delta}{\delta} \right).
$$

Intuitively, arms from different classes are governed by different types of constraints. For the best arm, the lower bound is determined by the most critical feasible constraint, i.e., the one closest to violation. In contrast, for infeasible arms, the lower bound reflects the combined effect of all violated constraints.

### A.4. Proof of Proposition 3.3

Proposition 3.3 follows from a direct extension of the proof of Theorem 3 in (Jedra & Proutiere, 2020), together with the tracking argument in Lemma 17 of (Garivier & Kaufmann, 2016). Since the extension follows the same Track-and-Stop structure after replacing the original complexity term with our constrained covariate-dependent complexity term $\mathcal{H}^*(\mathcal{P})$, we omit the proof for brevity. Jedra & Proutiere (2020) considered the case where the optimal sampling ratio $\omega^*(\mathcal{P})$ may be non-unique. Specifically, it proposed the following sampling rule:

$$
z_{h(t+1)} = \arg \min_{z_h \in \mathcal{Z}} N_h(t) - \sum_{s=1}^{t} \omega_h^*(\hat{\mathcal{P}}(s)),
\tag{66}
$$

where $\omega^*(\hat{\mathcal{P}}(s))$ denotes an arbitrary measurable selection from the optimizer set $\mathcal{M}^*(\hat{\mathcal{P}}(s))$, and showed that the empirical sampling ratio converges to the set $\mathcal{M}^*(\mathcal{P})$, defined as

$$
\mathcal{M}^*(\mathcal{P}) = \arg \max_{\omega \in \Omega} \mathcal{H}(\omega, \mathcal{P})^{-1}.
\tag{67}
$$

This sampling rule in (66) can also be applied in our setting to handle the non-unique optimal sampling ratio case. Following the same analysis as in Lemma A.2, we can show that $\mathcal{H}(\omega, \mathcal{P})^{-1}$ is continuous in $(\omega, \mathcal{P})$ in a neighborhood of the true instance $\mathcal{P}$. Moreover, $\Omega$ is a non-empty, compact, and convex simplex. For each fixed $\mathcal{P}$, the function $\omega \mapsto \mathcal{H}(\omega, \mathcal{P})^{-1}$ is concave, since it can be expressed as the infimum over linear functions of $\omega$. Therefore, by the maximum theorem, the optimizer set

$$
\mathcal{M}^*(\mathcal{P}) = \arg \max_{\omega \in \Omega} \mathcal{H}(\omega, \mathcal{P})^{-1}
$$

is non-empty and compact. In addition, because $\mathcal{H}(\omega, \mathcal{P})^{-1}$ is concave in $\omega$ and $\Omega$ is convex, the optimizer set $\mathcal{M}^*(\mathcal{P})$ is convex. Hence, any convex combination of elements in $\mathcal{M}^*(\mathcal{P})$ also belongs to $\mathcal{M}^*(\mathcal{P})$. Therefore, this modification ensures that the empirical sampling ratio converges to the optimizer set $\mathcal{M}^*(\mathcal{P})$, i.e.,

$$
\mathrm{dist}(\omega(t), \mathcal{M}^*(\mathcal{P})) \to 0.
$$

## A.5. Proof of Lemma 3.4

*Proof.* This lemma establishes that the relaxed complexity $\mathcal{U}^*(\mathcal{P})$ serves as an upper bound on the instance-dependent complexity $\mathcal{H}^*(\mathcal{P})$. Note that for each $\omega \in \Omega$, $c_j \in \mathcal{C}$, we have

$$
\begin{aligned}
\Gamma(\omega, c_j, \mathcal{P}) &= \frac{1}{2} \min \left( \min_{x_i \neq x_{i^*(c_j)}} \left( \frac{((\phi(x_{i^*(c_j)}, c_j) - \phi(x_i, c_j))^\top \theta)^2}{\|\phi(x_{i^*(c_j)}, c_j) - \phi(x_i, c_j)\|_{\Lambda(\omega)^{-1}}^2} \mathbb{I}(x_i \in \mathcal{D}_1(c_j) \cup \mathcal{D}_3(c_j)) \right. \right. \\
&\quad \left. + \frac{(b - \beta^\top \phi(x_i, c_j))^2}{\|\phi(x_i, c_j)\|_{\Lambda(\omega)^{-1}}^2} \mathbb{I}(x_i \in \mathcal{D}_2(c_j) \cup \mathcal{D}_3(c_j)) \right), \frac{(b - \beta^\top \phi(x_{i^*(c_j)}, c_j))^2}{\|\phi(x_{i^*(c_j)}, c_j)\|_{\Lambda(\omega)^{-1}}^2} \right) \\
&\geq \frac{1}{2} \min_{x_i \in \mathcal{X}} \left( \frac{((\phi(x_{i^*(c_j)}, c_j) - \phi(x_i, c_j))^\top \theta)^2}{\|\phi(x_{i^*(c_j)}, c_j) - \phi(x_i, c_j)\|_{\Lambda(\omega)^{-1}}^2} \mathbb{I}(x_i \in \mathcal{D}_1(c_j)) \right. \\
&\quad \left. + \frac{(b - \beta^\top \phi(x_i, c_j))^2}{\|\phi(x_i, c_j)\|_{\Lambda(\omega)^{-1}}^2} \mathbb{I}(x_i \in \{x_{i^*(c_j)}\} \cup \mathcal{D}_2(c_j) \cup \mathcal{D}_3(c_j)) \right) \\
&= \Gamma^s(\omega, c_j, \mathcal{P}).
\end{aligned}
\tag{68}
$$

Then, we conclude that

$$
\mathcal{H}^*(\mathcal{P})^{-1} = \max_{\omega \in \Omega} \min_{c_j \in \mathcal{C}} \Gamma(\omega, c_j, \mathcal{P}) \geq \max_{\omega \in \Omega} \min_{c_j \in \mathcal{C}} \Gamma^s(\omega, c_j, \mathcal{P}) = \mathcal{U}^*(\mathcal{P})^{-1},
\tag{69}
$$

and therefore $\mathcal{H}^*(\mathcal{P}) \leq \mathcal{U}^*(\mathcal{P})$. $\qquad \square$

## A.6. Relaxation Gap Analysis

In this subsection, we analyze the gap between the relaxed bound $\mathcal{U}^*(\mathcal{P})$ and the original bound $\mathcal{H}^*(\mathcal{P})$.

Let $\Omega_+ \triangleq \{\omega \in \Omega : \Lambda(\omega) \succ 0\}$. For $x_i \neq x_{i^*(c_j)}$, define

$$
A_i(\omega, c_j) \triangleq \frac{\left((\phi(x_{i^*(c_j)}, c_j) - \phi(x_i, c_j))^\top \theta\right)^2}{\left\|\phi(x_{i^*(c_j)}, c_j) - \phi(x_i, c_j)\right\|_{\Lambda(\omega)^{-1}}^2},
$$

and

$$
B_i(\omega, c_j) \triangleq \frac{\left(b - \beta^\top \phi(x_i, c_j)\right)^2}{\|\phi(x_i, c_j)\|_{\Lambda(\omega)^{-1}}^2}.
$$

For notational convenience, set

$$
A_{i^*(c_j)}(\omega, c_j) = 0.
$$

When $\Lambda(\omega)$ is singular, all quantities involving $\Lambda(\omega)^{-1}$ are interpreted under the extended inverse convention. In particular, values at boundary points in $\Omega \setminus \Omega_+$ are understood as limits from $\Omega_+$.

Assume that the following finite instance-dependent constant exists:

$$
\gamma \triangleq \sup_{\omega \in \Omega_+} \sup_{c_j \in \mathcal{C}} \sup_{x_i \in \mathcal{D}_3(c_j)} \frac{A_i(\omega, c_j)}{B_i(\omega, c_j)} < \infty.
$$

When $\mathcal{D}_3(c_j)$ is empty, the corresponding supremum is taken to be zero.

By the definition of $\gamma$, for any $x_i \in \mathcal{D}_3(c_j)$ and $\omega \in \Omega_+$, we have

$$
A_i(\omega, c_j) + B_i(\omega, c_j) \leq (1 + \gamma) B_i(\omega, c_j).
$$

Therefore, for each $\omega \in \Omega_+$ and $c_j \in \mathcal{C}$,

$$\Gamma(\omega, c_j, \mathcal{P}) = \frac{1}{2} \min \left[ \min_{x_i \neq x_{i^*(c_j)}} \left( A_i(\omega, c_j) \mathbb{I}\big(x_i \in \mathcal{D}_1(c_j) \cup \mathcal{D}_3(c_j)\big) \right.\right.$$
$$\left. + B_i(\omega, c_j) \mathbb{I}\big(x_i \in \mathcal{D}_2(c_j) \cup \mathcal{D}_3(c_j)\big) \right),$$
$$\left. \frac{\big(b - \beta^\top \phi(x_{i^*(c_j)}, c_j)\big)^2}{\big\|\phi(x_{i^*(c_j)}, c_j)\big\|^2_{\Lambda(\omega)^{-1}}} \right]$$
$$\leq (1 + \gamma) \frac{1}{2} \min_{x_i \in \mathcal{X}} \left( A_i(\omega, c_j) \mathbb{I}\big(x_i \in \mathcal{D}_1(c_j)\big) \right.$$
$$\left. + B_i(\omega, c_j) \mathbb{I}\big(x_i \in \{x_{i^*(c_j)}\} \cup \mathcal{D}_2(c_j) \cup \mathcal{D}_3(c_j)\big) \right)$$
$$= (1 + \gamma) \Gamma^s(\omega, c_j, \mathcal{P}).$$

The same inequality extends to $\Omega \setminus \Omega_+$ by the extended inverse convention. Hence,

$$\mathcal{H}^*(\mathcal{P})^{-1} = \max_{\omega \in \Omega} \min_{c_j \in \mathcal{C}} \Gamma(\omega, c_j, \mathcal{P}) \leq (1 + \gamma) \max_{\omega \in \Omega} \min_{c_j \in \mathcal{C}} \Gamma^s(\omega, c_j, \mathcal{P}) = (1 + \gamma) \mathcal{U}^*(\mathcal{P})^{-1}.$$

Hence,

$$\mathcal{H}^*(\mathcal{P}) \leq \mathcal{U}^*(\mathcal{P}) \leq (1 + \gamma) \mathcal{H}^*(\mathcal{P}).$$

The bound for $\mathcal{U}^*(\mathcal{P})$ becomes tight when, for arms in $\mathcal{D}_3(c_j)$, the objective-gap contribution is negligible relative to the feasibility-gap contribution. In this case, the ratio $A_i(\omega, c_j)/B_i(\omega, c_j)$ is small, and hence the constant $\gamma$ is close to zero.

We also propose an alternative relaxed bound $\tilde{\mathcal{U}}^*(\mathcal{P})$ by further partitioning the set $\mathcal{D}_3(c_j)$. For each $\omega \in \Omega$ and $c_j \in \mathcal{C}$, partition $\mathcal{D}_3(c_j)$ into

$$\mathcal{M}_1(\omega, c_j) \triangleq \{x_i \in \mathcal{D}_3(c_j) : B_i(\omega, c_j) \leq A_i(\omega, c_j)\},$$

and

$$\mathcal{M}_2(\omega, c_j) \triangleq \{x_i \in \mathcal{D}_3(c_j) : B_i(\omega, c_j) > A_i(\omega, c_j)\}.$$

Thus, for arms in $\mathcal{M}_1(\omega, c_j)$, the objective-gap contribution is at least as large as the feasibility-gap contribution, whereas for arms in $\mathcal{M}_2(\omega, c_j)$, the feasibility-gap contribution is larger.

Based on this partition, define

$$\tilde{\mathcal{U}}^*(\mathcal{P})^{-1} \triangleq \max_{\omega \in \Omega} \min_{c_j \in \mathcal{C}} \tilde{\Gamma}^s(\omega, c_j, \mathcal{P}),$$

where

$$\tilde{\Gamma}^s(\omega, c_j, \mathcal{P}) = \frac{1}{2} \min_{x_i \in \mathcal{X}} \left( A_i(\omega, c_j) \mathbb{I}\big(x_i \in \mathcal{D}_1(c_j) \cup \mathcal{M}_1(\omega, c_j)\big) \right. \tag{70}$$
$$\left. + B_i(\omega, c_j) \mathbb{I}\big(x_i \in \{x_{i^*(c_j)}\} \cup \mathcal{D}_2(c_j) \cup \mathcal{M}_2(\omega, c_j)\big) \right).$$

By convention, any term multiplied by a zero indicator is defined as zero, even if the corresponding fraction is of the form $0/0$.

For each $x_i \in \mathcal{D}_3(c_j)$, the original objective contains $A_i(\omega, c_j) + B_i(\omega, c_j)$, whereas the alternative surrogate keeps $\max\{A_i(\omega, c_j), B_i(\omega, c_j)\}$. Therefore,

$$\max\{A_i(\omega, c_j), B_i(\omega, c_j)\} \leq A_i(\omega, c_j) + B_i(\omega, c_j) \leq 2\max\{A_i(\omega, c_j), B_i(\omega, c_j)\}.$$

It follows that, for every $\omega \in \Omega$ and $c_j \in \mathcal{C}$,

$$\tilde{\Gamma}^s(\omega, c_j, \mathcal{P}) \leq \Gamma(\omega, c_j, \mathcal{P}) \leq 2\tilde{\Gamma}^s(\omega, c_j, \mathcal{P}).$$

Consequently,

$$\mathcal{H}^*(\mathcal{P}) \leq \tilde{\mathcal{U}}^*(\mathcal{P}) \leq 2\mathcal{H}^*(\mathcal{P}).$$

The bound $\mathcal{U}^*(\mathcal{P})$ becomes tight when, for arms in $\mathcal{D}_3(c_j)$, the objective-gap contribution is negligible relative to the feasibility-gap contribution. When this is not the case, the alternative bound $\tilde{\mathcal{U}}^*(\mathcal{P})$ provides a uniform factor-2 relaxation gap.

Since the theoretical analysis of the two bounds is essentially the same, we focus on $\mathcal{U}^*(\mathcal{P})$ in the main paper for notational simplicity.

### A.7. Proof of Theorem 3.5

*Proof.* Consider the following primal optimization problem in (18):

$$\max_{\omega \in \Omega} \min_{c_j \in \mathcal{C}} \Gamma^s(\omega, c_j, \mathcal{P}), \tag{71}$$

where

$$\Gamma^s(\omega, c_j, \mathcal{P}) = \frac{1}{2} \min_{x_i \in \mathcal{X}} \left( \frac{((\phi(x_{i^*(c_j)}, c_j) - \phi(x_i, c_j))^\top \theta)^2}{\|\phi(x_{i^*(c_j)}, c_j) - \phi(x_i, c_j)\|^2_{\Lambda(\omega)^{-1}}} \mathbb{I}(x_i \in \mathcal{D}_1(c_j)) \right.$$

$$\left. + \frac{(b - \beta^\top \phi(x_i, c_j))^2}{\|\phi(x_i, c_j)\|^2_{\Lambda(\omega)^{-1}}} \mathbb{I}(x_i \in \{x_{i^*(c_j)}\} \cup \mathcal{D}_2(c_j) \cup \mathcal{D}_3(c_j)) \right). \tag{72}$$

Since the factor $1/2$ does not affect the optimizer and all relevant terms are positive, maximizing the minimum information rate is equivalent to minimizing the maximum reciprocal variance term. We derive the Lagrange dual of this reciprocal reformulation, which has the same optimal sampling allocation as (18).

$$\min_{\omega \in \Omega} \max_{c_j \in \mathcal{C}, x_i \in \mathcal{X}} \left( \frac{\|\phi(x_{i^*(c_j)}, c_j) - \phi(x_i, c_j)\|^2_{\Lambda(\omega)^{-1}}}{((\phi(x_{i^*(c_j)}, c_j) - \phi(x_i, c_j))^\top \theta)^2} \mathbb{I}(x_i \in \mathcal{D}_1(c_j)) \right.$$

$$\left. + \frac{\|\phi(x_i, c_j)\|^2_{\Lambda(\omega)^{-1}}}{(b - \beta^\top \phi(x_i, c_j))^2} \mathbb{I}(x_i \in \{x_{i^*(c_j)}\} \cup \mathcal{D}_2(c_j) \cup \mathcal{D}_3(c_j)) \right). \tag{73}$$

For the dual derivation, we work on the relative interior of the simplex. The boundary case is handled by the extended convention that when $\Lambda(\omega)$ is singular, $\|\cdot\|_{\Lambda(\omega)^{-1}}$ is interpreted in the extended sense, with uncovered directions assigned infinite variance.

By introducing an auxiliary variable $\xi$, we can reformulate the optimization problem as:

$$\min_{\xi, \omega} \xi$$

$$\text{s.t.} \quad \frac{\|\phi(x_{i^*(c_j)}, c_j) - \phi(x_i, c_j)\|^2_{\Lambda(\omega)^{-1}}}{((\phi(x_{i^*(c_j)}, c_j) - \phi(x_i, c_j))^\top \theta)^2} \leq \xi, \forall c_j \in \mathcal{C}, x_i \in \mathcal{D}_1(c_j)$$

$$\frac{\|\phi(x_i, c_j)\|^2_{\Lambda(\omega)^{-1}}}{(b - \beta^\top \phi(x_i, c_j))^2} \leq \xi, \forall c_j \in \mathcal{C}, x_i \in \{x_{i^*(c_j)}\} \cup \mathcal{D}_2(c_j) \cup \mathcal{D}_3(c_j) \tag{74}$$

$$\sum_{h \in [D]} \omega_h = 1$$

$$\omega_h \geq 0, \forall h \in [D]$$

Let $\Phi \in \mathbb{R}^{D \times D}$ denote the design matrix whose rows are the design feature vectors, and let

$$\Sigma(\omega) = \text{diag}\left( \frac{\sigma_1^2}{\omega_1}, \dots, \frac{\sigma_D^2}{\omega_D} \right).$$

Since the design points span $\mathbb{R}^D$, $\Phi$ is invertible. Therefore,

$$\Lambda(\omega)^{-1} = \left( \sum_{h \in [D]} \omega_h \frac{\phi(z_h) \phi(z_h)^\top}{\sigma_h^2} \right)^{-1} = (\Phi^T \Sigma(\omega)^{-1} \Phi)^{-1} = \Phi^{-1} \Sigma(\omega) (\Phi^T)^{-1}. \tag{75}$$

Now, for each covariate $c_j \in \mathcal{C}$ and each arm $x_i \in \mathcal{D}_1(c_j)$, we have

$$
\frac{\|\phi(x_{i^*(c_j)}, c_j) - \phi(x_i, c_j)\|^2_{\Lambda(\omega)^{-1}}}{((\phi(x_{i^*(c_j)}, c_j) - \phi(x_i, c_j))^\top \theta)^2}
$$

$$
= \frac{(\phi(x_{i^*(c_j)}, c_j) - \phi(x_i, c_j))^\top \Lambda(\omega)^{-1} (\phi(x_{i^*(c_j)}, c_j) - \phi(x_i, c_j))}{((\phi(x_{i^*(c_j)}, c_j) - \phi(x_i, c_j))^\top \theta)^2}
$$

$$
= \frac{(\phi(x_{i^*(c_j)}, c_j) - \phi(x_i, c_j))^\top \Phi^{-1} \Sigma (\Phi^T)^{-1} (\phi(x_{i^*(c_j)}, c_j) - \phi(x_i, c_j))}{((\phi(x_{i^*(c_j)}, c_j) - \phi(x_i, c_j))^\top \theta)^2} \tag{76}
$$

$$
= \sum_{h \in [D]} \frac{\sigma_h^2 [(\Phi^T)^{-1}(\phi(x_{i^*(c_j)}, c_j) - \phi(x_i, c_j))]_h^2}{\omega_h ((\phi(x_{i^*(c_j)}, c_j) - \phi(x_i, c_j))^\top \theta)^2}
$$

$$
= \sum_{h \in [D]} \frac{\chi_h(x_i, c_j)}{\omega_h},
$$

where we define

$$
\chi_h(x_i, c_j) = \frac{\sigma_h^2 [(\Phi^T)^{-1}(\phi(x_{i^*(c_j)}, c_j) - \phi(x_i, c_j))]_h^2}{((\phi(x_{i^*(c_j)}, c_j) - \phi(x_i, c_j))^\top \theta)^2}, \tag{77}
$$

and $[v]_h$ denotes the $h$-th element of the vector $v$.

Similarly, for each covariate $c_j \in \mathcal{C}$ and each arm $x_i \in \{x_{i^*(c_j)}\} \cup \mathcal{D}_2(c_j) \cup \mathcal{D}_3(c_j)$, we have

$$
\frac{\|\phi(x_i, c_j)\|^2_{\Lambda(\omega)^{-1}}}{(b - \beta^\top \phi(x_i, c_j))^2}
$$

$$
= \frac{\phi(x_i, c_j)^\top \Lambda(\omega)^{-1} \phi(x_i, c_j)}{(b - \beta^\top \phi(x_i, c_j))^2}
$$

$$
= \frac{\phi(x_i, c_j)^\top \Phi^{-1} \Sigma (\Phi^T)^{-1} \phi(x_i, c_j)}{(b - \beta^\top \phi(x_i, c_j))^2} \tag{78}
$$

$$
= \sum_{h \in [D]} \frac{\sigma_h^2 [(\Phi^T)^{-1} \phi(x_i, c_j)]_h^2}{\omega_h (b - \beta^\top \phi(x_i, c_j))^2}
$$

$$
= \sum_{h \in [D]} \frac{\chi_h(x_i, c_j)}{\omega_h},
$$

where we define

$$
\chi_h(x_i, c_j) = \frac{\sigma_h^2 [(\Phi^T)^{-1} \phi(x_i, c_j)]_h^2}{(b - \beta^\top \phi(x_i, c_j))^2}. \tag{79}
$$

Hence, the optimization problem becomes:

$$
\min_{\omega, \xi} \xi
$$

$$
\text{s.t.} \sum_{h \in [D]} \frac{\chi_h(x_i, c_j)}{\omega_h} \le \xi, \forall c_j \in \mathcal{C}, x_i \in \mathcal{X} \quad (\lambda_{ij})
$$

$$
\sum_{h \in [D]} \omega_h = 1, \quad (\nu) \tag{80}
$$

$$
\omega_h \ge 0, \forall h \in [D]
$$

The corresponding Lagrangian function is:

$$
L(\xi, \omega, \lambda, \nu) = \xi + \sum_{j \in [M], i \in [K]} \lambda_{ij} \left( \sum_{h \in [D]} \frac{\chi_h(x_i, c_j)}{\omega_h} - \xi \right) + \nu \left( \sum_{h \in [D]} \omega_h - 1 \right). \tag{81}
$$

Let $(\xi^*, \omega^*, \lambda^*, \nu^*)$ denote the optimal primal-dual solution. The KKT conditions for this optimization problem are:

$$\sum_{j \in [M], i \in [K]} \lambda_{ij}^* = 1$$

$$-\sum_{j \in [M], i \in [K]} \lambda_{ij}^* \frac{\chi_h(x_i, c_j)}{(\omega_h^*)^2} + \nu^* = 0$$

$$\lambda_{ij}^* \left( \sum_{h \in [D]} \frac{\chi_h(x_i, c_j)}{\omega_h^*} - \xi^* \right) = 0, \forall j \in [M], i \in [K]$$

$$\lambda_{ij}^* \geq 0, \forall j \in [M], i \in [K] \tag{82}$$

$$\sum_{h \in [D]} \frac{\chi_h(x_i, c_j)}{\omega_h^*} \leq \xi^*, \forall c_j \in \mathcal{C}, x_i \in \mathcal{X}$$

$$\sum_{h \in [D]} \omega_h^* = 1$$

$$\omega_h^* \geq 0, \forall h \in [D].$$

From the second and sixth equations, we deduce the optimal form of $\omega_h^*$. Solving the second equation, we obtain:

$$\omega_h^* = \sqrt{\frac{\sum_{j \in [M], i \in [K]} \lambda_{ij}^* \chi_h(x_i, c_j)}{\nu^*}}, \tag{83}$$

Using the sixth equation, we normalize the solution:

$$\omega_h^* = \frac{\sqrt{\sum_{j \in [M], i \in [K]} \lambda_{ij}^* \chi_h(x_i, c_j)}}{\sum_{l \in [D]} \sqrt{\sum_{j \in [M], i \in [K]} \lambda_{ij}^* \chi_l(x_i, c_j)}}. \tag{84}$$

We now derive the Lagrange dual function.

$$g(\lambda, \nu) = \inf_{\xi, \omega} L(\xi, \omega, \lambda, \nu)$$

$$= \inf_{\xi, \omega} (1 - \sum_{j \in [M], i \in [K]} \lambda_{ij}) \xi + \sum_{j \in [M], i \in [K]} \lambda_{ij} \sum_{h \in [D]} \frac{\chi_h(x_i, c_j)}{\omega_h} + \nu (\sum_{h \in [D]} \omega_h - 1)$$

$$= \begin{cases} \inf_\omega \sum_{j \in [M], i \in [K]} \lambda_{ij} \sum_{h \in [D]} \frac{\chi_h(x_i, c_j)}{\omega_h} + \nu(\sum_{h \in [D]} \omega_h - 1) & \text{if } \sum_{j \in [M], i \in [K]} \lambda_{ij} = 1, \lambda_{ij} \geq 0 \\ -\infty & \text{o.w.} \end{cases} \tag{85}$$

$$= \begin{cases} 2\sqrt{\nu} \sum_{h \in [D]} \sqrt{\sum_{j \in [M], i \in [K]} \lambda_{ij} \chi_h(x_i, c_j)} - \nu & \text{if } \sum_{j \in [M], i \in [K]} \lambda_{ij} = 1, \lambda_{ij} \geq 0, \nu \geq 0 \\ -\infty & \text{o.w.} \end{cases}$$

By optimizing the variable $\nu$, we can obtain that the dual optimization problem is

$$\max_\lambda \left( \sum_{h \in [D]} \sqrt{\sum_{j \in [M], i \in [K]} \lambda_{ij} \chi_h(x_i, c_j)} \right)^2$$

$$\text{s.t.} \sum_{j \in [M], i \in [K]} \lambda_{ij} = 1 \tag{86}$$

$$\lambda_{ij} \geq 0, \forall i \in [K], j \in [M].$$

Because the square function is monotone increasing on $\mathbb{R}_+$, this problem has the same optimizer as

$$
\begin{aligned}
\min_{\lambda} \ -&\sum_{h \in [D]} \sqrt{\sum_{j \in [M], \, i \in [K]} \lambda_{ij} \chi_h(x_i, c_j)} \\
\text{s.t.} \quad &\sum_{j \in [M], \, i \in [K]} \lambda_{ij} = 1, \\
&\lambda_{ij} \geq 0, \quad \forall i \in [K], \ j \in [M],
\end{aligned}
\tag{87}
$$

which is the equivalent dual formulation stated in Theorem 3.5. $\qquad\square$

## A.8. Proof of Lemma 3.6

*Proof.* As shown in the proof of Theorem 3.1, the optimization problem (18) can be equivalently derived from the following formulation:

$$
\max_{\omega \in \Omega} \min_{c_j \in \mathcal{C}} \min \left( \inf_{\tilde{\mathcal{P}} \in \mathcal{O}(x_{i^*(c_j, \mathcal{P})}, c_j)} \mathcal{H}(\omega, \mathcal{P}, \tilde{\mathcal{P}})^{-1}, \ \min_{x_i \in \mathcal{D}_1(c_j)} \inf_{\tilde{\mathcal{P}} \in \mathcal{O}_1(x_i, c_j)} \mathcal{H}(\omega, \mathcal{P}, \tilde{\mathcal{P}})^{-1}, \right.
$$
$$
\left. \min_{x_i \in \mathcal{D}_2(c_j) \cup \mathcal{D}_3(c_j)} \inf_{\tilde{\mathcal{P}} \in \mathcal{O}_2(x_i, c_j)} \mathcal{H}(\omega, \mathcal{P}, \tilde{\mathcal{P}})^{-1} \right),
\tag{88}
$$

where the sets and functionals are defined as follows:

$$
\begin{aligned}
\mathcal{O}(x_{i^*(c_j, \mathcal{P})}, c_j) &= \left\{ \tilde{\mathcal{P}} \in \mathcal{S} : \tilde{\beta}^\top \phi(x_{i^*(c_j, \mathcal{P})}, c_j) > b \right\}, \\
\mathcal{O}_1(x_i, c_j) &= \left\{ \tilde{\mathcal{P}} \in \mathcal{S} : \tilde{\theta}^\top \left( \phi(x_i, c_j) - \phi(x_{i^*(c_j, \mathcal{P})}, c_j) \right) > 0 \right\}, \\
\mathcal{O}_2(x_i, c_j) &= \left\{ \tilde{\mathcal{P}} \in \mathcal{S} : \tilde{\beta}^\top \phi(x_i, c_j) \leq b \right\}, \\
\mathcal{H}(\omega, \mathcal{P}, \tilde{\mathcal{P}})^{-1} &= \sum_{h \in [D]} \omega_h \left( d(f(z_h), \tilde{f}(z_h)) + d(g(z_h), \tilde{g}(z_h)) \right).
\end{aligned}
\tag{89}
$$

Note that $\mathcal{H}(\omega, \mathcal{P}, \tilde{\mathcal{P}})^{-1}$ is a convex function of $\tilde{\mathcal{P}}$, due to the convexity of the KL divergence in its second argument. In the following infimum calculations, we replace the original alternative sets by their corresponding closed halfspace relaxations. This replacement does not change the infimum values, because $\mathcal{H}(\omega, \mathcal{P}, \tilde{\mathcal{P}})^{-1}$ is continuous in $\tilde{\mathcal{P}}$, and the boundary points of the relaxed sets can be approximated arbitrarily closely by valid alternative instances in $\mathcal{S}$. The relaxed feasible sets are convex, since they are defined by linear inequalities in $(\tilde{\theta}, \tilde{\beta})$.

Therefore, for fixed $\omega \in \Omega$, the following infimum values are obtained from convex minimization problems over the corresponding relaxed feasible sets:

$$
\begin{aligned}
\mathcal{L}(x_{i^*(c_j, \mathcal{P})}, c_j, \omega, \mathcal{P}) &= \inf_{\tilde{\mathcal{P}} \in \mathcal{O}(x_{i^*(c_j, \mathcal{P})}, c_j)} \mathcal{H}(\omega, \mathcal{P}, \tilde{\mathcal{P}})^{-1}, \\
\mathcal{L}_1(x_i, c_j, \omega, \mathcal{P}) &= \inf_{\tilde{\mathcal{P}} \in \mathcal{O}_1(x_i, c_j)} \mathcal{H}(\omega, \mathcal{P}, \tilde{\mathcal{P}})^{-1}, \\
\mathcal{L}_2(x_i, c_j, \omega, \mathcal{P}) &= \inf_{\tilde{\mathcal{P}} \in \mathcal{O}_2(x_i, c_j)} \mathcal{H}(\omega, \mathcal{P}, \tilde{\mathcal{P}})^{-1}.
\end{aligned}
\tag{90}
$$

The resulting functions $\mathcal{L}(x_{i^*(c_j, \mathcal{P})}, c_j, \omega, \mathcal{P})$, $\mathcal{L}_1(x_i, c_j, \omega, \mathcal{P})$, and $\mathcal{L}_2(x_i, c_j, \omega, \mathcal{P})$ are concave in $\omega$, because each is the pointwise infimum of functions that are affine in $\omega$. Consequently, the objective in (88) is the pointwise minimum of a finite collection of concave functions, and is therefore concave in $\omega$.

To justify strong duality, write (88) in hypograph form:

$$
\begin{aligned}
\max_{\omega, q} \quad & q \\
\text{s.t.} \quad & q \le \mathcal{L}(x_{i^*(c_j, \mathcal{P})}, c_j, \omega, \mathcal{P}), \quad \forall c_j \in \mathcal{C}, \\
& q \le \mathcal{L}_1(x_i, c_j, \omega, \mathcal{P}), \qquad \forall c_j \in \mathcal{C}, \ x_i \in \mathcal{D}_1(c_j), \\
& q \le \mathcal{L}_2(x_i, c_j, \omega, \mathcal{P}), \qquad \forall c_j \in \mathcal{C}, \ x_i \in \mathcal{D}_2(c_j) \cup \mathcal{D}_3(c_j), \\
& \sum_{h \in [D]} \omega_h = 1, \\
& \omega_h \ge 0, \qquad \forall h \in [D].
\end{aligned}
\tag{91}
$$

Since $\mathcal{L}, \mathcal{L}_1,$ and $\mathcal{L}_2$ are concave in $\omega$, the constraints in (91) are convex after being written as

$$
q - \mathcal{L}(\cdot) \le 0, \qquad q - \mathcal{L}_1(\cdot) \le 0, \qquad q - \mathcal{L}_2(\cdot) \le 0.
$$

Moreover, Slater's condition holds. Indeed, choose any $\omega^0$ in the relative interior of $\Omega$, for example $\omega_h^0 = 1/D$ for all $h \in [D]$. Since the KL divergence is nonnegative, all the above infimum values are bounded below. Thus, one can choose $q^0$ sufficiently small such that all hypograph inequalities hold strictly at $(\omega^0, q^0)$. Hence, by Slater's theorem, strong duality holds.

By (80), this optimization problem is equivalent to

$$
\begin{aligned}
\min_{\omega} f(\omega) &= \max_{c_j \in \mathcal{C}, x_i \in \mathcal{X}} \sum_{h \in [D]} \frac{\chi_h(x_i, c_j)}{\omega_h} \\
\text{s.t.} \sum_{h \in [D]} \omega_h &= 1, \\
\omega_h &\ge 0, \forall h \in [D]
\end{aligned}
\tag{92}
$$

We use the convention that $0/0 = 0$ and $a/0 = +\infty$ for any $a > 0$. Since $\chi_h(x_i, c_j) \ge 0$, for each fixed pair $(x_i, c_j)$, the function

$$
\sum_{h \in [D]} \frac{\chi_h(x_i, c_j)}{\omega_h}
$$

is convex on $\Omega$. Therefore, $f(\omega)$ is convex.

We now show that this problem admits a unique optimal solution. Let $\omega$ and $\omega'$ be two optimal solutions, and let

$$
\bar{\omega} = \lambda \omega + (1 - \lambda)\omega', \qquad \lambda \in (0, 1).
$$

By convexity of $f$, $\bar{\omega}$ is also optimal. Since the interior point $\omega_h = 1/D$ for all $h \in [D]$ yields a finite objective value, the optimal value is finite. Therefore, for any optimal solution $\bar{\omega}$ and any active pair $(x_i, c_j)$, if $\chi_h(x_i, c_j) > 0$, then $\bar{\omega}_h > 0$; otherwise, the corresponding term $\chi_h(x_i, c_j)/\bar{\omega}_h$ would be $+\infty$.

Let $\mathcal{A}(\bar{\omega})$ denote the set of active pairs $(x_i, c_j)$ at $\bar{\omega}$, i.e.,

$$
\mathcal{A}(\bar{\omega}) = \left\{ (x_i, c_j) : \sum_{h \in [D]} \frac{\chi_h(x_i, c_j)}{\bar{\omega}_h} = f(\bar{\omega}) \right\}.
$$

We first show that the supports of the active constraints cover all coordinates receiving positive allocation under $\bar{\omega}$:

$$
\{h : \bar{\omega}_h > 0\} \subseteq \bigcup_{(x_i, c_j) \in \mathcal{A}(\bar{\omega})} \{h : \chi_h(x_i, c_j) > 0\}.
\tag{93}
$$

Suppose otherwise. Then there exists $h_0$ such that $\bar{\omega}_{h_0} > 0$ and

$$
\chi_{h_0}(x_i, c_j) = 0, \qquad \forall (x_i, c_j) \in \mathcal{A}(\bar{\omega}).
$$

Let

$$S = \bigcup_{(x_i, c_j) \in \mathcal{A}(\bar{\omega})} \{h : \chi_h(x_i, c_j) > 0\}.$$

Under the non-degeneracy of the alternatives, $S$ is nonempty. Choose constants $a_h > 0$ for $h \in S$ such that $\sum_{h \in S} a_h = 1$, and define a feasible direction $v$ by

$$v_{h_0} = -1, \qquad v_h = a_h \text{ for } h \in S, \qquad v_h = 0 \text{ otherwise.}$$

For all sufficiently small $\epsilon > 0$,

$$\omega^\epsilon = \bar{\omega} + \epsilon v$$

belongs to $\Omega$. For every active pair $(x_i, c_j) \in \mathcal{A}(\bar{\omega})$, the coordinate $h_0$ does not appear in its support, while every coordinate in its support receives additional mass. Therefore,

$$\sum_{h \in [D]} \frac{\chi_h(x_i, c_j)}{\omega_h^\epsilon} < \sum_{h \in [D]} \frac{\chi_h(x_i, c_j)}{\bar{\omega}_h} = f(\bar{\omega})$$

for all active pairs and all sufficiently small $\epsilon > 0$. Since the number of inactive pairs is finite and each inactive pair has a strict slack below $f(\bar{\omega})$ at $\bar{\omega}$, by continuity, we also have

$$\sum_{h \in [D]} \frac{\chi_h(x_i, c_j)}{\omega_h^\epsilon} < f(\bar{\omega})$$

for all inactive pairs when $\epsilon$ is sufficiently small. Therefore,

$$f(\omega^\epsilon) < f(\bar{\omega}),$$

which contradicts the optimality of $\bar{\omega}$. This proves (93).

Now take any active pair $(x_i, c_j) \in \mathcal{A}(\bar{\omega})$. Since $\omega$ and $\omega'$ are both optimal,

$$\sum_{h \in [D]} \frac{\chi_h(x_i, c_j)}{\omega_h} \leq f(\omega), \qquad \sum_{h \in [D]} \frac{\chi_h(x_i, c_j)}{\omega_h'} \leq f(\omega').$$

Using the convexity of $1/x$, we have

$$f(\bar{\omega}) = \sum_{h \in [D]} \frac{\chi_h(x_i, c_j)}{\lambda \omega_h + (1 - \lambda) \omega_h'}$$

$$\leq \lambda \sum_{h \in [D]} \frac{\chi_h(x_i, c_j)}{\omega_h} + (1 - \lambda) \sum_{h \in [D]} \frac{\chi_h(x_i, c_j)}{\omega_h'}$$

$$\leq \lambda f(\omega) + (1 - \lambda) f(\omega') = f(\bar{\omega}).$$

Therefore, all inequalities above must hold with equality. For any coordinate $h$ such that $\chi_h(x_i, c_j) > 0$, the finiteness of the optimal value implies

$$\omega_h > 0, \qquad \omega_h' > 0, \qquad \bar{\omega}_h > 0.$$

Thus, the strict convexity of $1/x$ on $(0, \infty)$ applies. Equality in the convexity inequality then implies

$$\omega_h = \omega_h' \quad \text{for every } h \text{ such that } \chi_h(x_i, c_j) > 0.$$

Because the supports of the active constraints cover all coordinates with $\bar{\omega}_h > 0$, it follows that

$$\omega_h = \omega_h' \quad \text{for all } h \text{ with } \bar{\omega}_h > 0.$$

For coordinates with $\bar{\omega}_h = 0$, the nonnegativity of $\omega_h$ and $\omega_h'$ together with

$$\bar{\omega}_h = \lambda \omega_h + (1 - \lambda) \omega_h'$$

implies $\omega_h = \omega_h' = 0$. Hence $\omega = \omega'$, proving uniqueness. $\qquad \square$

## A.9. Proof of Lemma 4.1

*Proof.* Consider the dual optimization problem stated in Theorem 3.5:

$$\min_{\lambda} \mathcal{Q}(\lambda, \mathcal{P}) = - \sum_{h \in [D]} \sqrt{\sum_{i \in [K], j \in [M]} \lambda_{ij} \chi_h(x_i, c_j)}$$

$$\text{s.t.} \quad \sum_{i \in [K], j \in [M]} \lambda_{ij} = 1, \quad (\phi) \tag{94}$$

$$\lambda_{ij} \geq 0, \quad \forall i \in [K], j \in [M]. \quad (v_{ij})$$

For any feasible solution $\lambda$, the set of all feasible directions at $\lambda$ is defined by:

$$\mathcal{F}(\lambda) = \left\{ d \in \mathbb{R}^{KM} : \sum_{j \in [M], i \in [K]} d_{ij} = 0, \, d_{ij} \geq 0, \text{if } \lambda_{ij} = 0 \right\}. \tag{95}$$

The Lagrangian function for this problem is:

$$L(\lambda, \phi, v) = - \sum_{h \in [D]} \sqrt{\sum_{i \in [K], j \in [M]} \lambda_{ij} \chi_h(x_i, c_j)} + \phi \left( \sum_{i \in [K], j \in [M]} \lambda_{ij} - 1 \right) - \sum_{i \in [K], j \in [M]} v_{ij} \lambda_{ij}. \tag{96}$$

We consider feasible points at which the denominator in the gradient expression is strictly positive for every $h \in [D]$. For such a feasible point $\lambda$, the KKT stationarity conditions require the existence of multipliers $\phi$ and $v_{ij} \geq 0$ such that

$$-\frac{1}{2} \sum_{h \in [D]} \frac{\chi_h(x_i, c_j)}{\sqrt{\sum_{a \in [K], b \in [M]} \lambda_{ab} \chi_h(x_a, c_b)}} + \phi - v_{ij} = 0, \quad \forall i \in [K], j \in [M],$$

$$v_{ij} \lambda_{ij} = 0, \quad \forall i \in [K], j \in [M],$$

$$\lambda_{ij} \geq 0, \quad \forall i \in [K], j \in [M], \tag{97}$$

$$\sum_{i \in [K], j \in [M]} \lambda_{ij} = 1,$$

$$v_{ij} \geq 0, \quad \forall i \in [K], j \in [M].$$

From these KKT conditions, a feasible solution $\lambda$ is a stationary point if and only if there exists a scalar $\phi$ such that, if $\lambda_{ij} = 0$, then

$$\phi \geq \frac{1}{2} \sum_{h \in [D]} \frac{\chi_h(x_i, c_j)}{\sqrt{\sum_{a \in [K], b \in [M]} \lambda_{ab} \chi_h(x_a, c_b)}}, \tag{98}$$

and if $\lambda_{ij} > 0$, then

$$\phi = \frac{1}{2} \sum_{h \in [D]} \frac{\chi_h(x_i, c_j)}{\sqrt{\sum_{a \in [K], b \in [M]} \lambda_{ab} \chi_h(x_a, c_b)}}. \tag{99}$$

This implies that a feasible solution $\lambda$ is a stationary point of problem (19) if and only if:

$$-\frac{1}{2} \sum_{h \in [D]} \frac{\chi_h(x_i, c_j)}{\sqrt{\sum_{a \in [K], b \in [M]} \lambda_{ab} \chi_h(x_a, c_b)}} \geq -\frac{1}{2} \sum_{h \in [D]} \frac{\chi_h(x_{i'}, c_{j'})}{\sqrt{\sum_{a \in [K], b \in [M]} \lambda_{ab} \chi_h(x_a, c_b)}}, \tag{100}$$

for any $(i, j) \in [K] \times [M]$ and any $(i', j') \in [K] \times [M]$ such that $\lambda_{i'j'} > 0$. Now, fix a feasible solution $\lambda$ with $\lambda_{mn} > 0$. Define the reduced set:

$$\mathcal{D}^{m,n}(\lambda) = \left\{ e_{ij} - e_{mn} : i \neq m \text{ or } j \neq n \right\} \bigcup \left\{ e_{mn} - e_{ij} : i \neq m \text{ or } j \neq n, \lambda_{ij} > 0 \right\}, \tag{101}$$

where $e_{ij} \in \mathbb{R}^{KM}$ is the vector whose $(i, j)$-th component is one, and all other components are zero.

According to Proposition 3.4 of Lin et al. (2009), we have:

$$\mathcal{D}^{m,n}(\lambda) \subseteq \mathcal{F}(\lambda), \qquad \text{cone}\left(\mathcal{D}^{m,n}(\lambda)\right) = \mathcal{F}(\lambda). \tag{102}$$

Therefore, if $\lambda$ is a stationary point, then

$$\nabla \mathcal{Q}(\lambda, \mathcal{P})^\top d \geq 0, \quad \forall d \in \mathcal{F}(\lambda),$$

and hence the same inequality holds for all $d \in \mathcal{D}^{m,n}(\lambda)$.

Conversely, if

$$\nabla \mathcal{Q}(\lambda, \mathcal{P})^\top d \geq 0, \quad \forall d \in \mathcal{D}^{m,n}(\lambda),$$

then, since every feasible direction in $\mathcal{F}(\lambda)$ is a nonnegative linear combination of directions in $\mathcal{D}^{m,n}(\lambda)$, the inequality also holds for all $d \in \mathcal{F}(\lambda)$. Hence, $\lambda$ satisfies the first-order stationarity condition for problem (19). $\qquad \square$

## A.10. Proof of Theorem 4.2

The proof of Theorem 4.2 relies on several auxiliary lemmas. Lemma A.2 establishes the necessary continuity arguments. Lemma A.3 proves the $\delta$-PAC property of the proposed algorithm. Lemmas A.4 and A.5 present known results from the existing literature. Lemma A.6 establishes the convergence of the gradient descent procedures in Algorithm 2. Finally, we derive upper bounds, both almost surely and in expectation, for the stopping time $\tau$.

**Lemma A.2.** *Let* $\mathcal{U}(\omega, \mathcal{P})^{-1} = \min_{c_j \in \mathcal{C}} \Gamma^s(\omega, c_j, \mathcal{P})$ *denote the objective function of problem (18). Then,* $\mathcal{U}(\omega, \mathcal{P})^{-1}$ *is a continuous function with respect to both* $\omega$ *and* $\mathcal{P}$. *Moreover, the optimal sampling ratio* $\omega^*$ *satisfies* $\omega_h^* > 0$ *for all* $h \in [D]$.

*Proof.* We first prove continuity. Recall that the surrogate objective is given by

$$\Gamma^s(\omega, c_j, \mathcal{P}) = \frac{1}{2} \min_{x_i \in \mathcal{X}} \left( \frac{\left((\phi(x_{i^*(c_j)}, c_j) - \phi(x_i, c_j))^\top \theta\right)^2}{\left\|\phi(x_{i^*(c_j)}, c_j) - \phi(x_i, c_j)\right\|_{\Lambda(\omega)^{-1}}^2} \mathbb{I}\left(x_i \in \mathcal{D}_1(c_j)\right) \right. \tag{103}$$
$$\left. + \frac{\left(b - \beta^\top \phi(x_i, c_j)\right)^2}{\left\|\phi(x_i, c_j)\right\|_{\Lambda(\omega)^{-1}}^2} \mathbb{I}\left(x_i \in \{x_{i^*(c_j)}\} \cup \mathcal{D}_2(c_j) \cup \mathcal{D}_3(c_j)\right) \right).$$

Here, when $\Lambda(\omega)$ is singular, $\|\cdot\|_{\Lambda(\omega)^{-1}}$ is interpreted in the extended sense, with uncovered directions assigned infinite variance.

Because the $D$ design points span $\mathbb{R}^D$, the corresponding design matrix $\Phi$ is invertible. Thus, for any vector $v \in \mathbb{R}^D$,

$$\|v\|_{\Lambda(\omega)^{-1}}^2 = \sum_{h \in [D]} \frac{\sigma_h^2 \left[(\Phi^\top)^{-1} v\right]_h^2}{\omega_h}, \tag{104}$$

where the expression is understood in the extended sense when some $\omega_h = 0$. Therefore, each term appearing in $\Gamma^s(\omega, c_j, \mathcal{P})$ is continuous in $\omega$ under this convention.

Next, by Assumption 2.1, for each covariate $c_j$, the best feasible arm is unique. Since no arm lies on the constraint boundary and the arm set is finite, the feasible/infeasible classification remains unchanged in a sufficiently small neighborhood of the true instance $\mathcal{P}$. Moreover, the identity of the best feasible arm also remains unchanged. The possible switching between $\mathcal{D}_2(c_j)$ and $\mathcal{D}_3(c_j)$ does not affect $\Gamma^s$, because these two sets are merged in the surrogate objective. Within this neighborhood, the numerators in the above display are continuous functions of $\theta$ and $\beta$, and the denominators are continuous functions of $\omega$. Since $\Gamma^s(\omega, c_j, \mathcal{P})$ is the minimum of finitely many continuous functions, it is continuous in $(\omega, \mathcal{P})$ for each $c_j \in \mathcal{C}$. Taking the minimum over the finite set $\mathcal{C}$ preserves continuity. Therefore,

$$\mathcal{U}(\omega, \mathcal{P})^{-1} = \min_{c_j \in \mathcal{C}} \Gamma^s(\omega, c_j, \mathcal{P})$$

is continuous in $(\omega, \mathcal{P})$.

We now prove that the optimal sampling ratio is strictly positive. We exclude degenerate cases in which a design direction is irrelevant to all surrogate comparisons, as such cases can be removed from the design set without affecting the identification problem. Let $\omega^* \in \Omega$ be an optimal solution of problem (18). Suppose, for contradiction, that there exists $h \in [D]$ such that $\omega_h^* = 0$. Since we use exactly $D$ linearly independent design points to estimate the $D$-dimensional parameters, setting $\omega_h^* = 0$ leaves one design direction unobserved and makes $\Lambda(\omega^*)$ singular. Under the extended interpretation of $\Lambda(\omega)^{-1}$, this implies that the variance in the uncovered direction is infinite, and hence the corresponding information rate is zero. Therefore,

$$\mathcal{U}(\omega^*, \mathcal{P})^{-1} = \min_{c_j \in \mathcal{C}} \Gamma^s(\omega^*, c_j, \mathcal{P}) = 0.$$

On the other hand, consider the uniform sampling ratio $\bar{\omega}_h = 1/D$ for all $h \in [D]$. Then $\Lambda(\bar{\omega})$ is positive definite. Moreover, by Assumption 2.1, all relevant optimality and feasibility gaps are strictly positive. Hence,

$$\mathcal{U}(\bar{\omega}, \mathcal{P})^{-1} = \min_{c_j \in \mathcal{C}} \Gamma^s(\bar{\omega}, c_j, \mathcal{P}) > 0.$$

This contradicts the optimality of $\omega^*$. Therefore, it must hold that $\omega_h^* > 0, \qquad \forall h \in [D]$. $\qquad \square$

**Lemma A.3.** *The duality-based decomposition algorithm is $\delta$-PAC.*

*Proof.* The stopping rule of the duality-based decomposition algorithm is

$$\tau = \inf \left\{ t \in \mathbb{N} : t\mathcal{U}(\omega(t), \hat{\mathcal{P}}(t))^{-1} > \rho(t, \delta) \right\}, \tag{105}$$

where

$$\mathcal{U}(\omega(t), \hat{\mathcal{P}}(t))^{-1} = \min_{c_j \in \mathcal{C}} \Gamma^s(\omega(t), c_j, \hat{\mathcal{P}}(t)). \tag{106}$$

By construction, the stopping rule is evaluated only at times when the estimated feasible set is nonempty for every covariate. Hence, the recommendation rule and the statistic $\mathcal{U}(\omega(t), \hat{\mathcal{P}}(t))^{-1}$ are well defined whenever stopping can occur.

To establish the $\delta$-PAC property, it suffices to show that

$$\mathbb{P}\left( \tau < \infty, \exists c_j \in \mathcal{C}, x_{\hat{i}(c_j; \tau)} \neq x_{i^*(c_j)} \right) \leq \delta. \tag{107}$$

We begin by noting that, by the definition of the surrogate objective and Lemma 3.4, for every $\omega$ and every problem instance $\mathcal{P}'$,

$$\mathcal{U}(\omega, \mathcal{P}')^{-1} \leq \inf_{\tilde{\mathcal{P}} \in \mathcal{A}(\mathcal{P}')} \mathcal{H}(\omega, \mathcal{P}', \tilde{\mathcal{P}})^{-1},$$

where $\mathcal{A}(\mathcal{P}')$ denotes the exact alternative set under which the recommended best arm differs from that under $\mathcal{P}'$.

Therefore,

$$\mathbb{P}\left( \tau < \infty, \exists c_j \in \mathcal{C}, x_{\hat{i}(c_j; \tau)} \neq x_{i^*(c_j)} \right)$$

$$\leq \mathbb{P}\left( \exists t \in \mathbb{N}, \exists c_j \in \mathcal{C}, x_{\hat{i}(c_j; t)} \neq x_{i^*(c_j)}, t\mathcal{U}(\omega(t), \hat{\mathcal{P}}(t))^{-1} \geq \rho(t, \delta) \right) \tag{108}$$

$$\leq \mathbb{P}\left( \exists t \in \mathbb{N}, \exists c_j \in \mathcal{C}, x_{\hat{i}(c_j; t)} \neq x_{i^*(c_j)}, \inf_{\tilde{\mathcal{P}} \in \mathcal{A}(\hat{\mathcal{P}}(t))} t\mathcal{H}(\omega(t), \hat{\mathcal{P}}(t), \tilde{\mathcal{P}})^{-1} \geq \rho(t, \delta) \right).$$

On the event that the recommendation based on $\hat{\mathcal{P}}(t)$ is incorrect for some covariate, the true instance $\mathcal{P}$ belongs to the alternative set $\mathcal{A}(\hat{\mathcal{P}}(t))$. Therefore,

$$\mathbb{P}\left( \exists t \in \mathbb{N}, \exists c_j \in \mathcal{C}, x_{\hat{i}(c_j; t)} \neq x_{i^*(c_j)}, \inf_{\tilde{\mathcal{P}} \in \mathcal{A}(\hat{\mathcal{P}}(t))} t\mathcal{H}(\omega(t), \hat{\mathcal{P}}(t), \tilde{\mathcal{P}})^{-1} \geq \rho(t, \delta) \right)$$

$$\leq \mathbb{P}\left( \exists t \in \mathbb{N}, t\mathcal{H}(\omega(t), \hat{\mathcal{P}}(t), \mathcal{P})^{-1} \geq \rho(t, \delta) \right). \tag{109}$$

Since $\omega_h(t) = N_h(t)/t$, we have

$$
\begin{aligned}
&t\mathcal{H}(\omega(t), \hat{\mathcal{P}}(t), \mathcal{P})^{-1} \\
&= \sum_{h \in [D]} N_h(t)\big(d(\bar{F}(z_h; t), f(z_h)) + d(\bar{G}(z_h; t), g(z_h))\big).
\end{aligned}
\tag{110}
$$

Hence,

$$
\begin{aligned}
&\mathbb{P}\bigg(\exists t \in \mathbb{N}, t\mathcal{H}(\omega(t), \hat{\mathcal{P}}(t), \mathcal{P})^{-1} \geq \rho(t, \delta)\bigg) \\
&= \mathbb{P}\bigg(\exists t \in \mathbb{N}, \sum_{h \in [D]} N_h(t)\big(d(\bar{F}(z_h; t), f(z_h)) + d(\bar{G}(z_h; t), g(z_h))\big) \geq \rho(t, \delta)\bigg).
\end{aligned}
\tag{111}
$$

Under Assumption 2.3, applying the time-uniform concentration inequality in Proposition 12 of Garivier & Kaufmann (2016) to the joint collection of the $2D$ empirical statistics

$$
\{\bar{F}(z_h; t), \bar{G}(z_h; t) : h \in [D]\},
$$

we obtain

$$
\begin{aligned}
&\mathbb{P}\bigg(\exists t \in \mathbb{N}, \sum_{h \in [D]} N_h(t)\big(d(\bar{F}(z_h; t), f(z_h)) + d(\bar{G}(z_h; t), g(z_h))\big) \geq \rho(t, \delta)\bigg) \\
&\leq \sum_{t=1}^{\infty} e^{2D+1}\left(\frac{\rho(t, \delta)^2 \log t}{2D}\right)^{2D} e^{-\rho(t, \delta)}.
\end{aligned}
\tag{112}
$$

Let $\alpha > 1$ and choose

$$
\rho(t, \delta) = \log\left(\frac{Ct^\alpha \log(1/\delta)^{4D+1}}{\delta}\right),
\tag{113}
$$

for sufficiently small $\delta$. Therefore,

$$
\begin{aligned}
&\sum_{t=1}^{\infty} e^{2D+1}\left(\frac{\rho(t, \delta)^2 \log t}{2D}\right)^{2D} e^{-\rho(t, \delta)} \\
&= \delta \sum_{t=1}^{\infty} \frac{e^{2D+1}}{Ct^\alpha \log(1/\delta)^{4D+1}} \left(\frac{[\log(Ct^\alpha) + (4D+1)\log\log(1/\delta) + \log(1/\delta)]^2 \log t}{2D}\right)^{2D}.
\end{aligned}
\tag{114}
$$

We can choose a sufficiently large constant $C > 0$, independent of $\delta$, such that

$$
\sum_{t=1}^{\infty} \frac{e^{2D+1}}{Ct^\alpha \log(1/\delta)^{4D+1}} \left(\frac{[\log(Ct^\alpha) + (4D+1)\log\log(1/\delta) + \log(1/\delta)]^2 \log t}{2D}\right)^{2D} \leq 1.
\tag{115}
$$

Consequently,

$$
\sum_{t=1}^{\infty} e^{2D+1}\left(\frac{\rho(t, \delta)^2 \log t}{2D}\right)^{2D} e^{-\rho(t, \delta)} \leq \delta.
\tag{116}
$$

Therefore,

$$
\mathbb{P}\bigg(\tau < \infty, \exists c_j \in \mathcal{C}, x_{\hat{i}(c_j; \tau)} \neq x_{i^*(c_j)}\bigg) \leq \delta.
\tag{117}
$$

This proves that the duality-based decomposition algorithm is $\delta$-PAC. $\qquad\square$

The convergence analysis of the duality-based decomposition algorithm relies on a line search procedure to determine the step size. For completeness, we include the canonical line search algorithm along with its associated theoretical results.

---

**Algorithm 3** Line Search Algorithm

---

**Input:** Descent direction $d$, maximum feasible step size $s^{max}$, the current feasible solution $\lambda$, problem instance $\mathcal{P}$, parameter $\alpha, \nu \in (0, 1)$.

Set $s = s^{max}$

**while** $\mathcal{Q}(\lambda + sd, \mathcal{P}) > \mathcal{Q}(\lambda, \mathcal{P}) + \alpha s \nabla \mathcal{Q}(\lambda, \mathcal{P})^\top d$ **do**
∟ $s \leftarrow \nu s$

**return** the step size $s$.

---

**Lemma A.4** (Proposition 4.1 in Lin et al. (2009)). *Define a subsequence $\mathcal{T} \subset \{1, 2 \ldots\}$ such that the line search algorithm is invoked at time steps $t \in \mathcal{T}$. Let $\{\lambda(t)\}_{t \in \mathcal{T}}$ denote the corresponding sequence of solutions, and let $\{d(t)\}_{t \in \mathcal{T}}$ denote the associated descent directions. Then, the line search algorithm terminates in a finite number of iterations, producing a step size $s(t)$ that satisfies*

$$\mathcal{Q}(\lambda(t-1) + s(t)d(t), \hat{\mathcal{P}}(t)) \leq \mathcal{Q}(\lambda(t-1), \hat{\mathcal{P}}(t)) + \alpha s(t) \nabla \mathcal{Q}(\lambda(t-1), \hat{\mathcal{P}}(t))^\top d(t). \tag{118}$$

*Furthermore, suppose that $\lim_{t \to \infty} \lambda(t) = \bar{\lambda}$, and*

$$\lim_{t \to \infty} \mathcal{Q}(\lambda(t-1), \mathcal{P}) - \mathcal{Q}(\lambda(t-1) + s(t)d(t), \mathcal{P}) = 0. \tag{119}$$

*Then, it follows that*

$$\lim_{t \to \infty, t \in \mathcal{T}} s^{\max}(d(t), \lambda(t-1)) \nabla \mathcal{Q}(\lambda(t-1), \mathcal{P})^\top d(t) = 0, \tag{120}$$

*where $s^{\max}(d(t), \lambda(t-1))$ denotes the maximum feasible step size along direction $d(t)$ from $\lambda(t-1)$.*

**Lemma A.5** (Lemma 17 in Garivier & Kaufmann (2016)). *Consider the following sampling rule*

$$z_{h(t+1)} = \begin{cases} \arg\min_{z_h \in \mathcal{B}_t} N_h(t) & if \quad \mathcal{B}_t \neq \emptyset \\ \arg\min_{z_h \in \mathcal{Z}} N_h(t) - t\gamma_h(\hat{\mathcal{P}}(t)) & otherwise \end{cases}, \tag{121}$$

*where $\mathcal{B}_t = \{z_h \in \mathcal{Z} : N_h(t) < \sqrt{t} - D/2\}$. Then, for every design point $z_h \in \mathcal{Z}$, we have $N_h(t) \geq (\sqrt{t} - D/2)_+ - 1$. Furthermore, for any $\epsilon > 0$ and $t_0 > 0$ such that*

$$\sup_{t \geq t_0} \max_{h \in [D]} \left| \gamma_h(\hat{\mathcal{P}}(t)) - \omega_h^*(\mathcal{P}) \right| \leq \epsilon, \tag{122}$$

*there exists $t_1 > 0$ such that*

$$\sup_{t \geq t_1} \max_{h \in [D]} \left| \frac{N_h(t)}{t} - \omega_h^*(\mathcal{P}) \right| \leq 3(D-1)\epsilon. \tag{123}$$

The following lemma establishes the convergence of the gradient descent procedure in Algorithm 2. The analysis follows the proof of Proposition 6.1 in Lin et al. (2009) and Theorem 5 in Zhou et al. (2024).

**Lemma A.6.** *Let $\{\lambda(t)\}$ be the sequence generated by the duality-based algorithm. Then every limit point of this sequence is a stationary point of the dual optimization problem (19).*

*Proof.* For the convergence analysis, we ignore the fixed tolerance $\kappa_0$ and use only the vanishing thresholds in Algorithm 2. The constant $\kappa_0$ is introduced only as a small numerical tolerance in practical implementation.

According to Lemma A.5, the sampling rule of the duality-based decomposition algorithm guarantees that

$$N_h(t) \geq (\sqrt{t} - D/2)_+ - 1. \tag{124}$$

This lower bound implies that the number of samples allocated to each design point grows unbounded as $t \to \infty$. Consequently, by the strong law of large numbers, the estimators converge almost surely:

$$\hat{\theta}(t) \to \theta, \hat{\beta}(t) \to \beta \text{ and } \hat{\mathcal{P}}(t) \to \mathcal{P} \tag{125}$$

By Assumption 2.1, the uniqueness of the best arm and the strict feasibility/infeasibility margins imply that the estimated best arm $x_{\hat{i}(c_j;t)}$ converges almost surely to the true best arm $x_{i^*(c_j)}$ for all $c_j \in \mathcal{C}$. This establishes the consistency of the proposed duality-based decomposition algorithm.

We now establish useful continuity properties of the objective function $\mathcal{Q}(\lambda, \mathcal{P})$ and its gradient $\nabla \mathcal{Q}(\lambda, \mathcal{P})$. Recall that

$$\mathcal{Q}(\lambda, \mathcal{P}) = - \sum_{h \in [D]} \sqrt{\sum_{i \in [K], j \in [M]} \lambda_{ij} \chi_h(x_i, c_j, \mathcal{P})}. \tag{126}$$

At any feasible $\lambda$ such that

$$\sum_{a \in [K], b \in [M]} \lambda_{ab} \chi_h(x_a, c_b, \mathcal{P}) > 0, \qquad \forall h \in [D],$$

the gradient is given by, for each $i \in [K]$ and $j \in [M]$,

$$[\nabla \mathcal{Q}(\lambda, \mathcal{P})]_{ij} = - \sum_{h \in [D]} \frac{\chi_h(x_i, c_j, \mathcal{P})}{2 \sqrt{\sum_{a \in [K], b \in [M]} \lambda_{ab} \chi_h(x_a, c_b, \mathcal{P})}}. \tag{127}$$

It is straightforward to verify that $\mathcal{Q}(\lambda, \mathcal{P})$ is continuous in $\lambda$ on the region where it is differentiable. We now show that it is also continuous in $\mathcal{P}$. Since $\hat{\mathcal{P}}(t) \to \mathcal{P}$ and, by the definition of $\chi_h(x_i, c_j, \mathcal{P})$, the relevant optimality and feasibility gaps are bounded away from zero in a neighborhood of $\mathcal{P}$, there exists a constant $L > 0$ such that, for sufficiently large $t$,

$$\left| \chi_h(x_i, c_j, \mathcal{P}) - \chi_h(x_i, c_j, \hat{\mathcal{P}}(t)) \right| \leq L \left\| \mathcal{P} - \hat{\mathcal{P}}(t) \right\|_\infty, \tag{128}$$

for all $h \in [D]$, $i \in [K]$, and $j \in [M]$.

Fix a feasible $\lambda$ such that

$$\sum_{i \in [K], j \in [M]} \lambda_{ij} \chi_h(x_i, c_j, \mathcal{P}) > 0, \qquad \forall h \in [D].$$

Then, for sufficiently large $t$, there exists a constant $C_\lambda > 0$ such that

$$\sum_{i \in [K], j \in [M]} \lambda_{ij} \chi_h(x_i, c_j, \mathcal{P}) \geq C_\lambda, \qquad \sum_{i \in [K], j \in [M]} \lambda_{ij} \chi_h(x_i, c_j, \hat{\mathcal{P}}(t)) \geq C_\lambda, \qquad \forall h \in [D].$$

Therefore,

$$
\begin{aligned}
&\left| \mathcal{Q}(\lambda, \mathcal{P}) - \mathcal{Q}(\lambda, \hat{\mathcal{P}}(t)) \right| \\
&= \left| \sum_{h \in [D]} \sqrt{\sum_{i \in [K], j \in [M]} \lambda_{ij} \chi_h(x_i, c_j, \mathcal{P})} - \sum_{h \in [D]} \sqrt{\sum_{i \in [K], j \in [M]} \lambda_{ij} \chi_h(x_i, c_j, \hat{\mathcal{P}}(t))} \right| \\
&\leq \sum_{h \in [D]} \frac{\left| \sum_{i \in [K], j \in [M]} \lambda_{ij} \left( \chi_h(x_i, c_j, \mathcal{P}) - \chi_h(x_i, c_j, \hat{\mathcal{P}}(t)) \right) \right|}{\sqrt{\sum_{i \in [K], j \in [M]} \lambda_{ij} \chi_h(x_i, c_j, \mathcal{P})} + \sqrt{\sum_{i \in [K], j \in [M]} \lambda_{ij} \chi_h(x_i, c_j, \hat{\mathcal{P}}(t))}} \\
&\leq \sum_{h \in [D]} \frac{\sum_{i \in [K], j \in [M]} \lambda_{ij} \left| \chi_h(x_i, c_j, \mathcal{P}) - \chi_h(x_i, c_j, \hat{\mathcal{P}}(t)) \right|}{2 \sqrt{C_\lambda}} \\
&\leq \frac{DL}{2 \sqrt{C_\lambda}} \left\| \hat{\mathcal{P}}(t) - \mathcal{P} \right\|_\infty \\
&\triangleq \bar{C}_\lambda \left\| \hat{\mathcal{P}}(t) - \mathcal{P} \right\|_\infty.
\end{aligned}
\tag{129}
$$

This proves the continuity of $\mathcal{Q}(\lambda, \mathcal{P})$ with respect to $\mathcal{P}$ at every feasible $\lambda$ where the denominator terms are strictly positive.

We next show that $\nabla \mathcal{Q}(\lambda, \hat{\mathcal{P}}(t))$ is continuous in $\lambda$. Following the approach of Theorem 5 in Zhou et al. (2024), it holds that

$$\liminf_{t \to \infty} \sum_{a \in [K], b \in [M]} \lambda_{ab}(t) \chi_h(x_a, c_b, \hat{\mathcal{P}}(t)) > 0, \qquad \forall h \in [D]. \tag{130}$$

Therefore, for sufficiently large $t$, there exists a constant $C_{\min} > 0$ such that

$$\sum_{a \in [K], \, b \in [M]} \lambda_{ab} \chi_h(x_a, c_b, \hat{\mathcal{P}}(t)) \geq C_{\min}, \qquad \sum_{a \in [K], \, b \in [M]} \lambda'_{ab} \chi_h(x_a, c_b, \hat{\mathcal{P}}(t)) \geq C_{\min}, \qquad \forall h \in [D], \qquad (131)$$

for the feasible points $\lambda$ and $\lambda'$ under consideration. Then,

$$
\begin{aligned}
& \left| [\nabla \mathcal{Q}(\lambda, \hat{\mathcal{P}}(t))]_{ij} - [\nabla \mathcal{Q}(\lambda', \hat{\mathcal{P}}(t))]_{ij} \right| \\
&= \left| \sum_{h \in [D]} \frac{\chi_h(x_i, c_j, \hat{\mathcal{P}}(t))}{2\sqrt{\sum_{a \in [K], \, b \in [M]} \lambda_{ab} \chi_h(x_a, c_b, \hat{\mathcal{P}}(t))}} - \sum_{h \in [D]} \frac{\chi_h(x_i, c_j, \hat{\mathcal{P}}(t))}{2\sqrt{\sum_{a \in [K], \, b \in [M]} \lambda'_{ab} \chi_h(x_a, c_b, \hat{\mathcal{P}}(t))}} \right| \\
&\leq \sum_{h \in [D]} \frac{\chi_h(x_i, c_j, \hat{\mathcal{P}}(t))}{2} \left| \frac{1}{\sqrt{\sum_{a \in [K], \, b \in [M]} \lambda_{ab} \chi_h(x_a, c_b, \hat{\mathcal{P}}(t))}} - \frac{1}{\sqrt{\sum_{a \in [K], \, b \in [M]} \lambda'_{ab} \chi_h(x_a, c_b, \hat{\mathcal{P}}(t))}} \right| \qquad (132) \\
&\leq \sum_{h \in [D]} \frac{C_1}{4 C_{\min}^{3/2}} \left| \sum_{a \in [K], \, b \in [M]} (\lambda'_{ab} - \lambda_{ab}) \chi_h(x_a, c_b, \hat{\mathcal{P}}(t)) \right| \\
&\leq \frac{DKM C_1^2}{4 C_{\min}^{3/2}} \|\lambda' - \lambda\|_\infty \\
&\triangleq \tilde{C} \|\lambda' - \lambda\|_\infty,
\end{aligned}
$$

where $C_1 = \max_{i \in [K], \, j \in [M], \, h \in [D]} \sup_t \chi_h(x_i, c_j, \hat{\mathcal{P}}(t)) < \infty$, and $\tilde{C} = \frac{DKM C_1^2}{4 C_{\min}^{3/2}}$.

Finally, we show that $\nabla \mathcal{Q}(\lambda, \mathcal{P})$ is continuous with respect to $\mathcal{P}$. For each $i \in [K]$ and $j \in [M]$, we have

$$
\begin{aligned}
& \left| [\nabla \mathcal{Q}(\lambda, \hat{\mathcal{P}}(t))]_{ij} - [\nabla \mathcal{Q}(\lambda, \mathcal{P})]_{ij} \right| \\
&= \left| \sum_{h \in [D]} \frac{\chi_h(x_i, c_j, \hat{\mathcal{P}}(t))}{2\sqrt{\sum_{a \in [K], \, b \in [M]} \lambda_{ab} \chi_h(x_a, c_b, \hat{\mathcal{P}}(t))}} - \sum_{h \in [D]} \frac{\chi_h(x_i, c_j, \mathcal{P})}{2\sqrt{\sum_{a \in [K], \, b \in [M]} \lambda_{ab} \chi_h(x_a, c_b, \mathcal{P})}} \right| \\
&\leq \frac{1}{2} \sum_{h \in [D]} \left| \frac{\chi_h(x_i, c_j, \hat{\mathcal{P}}(t)) - \chi_h(x_i, c_j, \mathcal{P})}{\sqrt{\sum_{a \in [K], \, b \in [M]} \lambda_{ab} \chi_h(x_a, c_b, \hat{\mathcal{P}}(t))}} \right| \\
&\quad + \frac{1}{2} \sum_{h \in [D]} \chi_h(x_i, c_j, \mathcal{P}) \left| \frac{1}{\sqrt{\sum_{a \in [K], \, b \in [M]} \lambda_{ab} \chi_h(x_a, c_b, \hat{\mathcal{P}}(t))}} - \frac{1}{\sqrt{\sum_{a \in [K], \, b \in [M]} \lambda_{ab} \chi_h(x_a, c_b, \mathcal{P})}} \right| \qquad (133) \\
&\leq \frac{1}{2} \sum_{h \in [D]} \frac{L \|\hat{\mathcal{P}}(t) - \mathcal{P}\|_\infty}{\sqrt{C_{\min}}} \\
&\quad + \frac{1}{2} \sum_{h \in [D]} C_1 \frac{\left| \sum_{a \in [K], \, b \in [M]} \lambda_{ab} \left( \chi_h(x_a, c_b, \hat{\mathcal{P}}(t)) - \chi_h(x_a, c_b, \mathcal{P}) \right) \right|}{2 C_{\min}^{3/2}} \\
&\leq \frac{DL}{2\sqrt{C_{\min}}} \|\hat{\mathcal{P}}(t) - \mathcal{P}\|_\infty + \frac{DC_1 L}{4 C_{\min}^{3/2}} \|\hat{\mathcal{P}}(t) - \mathcal{P}\|_\infty \\
&= D \left( \frac{L}{2\sqrt{C_{\min}}} + \frac{C_1 L}{4 C_{\min}^{3/2}} \right) \|\hat{\mathcal{P}}(t) - \mathcal{P}\|_\infty.
\end{aligned}
$$

Thus, $\nabla \mathcal{Q}(\lambda, \mathcal{P})$ is continuous with respect to $\mathcal{P}$ on any region where the denominator terms are uniformly bounded away from zero.

Let $\mathcal{T}$ denote the set of time steps at which the line search is invoked. If $\mathcal{T}$ is finite, then $\lambda(t)$ is eventually constant. If its limiting value were not stationary, then, because the update thresholds are vanishing and $\hat{\mathcal{P}}(t) \to \mathcal{P}$, Algorithm 2 would

eventually detect a feasible descent direction and invoke the line search again, a contradiction. Hence, in this case, the limiting value is stationary.

It remains to consider the case where $\mathcal{T}$ is infinite. Let $\bar{\lambda}$ be a limit point of the sequence $\{\lambda(t)\}$. Then, by definition, there exists a subsequence $\mathcal{T}_1 \subset \mathcal{T}$ such that

$$\lim_{t \to \infty, t \in \mathcal{T}_1} \lambda(t-1) = \bar{\lambda}. \tag{134}$$

Since the index pair $(m(t), n(t)) \in [K] \times [M]$ takes values from a finite set, we can further extract a subsequence $\mathcal{T}_2 \subseteq \mathcal{T}_1$ and a fixed index pair $(m, n) \in [K] \times [M]$ such that, for all $t \in \mathcal{T}_2$,

$$(m(t), n(t)) = (m, n), \qquad \lambda_{mn}(t-1) \geq \eta. \tag{135}$$

Therefore, $\bar{\lambda}_{mn} \geq \eta > 0$. Define the feasible direction cone at $\bar{\lambda}$ by

$$\mathcal{F}(\bar{\lambda}) = \left\{ d \in \mathbb{R}^{KM} : \sum_{i \in [K], \, j \in [M]} d_{ij} = 0, \ d_{ij} \geq 0 \text{ if } \bar{\lambda}_{ij} = 0 \right\}. \tag{136}$$

By Proposition 3.4 of Lin et al. (2009), since $\bar{\lambda}_{mn} > 0$, we have

$$\text{cone}\left( \mathcal{D}^{m,n}(\bar{\lambda}) \right) = \mathcal{F}(\bar{\lambda}). \tag{137}$$

We proceed by contradiction. Suppose that $\bar{\lambda}$ is not a stationary point of the dual optimization problem (19). Then, by Lemma 4.1, there exists a direction $\bar{d} \in \mathcal{D}^{m,n}(\bar{\lambda})$ such that

$$\nabla \mathcal{Q}(\bar{\lambda}, \mathcal{P})^\top \bar{d} < 0. \tag{138}$$

Hence, there exists $\varepsilon_0 > 0$ such that

$$\nabla \mathcal{Q}(\bar{\lambda}, \mathcal{P})^\top \bar{d} \leq -4\varepsilon_0. \tag{139}$$

Since $\lambda(t-1) \to \bar{\lambda}$ along $\mathcal{T}_2$, and since $\bar{d} \in \mathcal{D}^{m,n}(\bar{\lambda})$, by passing to a further subsequence if necessary, we may assume that the sign pattern of the coordinates involved in $\bar{d}$ is stable. Hence, for all sufficiently large $t \in \mathcal{T}_2$,

$$\bar{d} \in \mathcal{D}^{m,n}(\lambda(t-1)). \tag{140}$$

Moreover, since $\hat{\mathcal{P}}(t) \to \mathcal{P}$ almost surely and $\nabla \mathcal{Q}(\lambda, \mathcal{P})$ is continuous in its arguments, for all sufficiently large $t \in \mathcal{T}_2$,

$$\nabla \mathcal{Q}(\lambda(t-1), \hat{\mathcal{P}}(t))^\top \bar{d} \leq -3\varepsilon_0. \tag{141}$$

By Proposition A.1 of Lin et al. (2009), there exists a constant $c > 0$ such that, for all sufficiently large $t \in \mathcal{T}_2$,

$$s^{\max}(\bar{d}, \lambda(t-1)) \geq c. \tag{142}$$

Therefore,

$$s^{\max}(\bar{d}, \lambda(t-1)) \nabla \mathcal{Q}(\lambda(t-1), \hat{\mathcal{P}}(t))^\top \bar{d} \leq -3c\varepsilon_0. \tag{143}$$

From the definition of the duality-based decomposition algorithm, $d(t)$ minimizes

$$s^{\max}(d, \lambda(t-1)) \nabla \mathcal{Q}(\lambda(t-1), \hat{\mathcal{P}}(t))^\top d$$

over $d \in \mathcal{D}^{m,n}(\lambda(t-1))$. Therefore, by (143), for all sufficiently large $t \in \mathcal{T}_2$,

$$s^{\max}(d(t), \lambda(t-1)) \nabla \mathcal{Q}(\lambda(t-1), \hat{\mathcal{P}}(t))^\top d(t) \leq -3c\varepsilon_0. \tag{144}$$

Again, by the continuity of $\nabla \mathcal{Q}$ and the finiteness of the reduced direction set, we also have, for all sufficiently large $t \in \mathcal{T}_2$,

$$s^{\max}(d(t), \lambda(t-1)) \nabla \mathcal{Q}(\lambda(t-1), \mathcal{P})^\top d(t) \leq -2c\varepsilon_0. \tag{145}$$

The following analysis is motivated by the proof of Theorem 6 in Zhou et al. (2024), aiming to mitigate the effect of noise and ensure that the objective function is monotone decreasing. Observe that

$$
\begin{aligned}
&\mathcal{Q}(\lambda(t-1), \mathcal{P}) - \mathcal{Q}(\lambda(t), \mathcal{P}) \\
=&\mathcal{Q}(\lambda(t-1), \mathcal{P}) - \mathcal{Q}(\lambda(t-1), \hat{\mathcal{P}}(t)) + \mathcal{Q}(\lambda(t-1), \hat{\mathcal{P}}(t)) - \mathcal{Q}(\lambda(t), \hat{\mathcal{P}}(t)) + \mathcal{Q}(\lambda(t), \hat{\mathcal{P}}(t)) - \mathcal{Q}(\lambda(t), \mathcal{P})
\end{aligned}
\tag{146}
$$

According to (130) the recovered target allocation is eventually bounded away from zero in every coordinate. By the continuity of $\mathcal{Q}(\lambda, \mathcal{P})$ in $\mathcal{P}$ and the law of the iterated logarithm, there exists a constant $C_e > 0$ such that, for all sufficiently large $t$,

$$
\left| \mathcal{Q}(\lambda(t-1), \mathcal{P}) - \mathcal{Q}(\lambda(t-1), \hat{\mathcal{P}}(t)) + \mathcal{Q}(\lambda(t), \hat{\mathcal{P}}(t)) - \mathcal{Q}(\lambda(t), \mathcal{P}) \right| \leq C_e \sqrt{\frac{\log \log t}{t}}.
\tag{147}
$$

Since the second derivative of $\mathcal{Q}(\lambda, \hat{\mathcal{P}}(t))$ with respect to $\lambda$ is bounded in the relevant neighborhood, Taylor's theorem yields

$$
\mathcal{Q}(\lambda(t-1) + s d(t), \hat{\mathcal{P}}(t)) \leq \mathcal{Q}(\lambda(t-1), \hat{\mathcal{P}}(t)) + s \nabla \mathcal{Q}(\lambda(t-1), \hat{\mathcal{P}}(t))^\top d(t) + \frac{s^2 \tilde{C}}{2} \|d(t)\|_2^2.
\tag{148}
$$

Since $d(t) \in \mathcal{D}^{m,n}(\lambda(t-1))$, we have $\|d(t)\|_2^2 \leq 2$. Hence, the Armijo line-search condition is satisfied whenever

$$
s \leq \frac{(1-\alpha)(-\mathcal{W}(t))}{\tilde{C}}.
\tag{149}
$$

Thus, the backtracking line search returns a step size satisfying

$$
s(t) \geq \min \left\{ s^{\max}(d(t), \lambda(t-1)), \frac{\nu(1-\alpha)(-\mathcal{W}(t))}{\tilde{C}} \right\}.
\tag{150}
$$

Using the line search condition, we obtain

$$
\begin{aligned}
&\mathcal{Q}(\lambda(t-1), \hat{\mathcal{P}}(t)) - \mathcal{Q}(\lambda(t), \hat{\mathcal{P}}(t)) \\
\geq &-\alpha s(t) \mathcal{W}(t) \\
\geq &\min \left\{ -\alpha s^{\max}(d(t), \lambda(t-1)) \mathcal{W}(t), \frac{\alpha \nu (1-\alpha) \mathcal{W}(t)^2}{\tilde{C}} \right\}.
\end{aligned}
\tag{151}
$$

By the update condition in Algorithm 2, whenever the line search is invoked, the maximal-step decrease or the squared-gradient decrease is at least of order $(\log t / t)^{1/2}$. Hence, there exists a constant $c_1 > 0$ such that, for all sufficiently large $t \in \mathcal{T}$,

$$
\mathcal{Q}(\lambda(t-1), \hat{\mathcal{P}}(t)) - \mathcal{Q}(\lambda(t), \hat{\mathcal{P}}(t)) \geq c_1 \left( \frac{\log t}{t} \right)^{1/2}.
\tag{152}
$$

Consequently, for all sufficiently large $t \in \mathcal{T}$,

$$
\begin{aligned}
&\mathcal{Q}(\lambda(t-1), \mathcal{P}) - \mathcal{Q}(\lambda(t), \mathcal{P}) \\
\geq &c_1 \left( \frac{\log t}{t} \right)^{1/2} - C_e \sqrt{\frac{\log \log t}{t}} > 0.
\end{aligned}
\tag{153}
$$

When the line search is not invoked, the algorithm sets $\lambda(t) = \lambda(t-1)$, so $\mathcal{Q}(\lambda(t), \mathcal{P})$ remains unchanged. Hence, the true objective sequence is eventually nonincreasing.

Moreover, note that $\mathcal{Q}(\lambda(t-1), \mathcal{P})$ is bounded below since, for any feasible $\lambda$,

$$
\mathcal{Q}(\lambda, \mathcal{P}) = - \sum_{h \in [D]} \sqrt{\sum_{i \in [K], j \in [M]} \lambda_{ij} \chi_h(x_i, c_j, \mathcal{P})} \geq - \sum_{h \in [D]} \sqrt{\sum_{i \in [K], j \in [M]} \chi_h(x_i, c_j, \mathcal{P})}.
\tag{154}
$$

Therefore, the monotone convergence theorem implies that $\{\mathcal{Q}(\lambda(t), \mathcal{P})\}$ converges to a finite value. In particular,

$$\lim_{t \to \infty, \, t \in \mathcal{T}_2} [\mathcal{Q}(\lambda(t-1), \mathcal{P}) - \mathcal{Q}(\lambda(t), \mathcal{P})] = 0. \tag{155}$$

Since $\lambda(t) = \lambda(t-1) + s(t)d(t)$ for $t \in \mathcal{T}_2$, equation (155) gives

$$\lim_{t \to \infty, \, t \in \mathcal{T}_2} [\mathcal{Q}(\lambda(t-1), \mathcal{P}) - \mathcal{Q}(\lambda(t-1) + s(t)d(t), \mathcal{P})] = 0. \tag{156}$$

By Lemma A.4, it follows that

$$\lim_{t \to \infty, \, t \in \mathcal{T}_2} s^{\max}(d(t), \lambda(t-1)) \nabla \mathcal{Q}(\lambda(t-1), \mathcal{P})^\top d(t) = 0. \tag{157}$$

This contradicts (145). Hence, the assumption that $\bar{\lambda}$ is not stationary is false. Therefore, every limit point $\bar{\lambda}$ must be a stationary point of the dual problem (19). $\qquad\square$

We are now ready to establish the sample complexity upper bound stated in Theorem 4.2. Our analysis builds on the framework proposed by Garivier & Kaufmann (2016), which has been widely adopted in the BAI literature (Juneja & Krishnasamy, 2019; Wang et al., 2021).

*Proof.* We begin by defining the following clean event:

$$\mathcal{E} = \left\{ \max_{h \in [D]} \left| \frac{N_h(t)}{t} - \omega_h^*(\mathcal{P}) \right| \to 0, \hat{\mathcal{P}}(t) \to \mathcal{P} \right\}. \tag{158}$$

By Lemma A.6, every limit point of the sequence $\{\lambda(t)\}$ generated by the algorithm is a stationary point of the dual problem (19). Since the dual problem is convex, every stationary point is globally optimal. Moreover, by Lemma 3.6, strong duality holds, and the corresponding primal solution can be recovered through (22). Hence, every limit point of $\{\gamma(\hat{\mathcal{P}}(t))\}$ is an optimal solution to the primal problem (18). By the uniqueness of the optimal sampling ratio, for any $\epsilon > 0$, there exists $t_0 > 0$ such that

$$\sup_{t \geq t_0} \max_{h \in [D]} \left| \gamma_h(\hat{\mathcal{P}}(t)) - \omega_h^*(\mathcal{P}) \right| \leq \epsilon. \tag{159}$$

Furthermore, by Lemma A.5, there exists $t_1 > 0$ such that

$$\sup_{t \geq t_1} \max_{h \in [D]} \left| \frac{N_h(t)}{t} - \omega_h^*(\mathcal{P}) \right| \leq 3(D-1)\epsilon. \tag{160}$$

In addition, since $N_h(t) \geq (\sqrt{t} - D/2)_+ - 1$, the strong law of large numbers implies that $\hat{\mathcal{P}}(t) \to \mathcal{P}$ almost surely. Therefore, we conclude that $\mathbb{P}(\mathcal{E}) = 1$.

Condition on the clean event $\mathcal{E}$. By Lemma A.2, the function $\mathcal{U}(\omega, \mathcal{P})^{-1}$ is continuous in both $\omega$ and $\mathcal{P}$. Thus, for any $\epsilon > 0$, there exists $t_2 > 0$ such that for all $t \geq t_2$,

$$\mathcal{U}(\omega(t), \hat{\mathcal{P}}(t))^{-1} \geq (1 - \epsilon)\mathcal{U}(\omega^*(\mathcal{P}), \mathcal{P})^{-1}. \tag{161}$$

Recall that

$$\rho(t, \delta) = \log(1/\delta) + \alpha \log t + (4D + 1) \log \log(1/\delta) + \log C. \tag{162}$$

For any fixed $\epsilon > 0$, since the additional terms in $\rho(t, \delta)$ beyond $\log(1/\delta)$ are of order $\log t + \log \log(1/\delta)$, there exists $t_3 = t_3(\epsilon, \delta) > 0$ such that for all $t \geq t_3$,

$$\rho(t, \delta) \leq \log(1/\delta) + \epsilon \mathcal{U}(\omega^*(\mathcal{P}), \mathcal{P})^{-1} t. \tag{163}$$

Moreover, $t_3$ can be chosen such that

$$t_3 = o(\log(1/\delta)), \qquad \text{as } \delta \to 0. \tag{164}$$

Then, on the clean event $\mathcal{E}$, the stopping time satisfies

$$
\begin{aligned}
\tau &= \inf\left\{ t \in \mathbb{N} : t\mathcal{U}(\omega(t), \hat{\mathcal{P}}(t))^{-1} \geq \rho(t, \delta) \right\} \\
&\leq \inf\left\{ t \in \mathbb{N}, \ t \geq t_2 + t_3 : t(1-\epsilon)\mathcal{U}(\omega^*(\mathcal{P}), \mathcal{P})^{-1} \geq \log(1/\delta) + \epsilon\mathcal{U}(\omega^*(\mathcal{P}), \mathcal{P})^{-1}t \right\} \\
&= \inf\left\{ t \in \mathbb{N}, \ t \geq t_2 + t_3 : t(1-2\epsilon)\mathcal{U}(\omega^*(\mathcal{P}), \mathcal{P})^{-1} \geq \log(1/\delta) \right\} \\
&\leq t_2 + t_3 + \inf\left\{ t \in \mathbb{N} : t(1-2\epsilon)\mathcal{U}(\omega^*(\mathcal{P}), \mathcal{P})^{-1} \geq \log(1/\delta) \right\} \\
&\leq t_2 + t_3 + \frac{\mathcal{U}(\omega^*(\mathcal{P}), \mathcal{P})\log(1/\delta)}{1 - 2\epsilon} + 1.
\end{aligned}
\tag{165}
$$

Therefore, since $t_2$ does not depend on $\delta$ and $t_3 = o(\log(1/\delta))$, we obtain

$$\limsup_{\delta \to 0} \frac{\tau}{\log(1/\delta)} \leq \frac{\mathcal{U}(\omega^*(\mathcal{P}), \mathcal{P})}{1 - 2\epsilon}. \tag{166}$$

Letting $\epsilon \to 0$, we obtain

$$\mathbb{P}\left( \limsup_{\delta \to 0} \frac{\tau}{\log(1/\delta)} \leq \mathcal{U}^*(\mathcal{P}) \right) = 1. \tag{167}$$

We now prove the expected sample complexity bound. For any fixed $\epsilon > 0$, define the good event

$$\mathcal{E}_T = \bigcap_{t \geq \lceil T^{1/4} \rceil} \left\{ \left\| \hat{\mathcal{P}}(t) - \mathcal{P} \right\|_\infty \leq \xi(\epsilon) \right\}. \tag{168}$$

where $\xi(\epsilon) > 0$ is chosen sufficiently small.

On $\mathcal{E}_T$, the estimated feasible sets are nonempty, the estimated optimal arms coincide with the true optimal arms, and $\hat{\mathcal{P}}(t)$ remains uniformly close to $\mathcal{P}$ for all $t \geq \lceil T^{1/4} \rceil$. Moreover, along the iterates generated by the algorithm, the denominator terms in $\nabla \mathcal{Q}$ are uniformly bounded away from zero for all sufficiently large $T$. Hence, on the relevant neighborhood of these iterates, $\mathcal{Q}(\cdot, \hat{\mathcal{P}}(t))$ and its subgradients are uniformly close to $\mathcal{Q}(\cdot, \mathcal{P})$ and its subgradients.

Let

$$\mathcal{Q}^*(\mathcal{P}) = \min_{\lambda \in \Delta_{KM}} \mathcal{Q}(\lambda, \mathcal{P}), \qquad \Delta_{KM} = \left\{ \lambda \in \mathbb{R}_+^{KM} : \sum_{i \in [K], j \in [M]} \lambda_{ij} = 1 \right\}.$$

Fix $\eta > 0$. By compactness and Lemma 4.1, there exists a constant $a_\eta > 0$ such that, whenever

$$\mathcal{Q}(\lambda, \mathcal{P}) \geq \mathcal{Q}^*(\mathcal{P}) + \eta,$$

the reduced direction set used in Algorithm 2 contains a feasible direction $d$ satisfying

$$s^{\max}(d)\nabla\mathcal{Q}(\lambda, \mathcal{P})^\top d \leq -a_\eta. \tag{169}$$

By the uniform closeness of empirical and true subgradients on $\mathcal{E}_T$, for all sufficiently large $T$ and all $t \geq \lceil T^{1/4} \rceil$, the same inequality holds for the empirical gradient up to a factor $1/2$. Therefore, whenever

$$\mathcal{Q}(\lambda(t-1), \mathcal{P}) \geq \mathcal{Q}^*(\mathcal{P}) + \eta,$$

Algorithm 2 selects a direction $d(t)$ such that

$$s^{\max}(d(t))\mathcal{W}(t) \leq -\frac{a_\eta}{2}. \tag{170}$$

Since the update thresholds in Algorithm 2 vanish, the line search is invoked for all sufficiently large $t$ satisfying the above condition. The Armijo line search and (170) imply that there exists a constant $b_\eta > 0$, independent of $t$, such that, on $\mathcal{E}_T$ and for all sufficiently large $T$,

$$\mathcal{Q}(\lambda(t-1), \mathcal{P}) - \mathcal{Q}(\lambda(t), \mathcal{P}) \geq b_\eta \tag{171}$$

whenever

$$\mathcal{Q}(\lambda(t-1), \mathcal{P}) \geq \mathcal{Q}^*(\mathcal{P}) + \eta.$$

Since $\mathcal{Q}(\cdot, \mathcal{P})$ is bounded below by $\mathcal{Q}^*(\mathcal{P})$, such uniform decreases can occur only finitely many times. Hence, on $\mathcal{E}_T$, there exists a finite constant $T_\eta^{\text{dual}}$, independent of $\delta$, such that, for all $t \geq T_\eta^{\text{dual}}$,

$$\mathcal{Q}(\lambda(t), \mathcal{P}) \leq \mathcal{Q}^*(\mathcal{P}) + \eta. \tag{172}$$

By strong duality and the continuity of the recovery map, choosing $\eta$ and $\xi(\epsilon)$ sufficiently small gives, for all sufficiently large $t$ on $\mathcal{E}_T$,

$$\mathcal{U}(\gamma(\hat{\mathcal{P}}(t)), \hat{\mathcal{P}}(t))^{-1} \geq \left(1 - \frac{\epsilon}{2}\right) \mathcal{U}(\omega^*(\mathcal{P}), \mathcal{P})^{-1}. \tag{173}$$

The tracking rule then implies that the empirical allocation $\omega(t)$ tracks the returned sampling ratios. Hence, by Lemma A.5, there exists a finite constant $T_0(\epsilon)$, independent of $\delta$, such that, for all $T \geq T_0(\epsilon)$ and all $t \in [\lceil T^{1/2} \rceil, T]$,

$$\mathcal{U}(\omega(t), \hat{\mathcal{P}}(t))^{-1} \geq (1 - \epsilon)\mathcal{U}(\omega^*(\mathcal{P}), \mathcal{P})^{-1}. \tag{174}$$

Therefore, on $\mathcal{E}_T$ and for all $T \geq T_0(\epsilon)$,

$$
\begin{aligned}
\min(\tau, T) &\leq \lceil T^{1/2} \rceil + \sum_{t=\lceil T^{1/2} \rceil}^{T} \mathbb{I}(\tau > t) \\
&\leq \lceil T^{1/2} \rceil + \sum_{t=\lceil T^{1/2} \rceil}^{T} \mathbb{I}\left(t\mathcal{U}(\omega(t), \hat{\mathcal{P}}(t))^{-1} \leq \rho(t, \delta)\right) \\
&\leq \lceil T^{1/2} \rceil + \sum_{t=\lceil T^{1/2} \rceil}^{T} \mathbb{I}\left(t \leq \frac{\rho(T, \delta)}{(1-\epsilon)\mathcal{U}(\omega^*(\mathcal{P}), \mathcal{P})^{-1}}\right) \\
&\leq \lceil T^{1/2} \rceil + \frac{\rho(T, \delta)\mathcal{U}(\omega^*(\mathcal{P}), \mathcal{P})}{1 - \epsilon} + 1.
\end{aligned}
\tag{175}
$$

Define

$$T_\epsilon^*(\delta) = \inf\left\{T \in \mathbb{N} : \lceil T^{1/2} \rceil + \frac{\rho(T, \delta)\mathcal{U}(\omega^*(\mathcal{P}), \mathcal{P})}{1 - \epsilon} + 1 \leq T\right\}.$$

Then for all $T \geq \bar{T}_\epsilon(\delta) \triangleq \max\{T_0(\epsilon), T_\epsilon^*(\delta)\}$, it holds that $\mathcal{E}_T \subseteq (\tau \leq T)$.

Thus, we obtain:

$$
\begin{aligned}
\mathbb{E}[\tau] &= \sum_{T=0}^{\infty} \mathbb{P}(\tau > T) \\
&\leq \bar{T}_\epsilon(\delta) + \sum_{T=\bar{T}_\epsilon(\delta)}^{\infty} \mathbb{P}(\mathcal{E}_T^c) \\
&\leq T_0(\epsilon) + T_\epsilon^*(\delta) + \sum_{T=1}^{\infty} \mathbb{P}(\mathcal{E}_T^c).
\end{aligned}
$$

By Lemma 18 of Garivier & Kaufmann (2016), we have

$$T_\epsilon^*(\delta) = \frac{\mathcal{U}(\omega^*(\mathcal{P}), \mathcal{P})}{1 - \epsilon} \left(O(\log(1/\delta)) + O(\log\log(1/\delta))\right). \tag{176}$$

We next bound the tail probability of $\mathcal{E}_T^c$. Since the design matrix $\Phi$ is invertible, there exists a constant $C_\Phi > 0$ such that

$$\|\hat{\mathcal{P}}(t) - \mathcal{P}\|_\infty \leq C_\Phi \max_{h \in [D]} \left( |\bar{F}(z_h; t) - f(z_h)| + |\bar{G}(z_h; t) - g(z_h)| \right).$$

Let $\bar{\xi}(\epsilon) = \frac{\xi(\epsilon)}{2C_\Phi}$. Then

$$\mathbb{P}(\mathcal{E}_T^c) \leq \sum_{t \geq \lceil T^{1/4} \rceil} \sum_{h=1}^{D} \left[ \mathbb{P}\left( |\bar{F}(z_h; t) - f(z_h)| > \bar{\xi}(\epsilon) \right) + \mathbb{P}\left( |\bar{G}(z_h; t) - g(z_h)| > \bar{\xi}(\epsilon) \right) \right]. \tag{177}$$

For sufficiently large $t$, Lemma A.5 implies $N_h(t) \geq \sqrt{t} - D$.

Although $N_h(t)$ is random, we take a union bound over all possible sample counts. Hence, for each $h \in [D]$,

$$\mathbb{P}\left( \bar{F}(z_h; t) < f(z_h) - \bar{\xi}(\epsilon) \right)$$

$$\leq \sum_{s = \lceil \sqrt{t} - D \rceil}^{t} \mathbb{P}\left( \bar{F}_s(z_h) \leq f(z_h) - \bar{\xi}(\epsilon) \right)$$

$$\leq \sum_{s = \lceil \sqrt{t} - D \rceil}^{t} \exp\left( -s \, d(f(z_h) - \bar{\xi}(\epsilon), f(z_h)) \right) \tag{178}$$

$$\leq \frac{\exp\left( -(\sqrt{t} - D) d(f(z_h) - \bar{\xi}(\epsilon), f(z_h)) \right)}{1 - \exp\left( -d(f(z_h) - \bar{\xi}(\epsilon), f(z_h)) \right)}.$$

Similarly,

$$\mathbb{P}\left( \bar{F}(z_h; t) > f(z_h) + \bar{\xi}(\epsilon) \right) \leq \frac{\exp\left( -(\sqrt{t} - D) d(f(z_h) + \bar{\xi}(\epsilon), f(z_h)) \right)}{1 - \exp\left( -d(f(z_h) + \bar{\xi}(\epsilon), f(z_h)) \right)}. \tag{179}$$

The same bounds hold for $\bar{G}(z_h; t)$.

Choose $\bar{\xi}(\epsilon) > 0$ small enough so that all shifted means remain in the parameter space, and define

$$c_\xi = \min_{h \in [D]} \min \left( d(f(z_h) - \bar{\xi}(\epsilon), f(z_h)), d(f(z_h) + \bar{\xi}(\epsilon), f(z_h)), \right.$$

$$\left. d(g(z_h) - \bar{\xi}(\epsilon), g(z_h)), d(g(z_h) + \bar{\xi}(\epsilon), g(z_h)) \right) > 0. \tag{180}$$

Then there exist finite constants $B'_\xi, c'_\xi > 0$ such that

$$\mathbb{P}(\mathcal{E}_T^c) \leq B'_\xi \exp(-c'_\xi T^{1/8}). \tag{181}$$

Therefore,

$$\sum_{T=1}^{\infty} \mathbb{P}(\mathcal{E}_T^c) < \infty.$$

Combining the above bounds, we obtain

$$\limsup_{\delta \to 0} \frac{\mathbb{E}[\tau]}{\log(1/\delta)} \lesssim \frac{1}{1 - \epsilon} \mathcal{U}(\omega^*(\mathcal{P}), \mathcal{P}). \tag{182}$$

Letting $\epsilon \to 0$ completes the proof. $\square$

## A.11. Computational Complexity

Since the main difference between Algorithm 1 (TS) and the proposed algorithm (DSR) lies in how the empirical sampling ratio is computed, we focus on this step. In TS, the empirical optimal sampling ratio $\omega^*(\hat{\mathcal{P}}(t))$ is obtained by repeatedly solving the primal optimization problem. If gradient descent is used, each iteration requires evaluating the objective over

all arm-covariate pairs and computing the inverse of $\Lambda(\omega)$, leading to a per-gradient-step cost of $\mathcal{O}(D^3 + KMD)$. Since $\mathcal{O}(1/\epsilon)$ gradient steps are required to reach accuracy $\epsilon$, the total computational cost is $\mathcal{O}(\frac{1}{\epsilon}(D^3 + KMD))$. In contrast, DSR performs only one dual-coordinate update at each sampling time. The inverse of the fixed design matrix $\Phi$ can be precomputed and reused. Given the quantities $\chi_h(x_i, c_j)$, computing the dual gradient over all arm-covariate pairs costs $\mathcal{O}(KMD)$. The line search only updates the $D$ quantities

$$\sum_{i\in[K],\, j\in[M]} \lambda_{ij}\chi_h(x_i, c_j), \qquad h \in [D],$$

and hence costs $\mathcal{O}(D\log(1/\epsilon'))$ to reach precision $\epsilon'$. Therefore, the per-iteration complexity of DSR is $\mathcal{O}(KMD + D\log(1/\epsilon'))$. Hence, DSR substantially reduces the computational cost by replacing full optimization in TS with a single dual-coordinate descent step.

### A.12. Numerical Experiment

This subsection provides the detailed parameter settings and pseudo-code for the benchmark algorithms used in the numerical experiments.

**DSR**. Algorithm 4 outlines the complete pseudo-code for the proposed duality-based decomposition algorithm. The overall framework follows the structure of the Track-and-Stop algorithm, with the key difference being that the sampling ratio $\gamma(\hat{\mathcal{P}}(t))$ is computed using Algorithm 2. In our implementation, we adopt a heuristic step size of $s(t) = 0.01$ and a threshold parameter $\rho(t, \delta) = \log((\log(t) + 1)/\delta)$, the latter of which is commonly used in the best arm identification (BAI) literature (Garivier & Kaufmann, 2016; Wang et al., 2021).

---

**Algorithm 4** Duality-based Decomposition Algorithm (DSR)

---

**Input:** Covariate set $\mathcal{C}$, arm set $\mathcal{X}$, design point set $\mathcal{Z}$, confidence level $\delta$, $\lambda(0) = 1/KM$.
**Initialization:** Sample each design point $z_h \in \mathcal{Z}$ $n_0$ times.
Set $t \leftarrow n_0 D$ and update $N_h(t), \omega_h(t), \hat{\mathcal{P}}(t), \Lambda(\omega(t))$.
**while** $t\mathcal{U}(\omega(t), \hat{\mathcal{P}}(t))^{-1} < \rho(t, \delta)$ **do**
    **if** $\mathcal{B}_t \neq \emptyset$ **then**
        $z_{h(t+1)} = \arg\min_{z_h \in \mathcal{B}_t} N_h(t)$
    **else**
        $\lambda(t), \gamma(\hat{\mathcal{P}}(t)) \leftarrow$ Algorithm 2 $(\mathcal{C}, \mathcal{X}, \mathcal{Z}, \kappa_0, \eta, \hat{\mathcal{P}}(t), \hat{\theta}(t), \hat{\beta}(t), \lambda(t-1))$
        $z_{h(t+1)} = \arg\min_{z_h \in \mathcal{Z}} N_h(t) - t\gamma_h(\hat{\mathcal{P}}(t))$
    Sample the design point $z_{h(t+1)}$ and obtain the observation $Z_{t+1}$.
    Set $t \leftarrow t + 1$, and update $N_h(t), \omega_h(t), \hat{\mathcal{P}}(t), \Lambda(\omega(t))$.
**return** For each covariate $c_j \in \mathcal{C}$, recommend the estimated best arm:

$$x_{\hat{i}(c_j;\tau)} = \arg\max_{x_i \in \mathcal{X}} \hat{\theta}(\tau)^\top \phi(x_i, c_j)$$
$$\text{s.t.} \quad \hat{\beta}(\tau)^\top \phi(x_i, c_j) \leq b$$

---

**USR**. Algorithm 5 presents the pseudo-code for the USR algorithm. At each time step $t$, it samples all design points uniformly, without incorporating any information from the arms.

---

**Algorithm 5** USR Algorithm

---
**Input:** Covariate set $\mathcal{C}$, arm set $\mathcal{X}$, design point set $\mathcal{Z}$, confidence level $\delta$.

**while** $t\mathcal{U}(\omega(t), \hat{\mathcal{P}}(t))^{-1} < \rho(t, \delta)$ **do**
  $z_{h(t+1)} = \arg\min_{z_h \in \mathcal{Z}} N_h(t)$
  Sample the design point $z_{h(t+1)}$ and obtain the observation $Z_{t+1}$.
  Set $t \leftarrow t + 1$, and update $N_h(t), \omega_h(t), \hat{\mathcal{P}}(t), \Lambda(\omega(t))$.
**return** For each covariate $c_j \in \mathcal{C}$, recommend the estimated best arm:

$$x_{\hat{i}(c_j; \tau)} = \arg\max_{x_i \in \mathcal{X}} \hat{\theta}(\tau)^\top \phi(x_i, c_j)$$
$$\text{s.t.} \quad \hat{\beta}(\tau)^\top \phi(x_i, c_j) \leq b$$

---

Algorithm 6 presents the pseudo-code for the BCSR, GOSR, and GFSR algorithms. All three algorithms employ a score-based approach to determine the sampling rule, with the key distinction being how each algorithm defines its respective score.

**BCSR**. This algorithm is inspired by the state-of-the-art Best Challenger algorithm proposed by Garivier & Kaufmann (2016). It relies solely on the optimality information of each arm. For each design point, the score at time step $t$ is defined as:

$$S_h(\hat{\mathcal{P}}(t), \omega(t)) = \frac{(\hat{f}(z_h; t) - \hat{f}(x_{\hat{i}(c_j; t)}, c_j))^2}{\sigma_h^2 / N_h(t)}, \tag{183}$$

where $x_{\hat{i}(c_j; t)} = \arg\max_{x_i \in \mathcal{X}} \hat{\theta}(t)^\top \phi(x_i, c_j)$ denotes the estimated best arm under covariate $c_j$. This score captures a trade-off between the estimated optimality gap and the sampling variance.

If the design point corresponds to the estimated best arm, then its score is defined as:

$$S_h(\hat{\mathcal{P}}(t), \omega(t)) = \min_{z_l \in \mathcal{Z} \setminus \{z_h\}} S_l(\hat{\mathcal{P}}(t), \omega(t)), \tag{184}$$

meaning the best arm is assigned the minimum score. The algorithm then randomly selects among design points with the lowest score for sampling.

---

**Algorithm 6** BCSR/GOSR/GFSR Algorithm

---
**Input:** Covariate set $\mathcal{C}$, arm set $\mathcal{X}$, design point set $\mathcal{Z}$, confidence level $\delta$.
**Initialization:** Sample each design point $z_h \in \mathcal{Z}$ $n_0$ times.
Set $t \leftarrow n_0 D$ and update $N_h(t), \omega_h(t), \hat{\mathcal{P}}(t), \Lambda(\omega(t))$.
**while** $t\mathcal{U}(\omega(t), \hat{\mathcal{P}}(t))^{-1} < \rho(t, \delta)$ **do**
  **if** $\mathcal{B}_t \neq \emptyset$ **then**
    $z_{h(t+1)} = \arg\min_{z_h \in \mathcal{B}_t} N_h(t)$
  **else**
    $z_{h(t+1)} = \arg\min_{z_h \in \mathcal{Z}} S_h(\hat{\mathcal{P}}(t), \omega(t))$
  Sample the design point $z_{h(t+1)}$ and obtain the observation $Z_{t+1}$.
  Set $t \leftarrow t + 1$, and update $N_h(t), \omega_h(t), \hat{\mathcal{P}}(t), \Lambda(\omega(t))$.
**return** For each covariate $c_j \in \mathcal{C}$, recommend the estimated best arm:

$$x_{\hat{i}(c_j; \tau)} = \arg\max_{x_i \in \mathcal{X}} \hat{\theta}(\tau)^\top \phi(x_i, c_j)$$
$$\text{s.t.} \quad \hat{\beta}(\tau)^\top \phi(x_i, c_j) \leq b$$

---

**GOSR**. This algorithm is motivated by the surrogate optimization problem (18) and relies solely on optimality information. For each covariate $c_j \in \mathcal{C}$, the estimated best arm is defined as $x_{\hat{i}(c_j; t)} = \arg\max_{x_i \in \mathcal{X}} \hat{\theta}(t)^\top \phi(x_i, c_j)$. For each design

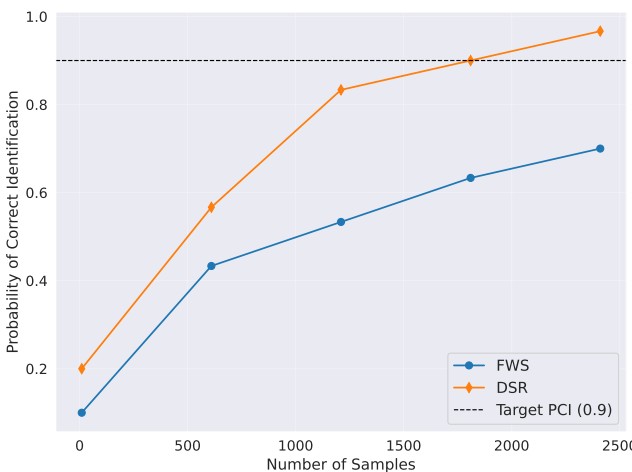

*Figure 4.* Comparison of empirical PCI

point, the score at time step $t$ is defined as

$$S_h(\hat{\mathcal{P}}(t), \omega(t)) = \frac{(\hat{f}(z_h; t) - \hat{f}(x_{\hat{i}(c_j; t)}, c_j))^2}{\|\phi(x_{\hat{i}(c_j; t)}, c_j) - \phi(x_i, c_j)\|^2_{\Lambda(\omega(t))^{-1}}}, \tag{185}$$

Similarly, the score for the estimated best arm is defined according to (184).

**GFSR**. The general algorithmic framework of GFSR is identical to that of GOSR, except that GFSR relies solely on feasibility information to determine the sampling rule. Specifically, the score for each design point at time step $t$ is defined as

$$S_h(\hat{\mathcal{P}}(t), \omega(t)) = \frac{(b - \hat{\beta}(t)^\top \phi(x_i, c_j))^2}{\|\phi(x_i, c_j)\|^2_{\Lambda(\omega)^{-1}}}, \tag{186}$$

where the score quantifies the deviation in feasibility performance. The score for the estimated best arm is defined in the same way as in (184).

**Comparison with Frank-Wolfe Sampling (Wang et al., 2021).** (Wang et al., 2021) propose a general framework for pure exploration via Frank–Wolfe. However, their algorithm cannot be directly applied to our constrained covariate-selection setting without substantial modification. First, the presence of constraints complicates the gradient computation in Proposition 1 of (Wang et al., 2021). The gradient calculation depends on the most confusing alternative instance. When constraints are considered, the alternative problem instance set $\mathcal{A}(\mathcal{P})$ becomes more complex, as it depends on both the optimality and feasibility of the arms. We need to classify arms into four subclasses and construct the alternative instance for each class differently. For infeasible arms with worse performance, the alternative instance is particularly complex, as it depends simultaneously on both the objective and the constraint performance measures. Second, the covariate selection setting makes the Frank–Wolfe update, which involves solving a game, more complicated. (Wang et al., 2021) handle non-smooth objectives via the $r$-subdifferential subspace. In our setting, covariate selection introduces an additional layer of optimization over all possible covariates in the sample complexity lower bound. This increases the number of non-smooth points in the overall objective, making the Frank-Wolfe update, which solves the game over a simplex and the convex hull of the gradient vectors, more time-consuming.

We also compare the numerical performance of the proposed DSR with Frank-Wolfe Sampling (FWS) on the same problem used in the numerical experiment. Each algorithm is run for 3000 iterations, and we report the total running time and the empirical PCI over 30 independent macro replications. The results show that DSR completes in 56 seconds, whereas FWS takes 917 seconds, which is approximately 16 times longer than DSR. Moreover, Figure 4 shows that DSR achieves a PCI exceeding 0.9, while the PCI of FWS is below 0.8. Therefore, DSR also demonstrates superior empirical performance compared with FWS.

**Parameter setting.** The experimental setup is inspired by the numerical example in Soare et al. (2014). There are two covariates, $\mathcal{C} = \{c_1, c_2\}$, four arms, $\mathcal{X} = \{x_1, \ldots, x_4\}$, and one constraint. The threshold parameter $b$ in the constraint

*Table 1.* Sample complexity comparison of various algorithms under different gaps

| Method | Mean (0.2) | Lower | Upper | Mean(0.3) | Lower | Upper |
|--------|-----------|----------|----------|-----------|---------|----------|
| USR | 12786.50 | 9756.38 | 15816.62 | 12313.53 | 9405.00 | 15222.06 |
| DSR | 5282.73 | 4023.10 | 6542.37 | 4100.33 | 2969.65 | 5231.01 |
| GOSR | 21274.73 | 14772.71 | 27776.76 | 8861.53 | 7200.30 | 10522.77 |
| GFSR | 6537.10 | 5241.74 | 7832.46 | 6526.23 | 5312.21 | 7740.25 |
| BCSR | 16936.70 | 12649.17 | 21224.23 | 7825.07 | 6505.59 | 9144.54 |

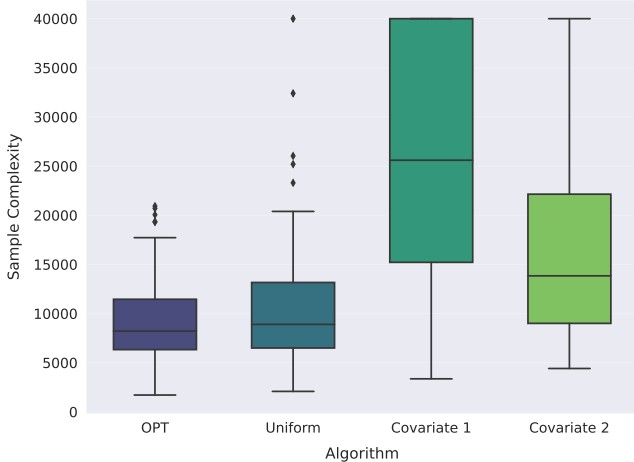

*Figure 5.* Sample complexity of DSR under different covariate selection rules

of problem (1) is set to $b = 0.5$. The dimension of the unknown parameter vectors $\theta$ and $\beta$ is $D = 7$. Specifically, $\theta = [1.0, 0.0, 0.0, 0.0, 1.0, 1.2, 0.0]^\top$, and $\beta = [0.45, 0.0, 0.0, 0.0, 0.0, 0.6, 0.8]^\top$. Let $e_l \in \mathbb{R}^D$ denote the $l$th standard basic vector, with the $l$th element equal to one and all other elements zero. The feature vectors of the arm-covariate pairs are defined as $\phi(x_1, c_1) = e_1, \phi(x_2, c_1) = e_2, \dots, \phi(x_3, c_2) = e_7$, and $\phi(x_4, c_2) = [\cos(0.4), \sin(0.4), 0, \dots, 0]^\top$. The design point set is $\mathcal{Z} = \{(x_1, c_1), (x_2, c_1), \dots, (x_3, c_2)\}$ with $|\mathcal{Z}| = 7$, meaning that the design points correspond to the standard basis vectors in $\mathbb{R}^D$. The variance of each arm-covariate pair is independently drawn from a uniform distribution over $[0.5, 1.0]$. For computational convenience during implementation, we use a heuristic step size $s(t) = 0.01$ and a threshold parameter $\rho(t, \delta) = \log((\log(t) + 1)/\delta)$, the latter of which is also employed in the BAI literature (Garivier & Kaufmann, 2016; Wang et al., 2021).

**Robustness evaluation.** We report additional sample complexity results for small ($\Delta = 0.2$) and large ($\Delta = 0.3$) feasibility and optimality gaps to assess the robustness of the proposed algorithm across different problem instances. Table 1 summarizes the sample complexity of various algorithms at a confidence level of $\delta = 0.1$, with "lower" and "upper" indicating the 90% confidence interval bounds. Our proposed Algorithm DSR consistently outperforms other methods, and larger gaps correspond to lower sample complexity.

**Effect of covariate selection rule.** We examine the importance of covariate selection by comparing the sample complexity of DSR under different covariate selection rules. Specifically, we consider four rules: (1) **OPT**: active covariate selection according to the optimal sampling ratio; (2) **Uniform**: covariates are passively sampled from a uniform distribution; (3) **Covariate 1**: the two covariates are sampled with probabilities 0.8 and 0.2, respectively; (4) **Covariate 2**: the two covariates are sampled with probabilities 0.2 and 0.8, respectively. Conditional on the covariate, the arm is sampled according to the optimal sampling ratio. To control the computation time, we set a maximum iteration limit of 40000, the algorithm terminates once the total number of samples reaches this threshold. Figure 5 presents the empirical sample complexity based on 100 independent macro-replications of DSR under the four covariate selection rules. The results indicate that optimal active covariate selection plays a crucial role in reducing sample complexity.

**Initial design points in $\mathcal{Z}$.** Figure 6 compares the sample complexity of DSR using three groups of different initial design

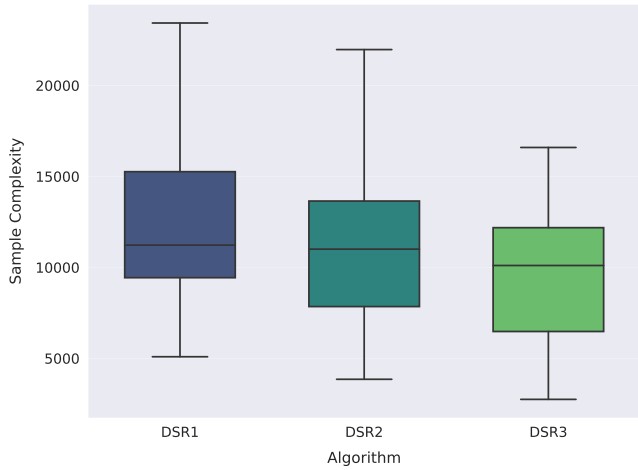

*Figure 6.* Sample complexity of DSR under different initial design points

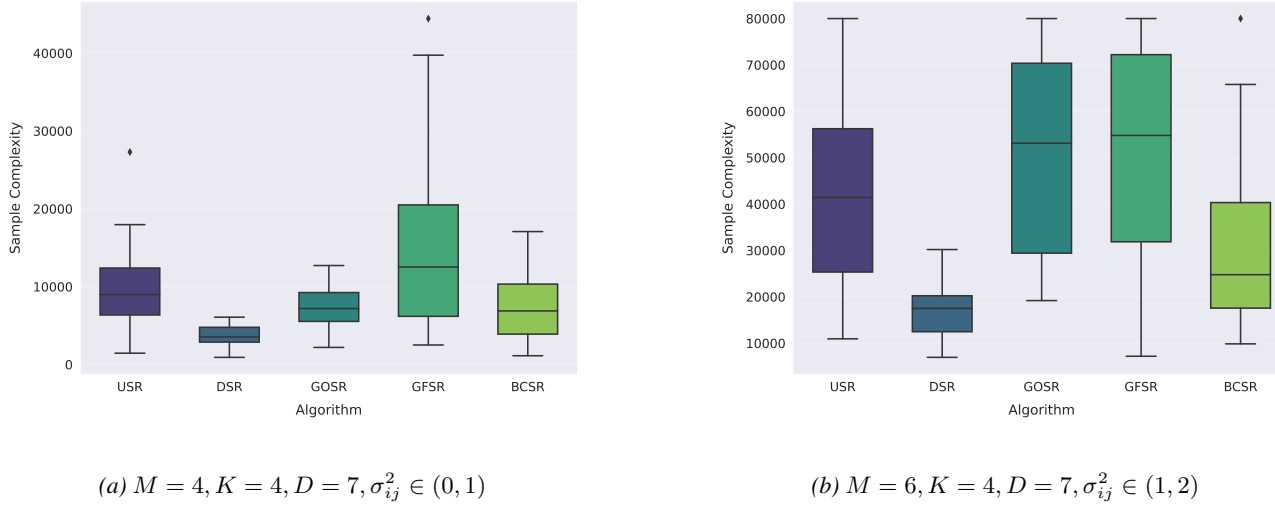

*(a)* $M = 4, K = 4, D = 7, \sigma_{ij}^2 \in (0, 1)$    *(b)* $M = 6, K = 4, D = 7, \sigma_{ij}^2 \in (1, 2)$

*Figure 7.* Sample complexity across different problem scales and noise levels

points $\mathcal{Z}$ in the current numerical example. The result shows that, although different initial design points do lead to variations in sample complexity, the differences are not substantial. This indicates that DSR is relatively robust to the choice of initial design points.

**Problem scale and noise level.** We compare the sample complexity under different problem scales and noise levels. To control computation time, we impose a maximum iteration limit of $80000$, and the algorithm terminates once the total number of samples reaches this threshold. Figure 7 reports the empirical sample complexity based on 30 independent macro-replications. As the problem size and noise level increase, the total number of samples required by all algorithms also increases. However, DSR consistently outperforms the other benchmarks.

**Compute resources.** The numerical experiments were conducted on a Windows machine equipped with an Intel® Xeon® Silver 4210R CPU @ 2.40GHz. Running the algorithm for 100 replications took less than 1 hour.

### A.13. Personalized Treatment for Diabetes Management

Diabetes mellitus (DM) affects over 500 million people globally (World Health Organization), with type 2 diabetes (T2D) comprising 90–95% of cases. Managing T2D is complex, with treatment options ranging from lifestyle modifications to various pharmacological therapies such as Metformin, each with differing efficacy and side effect profiles depending on individual patient characteristics (covariates). Therefore, it is important to identify the most suitable treatment plan tailored

*Table 2.* Comparison of Methods with different confidence level $\delta$

| Method | Mean (0.1) | Lower | Upper | Mean (0.2) | Lower | Upper |
|--------|-----------|-------|-------|-----------|-------|-------|
| USR | 38661.00 | 30180.15 | 47141.85 | 25130.70 | 19243.79 | 31017.61 |
| DSR | 13127.07 | 10211.51 | 16042.62 | 11114.93 | 8236.33 | 13993.54 |
| GOSR | 16892.83 | 13055.05 | 20730.62 | 13779.97 | 10091.68 | 17468.25 |
| GFSR | 51852.90 | 41498.39 | 62207.41 | 49358.80 | 37399.20 | 61318.40 |
| BCSR | 17004.70 | 12753.25 | 21256.15 | 13786.23 | 9995.75 | 17576.71 |

to each patient's specific characteristics.

We model this as a constrained linear BAI problem with covariate selection. Based on ADA/EASD clinical guidelines, we consider four drug classes—Metformin, Sulfonylureas, SGLT2 inhibitors, and GLP-1 receptor agonists—each with distinct benefits and risks. For example, Metformin improves insulin sensitivity and is generally well-tolerated; however, it is contraindicated in patients with severe renal impairment.

Patient covariates include HbA1c, BMI, and cardiovascular risk. Drug features include dose, frequency, hypoglycemia risk, and renal adjustment threshold. The goal is to identify the treatment that maximizes glycemic improvement while maintaining adverse effects below a risk threshold for each patient.

Table 2 compares the sample complexity of various algorithms in a setting with 2 patients, 7-dimensional features (D = 7), and confidence levels $\delta = 0.1$ and $\delta = 0.2$. Our algorithm DSR, which balances feasibility and optimality, consistently achieves the lowest sample complexity.

