# OpenReview forum: "Learning the Best Under Constraints: A Duality-Based Framework"
_ICML.cc/2026/Conference — ICML 2026 regular_

### Official Review · Reviewer_Cwae · 2026-02-27

**Soundness:** 2
**Presentation:** 1
**Significance:** 2
**Originality:** 3
**Overall Recommendation:** 3
**Confidence:** 3

**Summary:**

The authors study the problem of best arm selection. In their setting, there are covariates which are observed before an arm is chosen. The covariates change constraints on which arms can and can't be chosen. The authors specialize to the case where expected payoff is linear. In this case, they give a lower-bound on sample complexity to learn what they call a "PAC" policy, and then define an algorithm which meets this lower bound.

**Compliance With Llm Reviewing Policy:**

Affirmed.

**Final Justification:**

Many concerns addressed, experiments are still small, and lots of corrections to the description were needed.

**Key Questions For Authors:**

My main concern is with the size of the single numerical experiment.

**Limitations:**

There is no discussion of limitations or societal impact. I don't feel that there is need for a significant discussion of societal impact because this is a mostly mathematical paper. The main limitation seems to be the lack of computational evaluation of the method on large problems.

**Strengths And Weaknesses:**

Strengths

The problem studied seems to be very interesting. Constraints which vary by scenario are not often studied in bantit problems.

The mathematical details seem to be correct, where I could understand them. However due to the extremely brief explanation in much of the paper, I could not understand everything.

Limitations

I don't understand why the algorithm is called PAC (probably approximately correct) when the equation (2) states that the algorithm must be *exactly* correct with probability at least 1 - delta.

In (32) N_h is used but only N_h(tau) had been defined previously. I assume N_h here is N_h(tau).

The authors repeatedly cite Gravier and Kauffman (2016) and link to *Optimal Best Arm Identification with Fixed Confidence* but this paper has no Lemma 1. I assume they mean to cite their other paper, *On the Complexity of Best-Arm Identification in Multi-Armed Bandit Models* from 2015.

I don't know why the first inequality of (33) is there when the next one involves taking the infimum over all \tilde{P}. I think the second inequality isn't even true because of this (although it doesn't affect the result of the final inequality).

I don't like the notation in theorem 3.1. Statement (4) seems to say that if you take a sequence of algorithms A_delta which are delta-PAC (in their definition) then liminf (E[tau_delta]) / log(1-delta) is more than H^star(P) which I think is true. But the dependence on delta is hidden in the fact that tau is delta-PAC, so the algorithm somehow depends on delta. This problem seems to repeat itself in (7).

Overall, the notation is very unclear. The main part of the paper is very dense and hard to follow. The proofs contain some mistakes.

Only a single numerical experiment is considered, and it is tiny. The algorithm seems to perform well on this problem, but with only four arms and two covariates I am not sure this says much.

Overall while the problem is fun and most of the math seems to be correct, I was not convinced of the impact of this research.

---

> ### Author Rebuttal · Authors · 2026-03-28
>
> Thank you for your valuable comments and suggestions.
>
> **Comment 1**
> > I don't understand why the algorithm is called PAC (probably approximately correct) when the equation (2) states that the algorithm must be exactly correct with probability at least 1 - delta.
>
> **Response 1**
>
> This is standard terminology in the BAI literature: an algorithm satisfying a $(1-\delta)$ correctness guarantee is commonly called $\delta$-PAC; see, e.g., [1]. We follow this standard convention in the paper.
>
> [1] Garivier A, Kaufmann E. Optimal best arm identification with fixed confidence[C]//Conference on Learning Theory. PMLR, 2016: 998-1027.
>
> **Comment 2**
>
> >In (32) N_h is used but only N_h(tau) had been defined previously. I assume N_h here is N_h(tau).
>
> **Response 2**
>
> Here $N_h$ in Eq. (32) denotes the number of samples allocated to design point $z_h$ up to the stopping time, i.e., $N_h := N_h(\tau)$. We agree that this notation was introduced too implicitly, since only $N_h(t)$ had been defined earlier, and we will clarify this point in the revised version.
>
> **Comment 3**
>
> > The authors repeatedly cite Gravier and Kauffman (2016) and link to Optimal Best Arm Identification with Fixed Confidence but this paper has no Lemma 1. I assume they mean to cite their other paper, On the Complexity of Best-Arm Identification in Multi-Armed Bandit Models from 2015.
>
> **Response 3**
>
> Here we intended to refer to Theorem 1 in Garivier and Kaufmann (2016), whose proof builds on Lemma 1 in Kaufmann et al. (2016). We will clarify this in the revised version and correct the citation typo.
>
> **Comment 4**
>
> > I don't know why the first inequality of (33) is there when the next one involves taking the infimum over all \tilde{P}. I think the second inequality isn't even true because of this (although it doesn't affect the result of the final inequality).
>
> **Response 4**
>
> Thank you for pointing this out. This is a typo: the first inequality in (33) is redundant and should be removed. As you noted, this does not affect the final result or any subsequent argument. We will correct this in the revised version.
>
> **Comment 5**
>
> > I don't like the notation in theorem 3.1. Statement (4) seems to say that if you take a sequence of algorithms A_delta which are delta-PAC (in their definition) then liminf (E[tau_delta]) / log(1-delta) is more than H^star(P) which I think is true. But the dependence on delta is hidden in the fact that tau is delta-PAC, so the algorithm somehow depends on delta. This problem seems to repeat itself in (7).
>
> **Response 5**
>
> The statement in Theorem 3.1(4) characterizes a fundamental lower bound on the sample complexity as $\delta \to 0$, rather than a property of a specific algorithm $\mathcal{A}$. The dependence on $\delta$ is implicit through the stopping time $\tau$, which is in fact a function of $\delta$. As $\delta \to 0$, the stopping time $\tau$ necessarily increases. We omitted this dependence in the notation for simplicity. We will revise the notation to make the dependence of $\tau$ on $\delta$ explicit and avoid potential confusion.
>
> **Comment 6**
>
> > Overall, the notation is very unclear. The main part of the paper is very dense and hard to follow. The proofs contain some mistakes.
>
> **Response 6**
>
> We acknowledge that the notation is dense, but this partly reflects the complexity of the problem itself, which involves constraints, covariate selection, and multi-level optimization. We have already made efforts to keep the notation as concise as possible while maintaining mathematical precision.
>
> Regarding the proofs, aside from the typo in Eq. (32), which does not affect the result, we are not aware of any other technical errors. If there are additional specific concerns, we would be very happy to address them carefully.
>
>
> **Comment 7**
>
> > Only a single numerical experiment is considered, and it is tiny. The algorithm seems to perform well on this problem, but with only four arms and two covariates I am not sure this says much.
>
> **Response 7**
>
> Due to space limitations, the main text presents one representative experiment, but the paper does not rely on a single small-scale numerical study.
>
> In fact, Appendix A.13 contains several additional experiments, including comparisons with multiple baselines over 100 macro-replications, the effect of different covariate selection rules, robustness to different initial design points, and evaluations under different problem scales and noise levels. We also include comparisons with Frank--Wolfe Sampling (FWS) in both running time and statistical performance. In addition, Appendix A.14 presents an application-style experiment on personalized treatment for diabetes management.
>
> These additional results consistently support the same conclusion: DSR outperforms the competing methods in both efficiency and statistical performance across a much broader range of settings than the single representative example shown in the main text.

---

> > ### Author Rebuttal · Reviewer_Cwae · 2026-04-03
> >
> > Thank you for your detailed response.
> >
> > Many of the corrections the authors have made will make the paper more readable.
> >
> > While the appendix contains a few additional experiments, they all also seem to only have 4 arms, and there don't seem to be comments in the paper which address this. However, I am aware that the primary contribution of this paper seems to be the theory, and there exist other published bandits papers which use similar experiment sizes.
> >
> > Is there no way larger experiments can be conducted? With a per-iteration cost of O(MK) it seems to me like K could reasonably be made much larger.

---

> > > ### Author Response · Authors · 2026-04-03
> > >
> > > Thank you again for the opportunity to clarify this point.
> > >
> > > Due to the page limit, we did not discuss in detail in the main text the additional numerical results already presented in Appendix A.13; we will clarify these results and their role more clearly in the revised version.
> > >
> > > Our problem jointly considers covariates and arms, so the effective problem size is $M\times K$. In Appendix A.13, we mainly increase the number of covariates, up to $M=6$ with $K=4$, which is equivalent to a BAI problem with 24 arms. For theory-oriented BAI papers, this is already a nontrivial scale: for example, the experiments in [1] use 5 arms, and those in [2] for linear BAI use 7 arms. We therefore believe our current experiments are sufficient to validate the theoretical findings.
> > >
> > > [1] Garivier A, Kaufmann E. Optimal best arm identification with fixed confidence[C]//Conference on Learning Theory. PMLR, 2016: 998-1027.
> > >
> > > [2] Wang P A, Tzeng R C, Proutiere A. Fast pure exploration via frank-wolfe[J]. Advances in Neural Information Processing Systems, 2021, 34: 5810-5821.
> > >
> > > From a computational perspective, although the per-iteration cost is $O(MK)$, our performance metric is sample complexity, so each run continues until the algorithm identifies the optimal arm with high probability. We also repeat each experiment 100 times to estimate the empirical sample complexity. In our constrained setting, this already leads to a noticeable computational cost, which is why we did not explore much larger $K$ in the current submission.
> > >
> > > More broadly, this is primarily a theory-driven paper. Our main message is to introduce a new dual perspective for a complicated constrained BAI problem: when the primal formulation is difficult to solve, its dual can have a more favorable structure and can be handled efficiently, e.g., via the decomposition method we propose. To the best of our knowledge, this dual-based perspective is new in the BAI literature.
> > >
> > > We appreciate the suggestion on the numerical experiments and will consider adding larger-scale experiments in the revised version.

---

### Official Review · Reviewer_eJSV · 2026-03-09

**Soundness:** 3
**Presentation:** 3
**Significance:** 2
**Originality:** 2
**Overall Recommendation:** 4
**Confidence:** 2

**Summary:**

This papers studies a constrained linear best arm identification problem with covariate selection in the fixed confidence setting, in which there are correlated unknown linear constrained problem depending on each covariate, and for each round the agent selects an arm and covariate so that miss-identification probability of the best arm is minimized for each covariate. The authors first derived a lower bound of the sample complexity (or characteristic time) and proposed a track-and-stop-type algorithm that achieves an asymptotic optimality. Moreover, they introduce an upper bound of the characteristic time $\mathcal{U}^{*}$ for computational efficiency, derived the convex dual problem of $\mathcal{U}^{\ast}$, and proposed an algorithm whose sample complexity is asymptotically given using $\mathcal{U}^{\ast}$. Finally, they conduct numeral experiments in synthetic environments.

**Compliance With Llm Reviewing Policy:**

Affirmed.

**Final Justification:**

Author reponses resolved my questions. Therefore, I will keep my score.

**Key Questions For Authors:**

- Could the authors explain the importance of the problem setting in more detail? In addition, if the arm-covariate pair is linearly independent, is there an existing paper with the same problem setting?
- Could the authors discuss how $\mathcal{U}^{\ast}/\mathcal{H}^{\ast}$ is large.
- The authors state that the lower bound in (Jedra & Proutiere, 2020) is a special case of Theorem 3.1. In my understanding, the first term in Eq (6) matches the result of (Jedra & Proutiere, 2020), but the result of (Jedra & Proutiere, 2020) does not have the third term. Could the authors explain the correspondence between Theorem 3.1 and result of (Jedra & Proutiere, 2020) in more detail? Also, only the third term does not involve an indicator function. Is it correct?

**Limitations:**

This is a theory paper and limitations are stated as assumptions or statements.

**Strengths And Weaknesses:**

## Strengths
- The authors propose asymptotically optimal algorithm.
- Although Algorithm 2 is not optimal, it is novel.

## Weaknesses
- While the first algorithm is asymptotically optimal, the implementable algorithm (second one) is not.
- Since this problem setting is new, it would be better if the authors elaborate on the practical applications.

---

> ### Author Rebuttal · Authors · 2026-03-28
>
> Thank you for your valuable comments and suggestions. We appreciate the opportunity to address your concerns.
>
> > Asymptotically optimal of the implementable algorithm.
>
> **Response 1:**
>
> While Algorithm 1 is theoretically sound and asymptotically optimal, it is not implementable in practice because it relies on solving an intractable optimization problem at each iteration. The main contribution of the paper is precisely the nontrivial development of Algorithm 2, which removes this oracle requirement through our dual-based framework.
>
> Algorithm 2 is computationally efficient and performs well in practice, but this comes at the cost of sacrificing exact asymptotic optimality with respect to the original lower bound. We view this as a meaningful trade-off: the first algorithm serves mainly as a theoretical benchmark, whereas the second provides a practical and rigorously justified solution.
>
> > Practical applications.
>
> **Response 2:**
>
> We have added an application example in Appendix~A.14 based on personalized treatment selection, which illustrates the practical relevance of our setting. We will also further strengthen the discussion of practical applications in the revised version.
>
> > Importance of the problem setting.
>
> We will clarify the importance of this setting more explicitly in the revised version.
>
> This problem is motivated by personalized decision-making tasks in which the best action depends on observable covariates and must also satisfy feasibility constraints. As discussed in the introduction, examples include personalized medicine, where one aims to select the treatment with the highest expected efficacy for a patient profile while keeping side effects below a threshold, and inventory or operations management, where one seeks the best decision under contextual information while ensuring that other performance metrics remain acceptable. In such settings, both active covariate selection and feasibility constraints are essential, and ignoring either aspect can lead to inefficient or even invalid decisions.
>
> To the best of our knowledge, we are not aware of any existing work that studies the same problem setting, even in the case where the arm-covariate features are linearly independent. Existing related work typically considers either unconstrained linear BAI, BAI without covariate selection, or contextual settings where covariates are passively observed rather than actively selected.
>
> > Performance Gap.
>
> **Response 3:**
>
> The ratio $\mathcal{U}^{\ast}/\mathcal{H}^{\ast}$ can be large in pathological instances where the surrogate objective becomes substantially more conservative than the original one. This mainly occurs for arms in $D_3(c_j)$, since in the original objective their contribution is determined by the easier of the two cases, while in the surrogate objective this distinction is removed. As a result, when such arms are more easily identified as suboptimal than as infeasible, the surrogate approximation can become loose.
>
> By contrast, the relaxation is much tighter when the arms in $D_3(c_j)$ are easier to identify through feasibility, for example, when their objective values are close to that of the best arm. This is also consistent with our numerical study on 1000 randomly generated instances, where the ratio is typically close to $1$, although it may become large in specially constructed pathological cases.
>
> > Result Comparison.
>
> Compared with Jedra and Proutiere (2020), our result provides a more refined characterization of how different arm classes contribute to the complexity through both feasibility and optimality.
>
> More specifically, in our setting, the complexity must account for four different roles: the feasibility of the optimal arm, suboptimal feasible arms, infeasible arms with better objective value than the optimal arm, and infeasible arms with worse objective value than the optimal arm. This is why Eq. (6) contains multiple terms: they correspond to the different ways in which an alternative instance can invalidate the correct recommendation.
>
> The result of Jedra and Proutiere (2020) is recovered as a special case when all arms are known to be feasible. In that case, the third term disappears because the optimal arm is always feasible, and the feasibility-related contributions of infeasible arms are no longer present. The remaining term then reduces to the standard linear BAI complexity.
>
> Regarding the indicator functions, this is correct. The third term corresponds to the feasibility of the optimal arm itself, so it is always relevant in the constrained setting and therefore does not require an indicator. By contrast, the other terms only appear for specific arm classes, which is why indicator functions are needed there.

---

> > ### Author Rebuttal · Reviewer_eJSV · 2026-04-02
> >
> > I appreciate the detailed responses that resolved my questions, and I will keep my positive assessment.

---

### Official Review · Reviewer_5Wdj · 2026-03-10

**Soundness:** 2
**Presentation:** 2
**Significance:** 2
**Originality:** 2
**Overall Recommendation:** 3
**Confidence:** 4

**Summary:**

This paper studies linear Best Arm Identification (BAI) under (unknown) linear constraints: at each step of the learning protocol, the learner chooses a design point (a pair of action and covariate) and observes a (linear) reward signal and a (linear) cost signal. The goal of the learner is to stop as soon as they can identify the arm that maximises the reward signal, under a cost signal constraint, for all covariates, with probability at least $1 - \delta$. The paper provides a lower bound on the sample complexity of this task, and a Track-And-Stop algorithm that matches the lower bound asymptotically in $\delta$. Since the proposed Track-And-Stop algorithm is computationally expensive, the paper introduces a surrogate objective model, and a "Duality-Based Decomposition" (DSR) algorithm with better complexity per iteration compared to Track-And-Stop, but with a worse "relaxed" sample complexity upper bound. In the experiments, the paper compares the performance of DSR to different benchmarks, for a small simulated toy example.

**Compliance With Llm Reviewing Policy:**

Affirmed.

**Final Justification:**

As detailed in the discussion below, I believe that my main concerns ($\epsilon$-BAI lower bound, the relaxation gap, and the construction of the design points) were not fully addressed in the rebuttal. That is why I maintain my initial evaluation.

**Key Questions For Authors:**

1) **Assumptions**
  - Assumption 2.1: Having a unique best arm is indeed a classic assumption in BAI. However, the assumption that no arm lies exactly on the constraint is not. Why is this assumption needed? What happens if it is relaxed? Appendix A.3 studies the $\epsilon$-optimal and feasible arms; however, it does not discuss how to identify the exact optimal solution with potentially binding constraints at the optimum. Also, there are problems with the proof of that section, see the questions about the proofs below.

  - Assumption 2.2: It is important to provide more intuition on why the reward and cost means are linear with respect to the *same* feature map $\phi$. What happens if each of these means is linear with respect to different feature maps $\phi$ and $\phi'$?

  - Design points: It is not well explained how the $D$ design points are chosen. Shouldn't the algorithm itself identify these points? On what do they depend?

2) **Remarks on the proofs**:
  - Eq (32) is true because of the independence of the reward and cost signal in Assumption 2.3. This should be highlighted in the proof. Also, more details on the canonical model and how exactly is Lemma 1 from [Garivier & Kaufmann, 2016] is used here should be provided in the proof, as that Lemma is the KL decomposition for finite-armed classic bandits, which again is related to how the design points are chosen (question before).
  - There is no proof of Theorem A.1, that generalises the lower bound from having one constraint to multiple ones.
  - For the proof of [Jedra & Proutiere, 2020] to be applied cleanly in your setting, some properties of the transport function from the lower bound should be checked (continuity, differentiability, convexity etc)
  - Shouldn't the design set $\mathcal{Z}$ span $\mathbb{R}^D$? Otherwise, the inverse of the design matrix is not guaranteed to exist.
  - The result and proof of Appendix A.3 is false: If the goal is to identify "an $\epsilon$-optimal solution" of the optimisation problem, then the problem becomes an $\epsilon$-BAI problem, for which the best known lower bound of [Degenne and Koolen, 2019] is first only asymptotic in $\delta$, and second, contains an additional $sup_{i \in i^\star(c_j, P, eps)}$ in the lower bound, where $i^\star(c_j, P, eps)$ is the set of indexes that are $\epsilon$-optimal for context $c_j$ in P. The proof in Appendix A.3 is false, because it does not adapt the notion of the alternative sets to the new notion of recommendation, and still uses the old $i^\star$.

3) **DSR vs Track-and-Stop**:
  - The "Relaxation Gap Analysis" of Appendix A.7 is unsatisfactory: how big can the $\gamma$ be? In Line 300, left column, you say "e establish a constant relaxation gap for some problem instance-dependent constant C > 1." The wording " problem instance-dependent constant " does not make sense.
  - How does DSR perform compared to TS emprically in terms of sample complexity, and running time, in your toy examples?
  - A classic technique to overcome the computational burden of TS is using Lazy TS (see [Jedra & Proutiere, 2020]). How does DSR compare to Lazy TS?

**Limitations:**

There is no limitations section in the paper.

**Strengths And Weaknesses:**

Strengths:
- The paper studies an interesting setting
- The paper provides a complete set of results: matching sample complexity lower and upper bounds, a surrogate objective and a better optimised algorithm
- The paper also provides numerical experiments

Weaknesses:
- Some assumptions are not well-motivated/can be very strong (see questions below)
- Some parts of the proofs are false/can be rewritten better (see questions below)
- The techniques used to generate the sample complexity lower and upper bounds are straightforward applications of classic techniques in the BAI literature ([Garivier & Kaufmann, 2016] for the lower bound, and [Jedra & Proutiere, 2020] for the upper bound). The extra difficulty introduced by having (unknown) linear constraints in the linear BAI problem is not clearly highlighted in the proof sketches, and seems to be just having to deal with extra KKT conditions in the proofs
- Both theoretically and experimentally, it is not clear how better DSR is compared to Track and Stop (see questions below). Also, the intuition behind the proposed surrogate objective is not explained at all

---

> ### Author Rebuttal · Authors · 2026-03-28
>
> Thank you for your valuable comments and suggestions. We appreciate the opportunity to address your concerns.
>
> > Assumption 2.1
>
> **Response 1**
>
> This assumption is analogous to the standard uniqueness assumption in BAI. If an arm lies exactly on the constraint boundary, then noisy observations generally make exact feasible/infeasible identification impossible, just as exact best-arm identification is impossible under ties. Similar assumptions also appear in constrained simulation optimization [1].
>
> If this assumption is removed, the objective should also be relaxed. That is why Appendix A.3 studies the goal of identifying an  $\epsilon$-optimal and feasible arm when boundary cases are allowed.
>
> [1] Hunter S R, Pasupathy R. Optimal sampling laws for stochastically constrained simulation optimization on finite sets, 2013.
>
> > Assumption 2.2
>
> **Response 2**
>
> Using the same feature map for reward and cost is mainly a simplification. It lets the two models share one design matrix, which streamlines notation and estimation. The theory itself does not rely on this. If reward and cost use different feature maps, the extension is straightforward by introducing separate design matrices and estimating the parameters separately.
>
> > Design points.
>
> **Response 3**
>
> The design points are a fixed set of sampling points used to estimate the unknown linear parameters. There are many standard ways to choose $Z$, such as space-filling designs, Latin hypercube sampling, and optimal design criteria [2]. Our experiments show that different initial design points only mildly affect sample complexity, suggesting that DSR is fairly robust to this choice. See Appendix A.13.
>
> [2] Ankenman B, Nelson B L, Staum J. Stochastic kriging for simulation metamodeling, 2010.
>
> > More proof details on Eq (32).
>
> **Response 4**
>
> We will give more details in the proof on the independence assumption and on how the change-of-measure lemma is applied. In particular, the design points are fixed in advance, using the method described in Response 3.
>
> > Proof of Theorem A.1
>
> **Response 5**
>
> The proof of Theorem A.1 follows essentially the same argument as Theorem 3.1. Extending from one constraint to multiple constraints is straightforward once the feasible and infeasible sets are defined for each arm-covariate pair. We have clarified this in Appendix A.4 and will include a full proof in the revised version.
>
> >  Proof of [Jedra & Proutiere, 2020]
>
> **Response 6**
>
> Proposition 3.3 does not require new problem-specific regularity arguments beyond Jedra and Proutiere (2020). The same proof idea applies, since continuity, differentiability, and convexity come from the same standard properties of the transport function and KL divergence.
>
> We therefore did not include the full proof of Proposition 3.3, since it would largely repeat the argument in Jedra and Proutiere (2020). More importantly, Proposition 3.3 is not a technical contribution; its role is mainly to motivate our dual-based algorithm design.
>
> >  Inverse of the design matrix
>
> **Response 7**
>
> We will clarify in the revised manuscript that $Z$ is assumed to span $\mathbb{R}^D$, so the design matrix is full rank and invertible.
>
> > Proof of Appendix A.3.
>
> **Response 8**
>
> The proof in Appendix A.3 is correct, but the resulting complexity is not the standard $\epsilon$-BAI complexity of Degenne and Koolen (2019).
>
> We did consider extending the framework of Degenne (2019), but this is highly nontrivial. It changes the lower-bound structure, the oracle allocation geometry. As a result, our dual-based method does not directly extend, and we view this generalization as beyond the scope of the paper.
>
> Appendix A.3 instead studies a more conservative complexity (upper bound of the complexity in Degenne(2019)) for handling boundary cases. It considers alternatives where the optimal arm is no longer $\epsilon$ optimal and feasible. This still guarantees $\delta$-PAC remains fully consistent with our dual-based method, and incorporates the effect of $\epsilon$. Similar approaches to $\epsilon$-identification have also been used in [3]. We will clarify this distinction in the revised version
>
> [3] Taupin J, Jedra Y, Proutiere A. Best policy identification in discounted linear mdps[C], 2023.
>
> > Relaxation Gap
>
> **Response 9**
>
> See Response 2 to Reviewer m8LA.
>
> > DSR vs TS and Lazy TS
>
> Our theory and experiments both explain why DSR is preferable to TS. In theory, Appendix A.12 compares the per-iteration computational complexity of the two methods.  DSR performs a single gradient update at each iteration, and its computational cost is substantially lower. Practically, TS is computationally infeasible even in our small-scale experiments. We therefore compare DSR with FWS, an efficient method with comparable performance with Lazy TS [4], and show in Appendix A.13 that DSR performs better in both runtime and statistical efficiency.
>
> [4] Wang P A, Tzeng R C, Proutiere A. Fast pure exploration via frank-wolfe, 2021.

---

> > ### Author Rebuttal · Reviewer_5Wdj · 2026-04-02
> >
> > I thank the authors for their detailed rebuttal. However, my core concerns (the $\epsilon$-BAI lower bound, controlling the relaxation gap, choosing the spanning design points) are still not fully addressed by the rebuttal, and combined with the multiple inaccuracies pointed out by the other reviewers, would require a significant update to the paper.

---

> > > ### Author Response · Authors · 2026-04-02
> > >
> > > Thank you for the response. We respectfully note that these points have already been carefully addressed in the rebuttal.
> > >
> > > First, the $\epsilon$-optimal discussion is not a main contribution of the paper. Its purpose is only to provide one possible way to handle boundary cases when exact identification is not statistically learnable. In this sense, Appendix A.3 is auxiliary rather than central. The proof there is technically correct for the relaxed objective we consider, and similar approaches have also been used in the literature [3]. The resulting complexity should not be interpreted as the $\epsilon$-BAI lower bound in the sense of Degenne and Koolen (2019). A full extension of that lower-bound theory to our constrained setting with covariates is nontrivial and beyond the scope of the current paper. Our main contribution is instead the dual perspective and the resulting algorithmic framework for constrained BAI with covariates.
> > >
> > > Second, regarding the relaxation gap, our current surrogate turns a ratio-computation problem that is not efficiently solvable into one that is computationally tractable, while still enjoying an instance-dependent upper bound. Empirically, this relaxation does not lead to a noticeable performance loss in our numerical studies. More importantly, our analysis also yields a sharper constant upper bound via a finer classification using the $D_3(c_j)$ set, as discussed in Response 2 to Reviewer m8LA.
> > >
> > > Third, the issue of choosing spanning design points only requires clarification, not additional technical development. We only need to state explicitly that the support of the design set $Z$ spans $\mathbb{R}^D$, so that the design matrix is full rank. This is a standard identifiability assumption in linear BAI and does not require any change to the method itself.
> > >
> > > We hope this clarifies why these points, while worth clarifying in revision, do not affect the main technical contribution of the paper. If there are other specific concerns that remain unresolved, we would be very happy to address them directly.

---

### Official Review · Reviewer_m8LA · 2026-03-11

**Soundness:** 2
**Presentation:** 3
**Significance:** 4
**Originality:** 4
**Overall Recommendation:** 4
**Confidence:** 2

**Summary:**

This paper introduces a novel variant of the best arm identification (BAI) problem that incorporates both hard constraints and active covariate selection. At each round, the agent selects a design point-- i.e., a pair consisting of an arm and a covariate—and receives two noisy observations: one corresponding to the objective reward and another to a constraint signal. The goal is to, for every covariate, identify the arm that maximizes the expected objective while satisfying a mean constraint, with probability at least 1-\delta, and using as few samples as possible in expectation.
The authors derive an instance-dependent lower bound on the sample complexity that explicitly accounts for the feasibility and optimality structure of arms. They propose a Track-and-Stop-style algorithm that asymptotically matches this bound, albeit at considerable computational cost. To address this, they introduce a relaxed version of the lower bound and formulate its convex dual. Leveraging this dual representation, they develop a duality-based decomposition algorithm that updates two coordinates per iteration via a single gradient step. This approach achieves high computational efficiency while asymptotically attaining the relaxed bound. Empirical results are provided to illustrate the practical behavior of the proposed method. The work connects meaningfully with existing literature on BAI, constrained bandits, and experimental design with covariates.

**Compliance With Llm Reviewing Policy:**

Affirmed.

**Final Justification:**

This paper introduces a novel and well-motivated formulation of the best-arm identification problem that integrates constraints and active covariate selection. The theoretical analysis provides a new instance-dependent lower bound, and the proposed duality-based algorithmic framework is a clear and original contribution.

My initial concerns primarily revolved around the clarity and completeness of the theoretical presentation: the inconsistent notation in the weighted least-squares estimator, the theoretical guarantee for the relaxation gap, and the explicit statement of the design matrix invertibility condition.

The authors have satisfactorily addressed these points in their rebuttal and subsequent comments. They have clarified that the factor of 1/2 is a notation inconsistency that will be corrected, proposed a refined analysis to provide a uniform constant-factor bound for the relaxation gap, and committed to explicitly stating the full-rank design condition in the main text.

These are minor revisions that do not affect the core contributions of the work. Given the authors' clear commitments to address these points in the final version, I now recommend Weak Accept. The paper makes a solid contribution to the bandit literature, and the identified issues can be resolved through careful editing and the promised clarifications.

**Key Questions For Authors:**

1. Estimator normalization
In Theorem 3.1 and surrounding discussion, the estimator θ(t) appears inconsistent with the definition Λ(ω). A standard weighted least squares derivation would include a factor of 1/2. Is this omission intentional (e.g., absorbed into the definition of Lambda)? If not, how does it affect the subsequent theory and experiments?
2. Relaxation gap
The surrogate complexity U^∗(P)is used to motivate Algorithm 2, but the gap between U^∗(P) and the true lower bound H^∗(P) is instance-dependent and potentially unbounded. Can the authors rule out scenarios where this gap grows arbitrarily large? If not, how should we interpret the asymptotic optimality claim in Theorem 4.2?
3. Convexity justification
Lemma 3.6 relies on an equivalence that assumes Φ is invertible and that the parameter estimators are correctly specified. Are these assumptions necessary? Could the authors provide a more robust proof of convexity that holds under milder conditions (e.g., when Φ is rank-deficient)?
4. Feasibility of baselines
The BCsR baseline ignores constraint information during exploration but enforces feasibility at recommendation time. How frequently does it output infeasible arms due to estimation error? Reporting this would better highlight the advantage of constraint-aware sampling.

**Limitations:**

The authors transparently acknowledge key limitations: Gaussian noise, linearity of rewards/constraints, and the need to discretize continuous covariates. Extensions to epsilon-optimality and multiple constraints are sketched in the appendix. Given the methodological nature of the work, the omission of a dedicated societal impact discussion is reasonable. The primary applications—e.g., personalized treatment selection—are broadly beneficial, and no immediate harmful use cases are apparent.

**Strengths And Weaknesses:**

*Strengths:*

Soundness.
1.The paper rigorously formulates the BAI problem with constraints and active covariate selection, establishing a clear mathematical framework.
2.The instance-dependent lower bound derivation in Section 3 appears mathematically sound and provides a meaningful theoretical benchmark.
3.Algorithm 1 is clearly presented with well-defined steps for parameter estimation and sampling.

Presentation.
1.The paper is exceptionally well-organized with clear structure: introduction → problem formulation → theoretical analysis → algorithms → experiments
2. Notation is systematic and consistent throughout, making the complex mathematical development accessible.
3. Figures and pseudocode enhance understanding of the algorithms and their performance.
4. The related work section is comprehensive and accurately situates the contribution within the broader literature.
5. Writing is clear, precise, and professionally executed

Significance.
1. The problem setting is highly relevant to real-world applications including precision medicine, targeted advertising, and personalized decision-making.
2. The unification of constrained BAI with active covariate selection represents an important conceptual advancement.
3. The duality-based algorithmic framework offers promising trade-offs between statistical efficiency and computational tractability.
4. It opens new research directions that generalize both classical and linear BAI frameworks.
5. It has potential for substantial impact on both theoretical and applied communities.

Originality:
1. The paper formulates a novel BAI problem that jointly considers constraints and active covariate choice.
2. It introduces a refined lower bound that distinguishes arms based on their feasibility status.
3. The authors propose a decomposition algorithm grounded in convex duality with innovative two-coordinate update mechanisms.
4. The integration of convex duality into the BAI framework represents a genuinely inventive methodological approach.
5. The work provides fresh analytical insights that depart from conventional oracle-based solvers.

*Weaknesses:*

Soundness.
1. The weighted least squares estimator appears to omit a factor of 1/2, potentially biasing parameter estimates and undermining the validity of subsequent analyses.
2. The surrogate complexity measure U(P) lacks uniform bounds on γ, allowing it to be arbitrarily larger than H(P) in pathological cases, which questions the practical relevance of asymptotic optimality guarantees.
3. Lemma 3.6's convexity claim depends on unverified assumptions about design matrix invertibility and parameter estimation correctness.
4. The argument may fail in high-dimensional or redundant feature settings where Φ is rank-deficient, challenging the convexity conclusion.
5. Collectively, these technical issues substantially undermine confidence in the theoretical guarantees' validity.

Presentation.
1. Some cross-references in the appendix refer to equations defined later in the text, creating logical flow issues.
2. Equation (28) contains a typographical error in the alternative set definition.
3. Critical assumptions regarding design matrix invertibility and parameter estimation correctness are not properly justified in the main text.
4. The analysis does not adequately address pathological scenarios that could invalidate the convexity argument.
5. The methodology explanation could benefit from more intuitive explanations of the duality-based approach for broader accessibility.

 Significance.
1.The unproven convexity and potentially large relaxation gap limit the immediate practical applicability of the proposed methods.
2. The identified theoretical issues raise significant concerns about algorithm robustness in real-world implementations.
3. Without resolution of the theoretical problems, the broader impact and generalization of the contributions remain uncertain.
4. While experimental results are presented, their interpretation depends on theoretical foundations that require further validation.
5. The unresolved technical issues may create adoption barriers for practitioners in the field.

Originality.
1. While the convex duality approach is inventive, it depends heavily on assumptions that may not always hold in practice.
2. The original formulation's novelty is compromised by the identified correctness issues that need resolution.
3. The implementation's innovative aspects are contingent on successfully addressing the theoretical problems identified.
4. The unification of multiple BAI frameworks introduces complexity that may limit widespread adoption in practical applications.
5. While the problem formulation itself is original, the algorithmic solution requires further validation to establish its full potential.

---

> ### Author Rebuttal · Authors · 2026-03-28
>
> Thank you for your valuable comments and suggestions. We appreciate the opportunity to address your concerns.
>
> >  Weighted least squares estimator
>
> **Response 1**
>
> This is a notation typo: the factor $1/2$ should be removed from the definition of $\Lambda(\omega)$. With this correction, equation (10) matches the standard weighted least-squares estimator, and the complexity term $\mathcal{H}$ is also consistent. We will revise the notation throughout the paper to avoid this ambiguity.
>
> > Relaxation Gap
>
> **Response 2**
>
> The upper bound is instance-dependent. In our random-instance study (1000 generated instances for each $b$; Appendix A.7), the approximation ratio is empirically close to $1$, suggesting that the surrogate is practically tight for the instances we tested. The numerical results also show that this relaxation does not hurt performance.
>
> Using the same proof idea, we can further refine the surrogate bound by partitioning the set $\mathcal D_3(c_j)$ into two subclasses, depending on whether the reward-gap term or the feasibility-gap term is smaller. This refinement rules out the pathological case in which identifying an arm as suboptimal is easier than identifying it as infeasible, and leads to a uniform constant-factor bound of $2$. The rest of the algorithm design and analysis remains essentially unchanged. We will clarify this refinement in the appendix of the revised version.
>
> >  Lemma 3.6's convexity
>
> **Response 3**
>
> The convexity argument in Lemma 3.6 does not rely on accurate parameter estimation; it is an intrinsic property of the optimization problem and holds for any problem instance $\mathcal{P}\in\mathcal{S}$. The main requirement in the derivation is the invertibility of the design matrix. We will clarify that this is ensured by choosing design points $Z$ that span $\mathbb{R}^D$.
>
> > Rank-deficient
>
> **Response 4**
>
> A full rank design matrix is a widely used assumption in linear BAI literature [1]. Regarding the rank-deficient case, to the best of our knowledge, this setting is rarely studied in linear BAI, since the usual goal is to obtain $\delta$-PAC guarantees, consistency of the parameter estimators, and instance-dependent asymptotic optimality, all of which typically require identifiability. While rank-deficient designs are important and have been discussed in regret minimization, whether one can still obtain $\delta$-PAC guarantees and problem-dependent optimality in such settings remains, in our view, an open question and is beyond the scope of this paper.
>
> [1] Jedra Y, Proutiere A. Optimal best-arm identification in linear bandits, 2020.
>
> >  Cross-references
>
> **Response 5**
>
> This is a cross-reference typo caused by equation numbering. We will correct the appendix references to ensure the logical flow is clear.
>
> >  Equation (28)
>
> **Response 6**
>
> We will correct the notation so that the same covariate index is used throughout the definition.
>
> > Critical assumptions
>
> **Response 7**
>
> We will clarify in the main text that the required invertibility condition, which is the standard identifiability assumption in linear BAI. By contrast, correctness of parameter estimation is not assumed; it is a consequence of the analysis.
>
> > Intuitive explanations of the duality-based approach
>
> **Response 8**
>
> We will add more intuition to the presentation of the duality-based approach. The key idea is that the primal problem requires directly computing the optimal sampling ratio under coupled feasibility, optimality, and covariate-selection constraints, which leads to a complicated nonsmooth optimization. The dual reformulation reveals a cleaner structure: it decouples these interactions and yields subproblems that can be handled efficiently via decomposition. In the revision, we will add a more intuitive high-level explanation of this primal-versus-dual perspective and further clarify the role of the dual variables.
>
> > broader impact
>
> Our paper already establishes several core technical contributions: a novel instance-dependent lower bound that jointly captures covariates, feasibility, and optimality; a surrogate objective with a controlled relaxation gap; a duality-based analysis with convexity and strong duality; and a convergence guarantee showing that the proposed method attains the relaxed bound. While extensions such as the low-rank setting are interesting, they are beyond the scope of this paper. More broadly, our results suggest that when the primal ratio-computation problem is hard in BAI, the dual problem may offer a more tractable structure for algorithm design.
>
> >BCSR algorithm
>
> From Figure~2, we can directly read off the overall recommendation error rate as $1-\mathrm{PCI}$. For example, for BCSR this is about $18\%$ at 2000 samples, $13\%$ at 3000 samples, and $11\%$ at 4000 samples. These errors include both recommending infeasible arms and recommending feasible but suboptimal arms.

---

> > ### Author Rebuttal · Reviewer_m8LA · 2026-04-02
> >
> > Thank you for your detailed rebuttal. While some concerns have been partially clarified, two key points still require further clarification, including a suggestion and follow-up questions as below:
> >
> > 1. Regarding the Weighted least squares estimator: We suggest the factor 1/2 in \(\Lambda(w)\) is not a simple typo, as it appears in Theorem 3.1’s proof, the algorithm pseudocode, and most numerical experiments. We recommend revising all relevant parts and clarifying how this correction ensures consistency across proofs, the algorithm, and experiments.
> >
> > 2. Regarding the Relaxation Gap: While partitioning \(D_{\text{glc}}\) is interesting, the mentioned "uniform constant factor bound" is confusing: you describe it as instance-dependent, yet call it uniform. Please resolve this contradiction and also address the high-dimensional performance concern.
> >
> > We look forward to your further clarification.

---

> > > ### Author Response · Authors · 2026-04-02
> > >
> > > Thank you very much for the follow-up and for giving us the opportunity to clarify these points further.
> > >
> > > **Regarding the weighted least-squares estimator**, we carefully re-checked the proof of Theorem 3.1, the algorithm description, the implementation, and the numerical experiments. The factor $1/2$ originates from the Gaussian KL expression. In the lower-bound derivation, this constant was absorbed into the definition of $\Lambda(\omega)$, which is why it does not appear explicitly in Theorem 3.1. However, when writing the weighted least-squares estimator, we reused the same notation $\Lambda(\omega)$ for the design matrix, whereas the standard design matrix in weighted least squares does not include this extra factor $1/2$. Thus, the issue is a notation inconsistency rather than a substantive problem in the theory or implementation.
> > >
> > > This can be resolved cleanly in either of two equivalent ways: (i) revise the definition of $\Lambda(\omega)$ and write the factor $1/2$ explicitly in the lower bound, which does not affect the optimizer since it is only a constant scaling; or (ii) introduce a separate notation for the weighted least-squares design matrix in the estimator, so that the lower-bound quantity and the estimation matrix are no longer overloaded. In our implementation, we use the standard weighted least-squares estimator, so there is no incorrect scaling in the code or experiments (as demonstrated in Figure 2, where the algorithm appears to be consistent). We will revise the notation in Theorem 3.1 to make the notation fully consistent. We hope this notation inconsistency will not obscure the main contributions of the paper.
> > >
> > > **Regarding the relaxation gap**, our point is that the current relaxation is already sufficiently strong for the purposes of this paper. First, it turns a ratio-computation problem that is computationally intractable in practice into one that can be solved efficiently. Second, it comes with an instance-dependent bound relating the surrogate complexity to the original one. Third, the numerical results show that the resulting approximation performs very well in practice: it significantly outperforms existing methods and we do not observe a noticeable loss caused by the relaxation.
> > >
> > > The remaining question is whether one can also derive a uniform constant-factor bound. As we explained in the rebuttal, our current analysis can in fact be extended to obtain a constant bound of $2$. The main difficulty in the original problem comes from arms in the set $D_3(c_j)$: such arms can be eliminated either by certifying them as suboptimal or by certifying them as infeasible. The coupling of these two mechanisms makes the exact optimization problem difficult. The current surrogate simplifies this by only accounting for elimination through infeasibility, while dropping the suboptimality term. The potentially pathological case is therefore one where an arm could have been removed cheaply via suboptimality, but the surrogate instead allocates additional samples to certify infeasibility. A direct refinement is to further partition the arms in $D_3(c_j)$ according to whether suboptimality or infeasibility is easier to establish; this yields a constant-factor (constant is 2) bound, at the expense of higher computational cost.
> > >
> > > For this reason, we do not view the absence of a uniform constant bound in the current presentation as a fundamental limitation. The present relaxation is already computationally effective, theoretically controlled, and empirically strong. Moreover, a constant-factor guarantee can be obtained through the same analytical framework, without changing the overall algorithmic or theoretical perspective of the paper.
> > >
> > > **Regarding the high-dimensional performance** ：
> > > As long as the design points span $\mathbb{R}^D$, we can obtain a consistent estimator and hence the corresponding theoretical guarantees continue to hold. As noted in our rebuttal, the full-rank design condition is standard in the linear BAI literature. A more challenging regime is when the number of design points is smaller than the feature dimension. One possible way to handle this is through representation learning: if the data are generated from a lower-dimensional latent feature space, then one may first learn the representation and subsequently apply our algorithm in that latent space to identify the best arm. Without such additional structure, we believe it is difficult to obtain meaningful BAI guarantees, since controlling the mis-identification probability and sample complexity fundamentally relies on consistent parameter estimation. To the best of our knowledge, this remains an open problem.
> > >
> > > Finally, we sincerely thank the reviewer for the careful reading and thoughtful follow-up questions. The issues you raised will all be clarified in the revised version.

---

### Decision · Program_Chairs · 2026-04-30

**Decision:**

Accept (regular)

**Comment:**

This paper studies a constrained linear best arm identification problem with covariate selection in the fixed-confidence setting, where the objective and constraint is linear to the known embedding. In particular, the algorithm designer is interested in identifying the average best arm subject to feasibility constraint with minimal sample cost.

The lower bound and corresponding upper bound seems to be from standard change-of-measure argument, and track and stop algorithm follows the optimal sample allocation with plugin estimators. The dual perspective itself has clear precedents (e.g., Li et al. ICML'21, You et al. COLT'23).  Discussion after the author's response considers the main contribution concluded that the main contribution is its application on the complex problem and the practically optimal relaxation with constant bound. Reviewer 5Wdj describes the negative review with several technical points ($\varepsilon$-BAI lower bound and 2-constant relaxation) still remains but considers them fixable. Given this, I recommends an acceptance.